Analysis

# SpliceVault predicts the precise nature of variant-associated mis-splicing

Ruebena Dawes [1,2,3,4], Adam M. Bournazos [1,2,3,4], Samantha J. Bryen [1,2,3], Shobhana Bommireddipalli[1,3], Rhett G. Marchant[1,2,3], Himanshu Joshi[1,3,5] & Sandra T. Cooper [1,2,3,5] ✉

Even for essential splice-site variants that are almost guaranteed to alter mRNA splicing, no current method can reliably predict whether exon-skipping, cryptic activation or multiple events will result, greatly complicating clinical interpretation of pathogenicity. Strikingly, ranking the four most common unannotated splicing events across 335,663 reference RNA-sequencing (RNA-seq) samples (300K-RNA Top-4) predicts the nature of variant-associated mis-splicing with 92% sensitivity. The 300K-RNA Top-4 events correctly identify 96% of exon-skipping events and 86% of cryptic splice sites for 140 clinical cases subject to RNA testing, showing higher sensitivity and positive predictive value than SpliceAI. Notably, RNA re-analyses showed we had missed 300K-RNA Top-4 events for several clinical cases tested before the development of this empirical predictive method. Simply, mis-splicing events that happen around a splice site in RNA-seq data are those most likely to be activated by a splice-site variant. The SpliceVault web portal allows users easy access to 300K-RNA for informed splice-site variant interpretation and classification.

Genetic variants that induce mis-splicing of precursor messenger RNA (pre-mRNA) are emerging as a common cause of inherited disorders[1,2]. Pre-mRNA splicing is a process that precisely removes noncoding intronic sequences and connects coding exons together to make the mature mRNA blueprint that encodes each protein (or noncoding RNA). The splicing reactions are performed by a large, protein–RNA complex called the spliceosome[3], which recognizes conserved splice-site motifs at either end of the intron. There are the following three main ways that mis-splicing can occur: one or more exons may be skipped, one or more cryptic splice sites may be activated or one or more introns may be retained, or a mixture.

Variants impacting essential splice sites (ESs), the almost invariant GT-AG flanking each intron, are almost guaranteed to induce mis-splicing. Variants affecting the wider splice-site motif can also disrupt pre-mRNA splicing, although because of greater sequence diversity, it is much harder to be certain if 'extended splice-site' variants will elicit mis-splicing. Due to triplet codons encoding each amino acid, mis-splicing of pre-mRNA that inserts or deletes nucleotides (nt) commonly induces a frameshift or encodes a premature termination codon (PTC). Consequently, the clinical guidelines for classifying genetic variants[4] (the American College of Medical Genetics and Genomics and the Association for Molecular Pathology (ACMG–AMP) guidelines) allow an ES variant to be considered under the 'null variant' code, called very strong evidence for pathogenicity (PVS1). PVS1 is used to classify single nucleotide variants (SNVs) and small insertions and deletions that similarly introduce a PTC[4]. In 2018, revised PVS1 guidelines recommended application of the PVS1 code for ES variants, at varying strengths, based upon theoretical assertion of the likelihood of 'null outcomes' arising from exon skipping, intron retention and use of any cryptic splice site within 20 nt (ref. [5]). However, only ~20% of variant-activated cryptic donors are within 20 nt (ref. [6]). Consideration of cryptic splice-site activation across a larger distance

[1]Kids Neuroscience Centre, Kids Research, Children's Hospital at Westmead, Sydney, New South Wales, Australia. [2]Discipline of Child and Adolescent Health, Faculty of Health and Medicine, University of Sydney, Sydney, New South Wales, Australia. [3]The Children's Medical Research Institute, Sydney, New South Wales, Australia. [4]These authors contributed equally: Ruebena Dawes, Adam Bournazos. [5]These authors jointly supervised this work: Himanshu Joshi, Sandra Cooper. ✉e-mail: sandra.cooper@sydney.edu.au

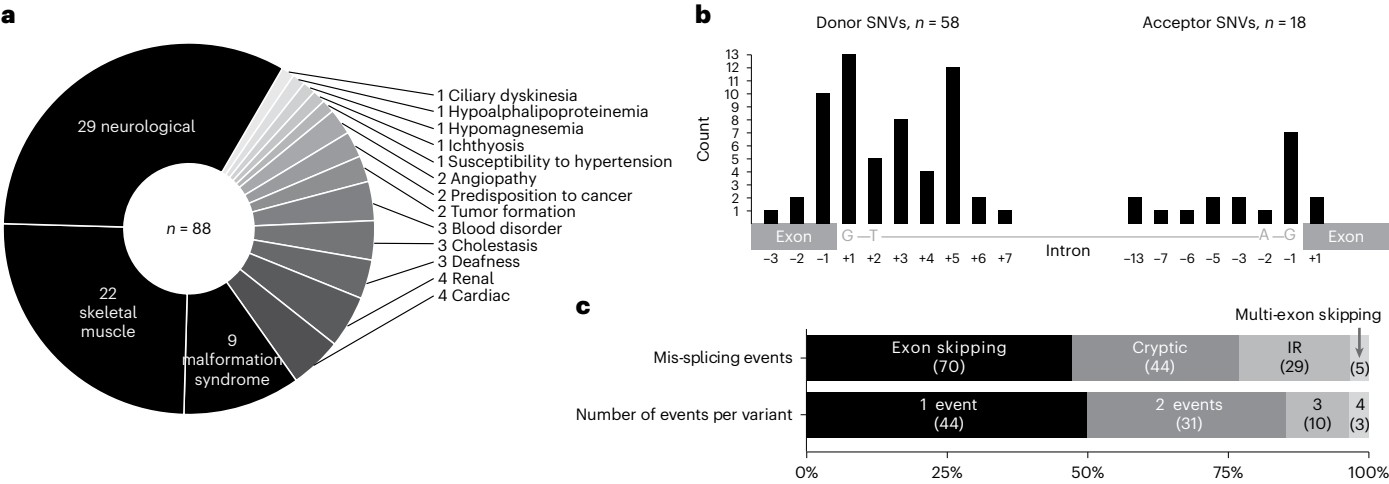

**Fig. 1 | Variant cohort details. a,b**, Phenotypes associated with 88 experimentally verified clinical splicing variants (**a**) and position of the 76/88 variants that are SNVs relative to the ESs (**b**). **c**, Nature of 148 unannotated splicing events (mis-splicing) induced by the 88 variants. IR = intron retention.

window is unfeasible in diagnostic genetic pathology, due to the large number of potential cryptic splice sites present in the genome. In addition, factors that induce multi-exon skipping (or retention of multiple introns) associated with some splice-altering variants are unknown. Currently, there is no reliable way to know exactly how a variant will disrupt pre-mRNA splicing, to assert with confidence if outcomes are likely to encode a PTC, or not.

Our laboratory is an Australasian reference center for RNA diagnostics, where we test RNA isolated from human specimens (blood cells, skin fibroblasts, urine cells or an available biopsy specimen) to experimentally validate if and how a candidate splice-altering variant identified by genomic sequencing alters pre-mRNA splicing[7]. We routinely interrogate RNA sequencing (RNA-seq) data from controls or disease controls (in-house, from Genotype-Tissue Expression dataset (GTEx[8] or ENCODE[9]) to assess alternative splicing of the target gene between the manifesting tissues and specimens commonly collected in a clinical setting, to determine their suitability for RNA testing. We observed that the predominant, variant-associated mis-spliced transcript(s) identified in specimens from affected individuals and heterozygotes were often observed as rare, stochastic splice junctions in control RNA-seq data. Brandão and colleagues detailed a similar finding, with dominant variant-induced mis-spliced *BRCA1* or *BRCA2* transcripts often seen as rare events in disease controls[10]. Additionally, Kremer et al.[11] found that splicing 'noise' often forecasts the location of variant-activated pseudoexons and reasoned that a population-based RNA-seq compendium could aid in variant prioritization.

In ref.[6], we analyzed 5,145 variants activating cryptic splice sites and established that 87% of activated cryptic splice sites are those detected as rare, unannotated splice junctions in 40,233 RNA-seq samples from GTEx[8] and Intropolis[12] (40K-RNA database[6]). The key insight is that cryptic donors activated by genetic variants are also seen as rare events in population-based RNA-seq data, which led us to explore whether other forms of variant-associated mis-splicing may be predicted by quantifying the relative prevalence of stochastic, natural and unannotated splicing events (referred to hereafter as mis-splicing events).

We therefore created 300K-RNA, an expanded resource detailing the most common unannotated splicing events local to each exon–intron junction of Ensembl[13] and RefSeq[14] transcripts, based on splice junctions detected across 335,663 publicly available RNA-seq samples from GTEx[8] and Sequence Read Archive (SRA)[15], uniformly processed in the recount3 project[16] (300K-RNA). 300K-RNA is updated to the

GRCh38 genome assembly and is hosted in a web resource called Splice-Vault, together with 40K-RNA (GRCh37)[6]. Unannotated splice junctions in 300K-RNA constitute evidence that a splicing event is biologically possible, with the requisite constellation of features for the splicing reactions to be executed, for example, recognizable and usable splice sites, suitable exon and intron lengths[17], the correct balance of exonic and intronic splicing enhancer and suppressor elements[18]. Our central hypothesis is that a genetic variant impeding spliceosomal use of an annotated splice site is most likely to enhance mis-splicing events that occur naturally.

We demonstrate that by knowing what mis-splicing events most commonly happen around a splice site, we can predict what mis-splicing events will happen when a genetic variant disrupts that splice site. The 300K-RNA Top-4 events correctly identify 96% of exon-skipping events (including multi-exon skipping) and 86% of activated cryptic splice sites induced by 88 variants in 74 genes for 140 affected individuals or heterozygotes subject to RNA diagnostics. We additionally compare the 300K-RNA empirical method with the deep-learning algorithm SpliceAI[19], applying custom interpretive rules to SpliceAI Δ-scores ±5,000 nt of variants to infer predictions of exon skipping, intron retention and cryptic splice-site activation.

## Results

### A set of experimentally verified splice-altering variants

We performed a retrospective analysis of 88 splice-site variants across 74 genes that are confirmed by RNA diagnostics[7] to disrupt pre-mRNA splicing (Supplementary Table 1). Variants affecting an annotated splice site and demonstrated to activate exon-skipping or cryptic splice-site use were included in this study. Variants creating or modifying a cryptic splice site (inappropriate for our method[6]) were excluded.

Reverse transcription PCR (RT-PCR) and/or RNA-seq were performed on RNA isolated from clinically accessible specimens from 140 affected individuals or heterozygotes with diverse Mendelian conditions[7,20–23] (Fig. 1a and Supplementary Table 1). The majority of probands had neurological (n = 29), skeletal muscle (n = 22) or malformation syndrome (n = 9) phenotypes. Thirty-two percent of variants affect the essential GT (n = 19) or AG (n = 9) splice sites and 68% affect the extended donor or acceptor splice-site regions. The dataset included 76 SNVs, four insertions, six deletions and two deletion–insertion variants (Fig. 1b).

Half of the variants (44/88) induced two or more mis-splicing events (Fig. 1c). Variants most frequently caused skipping of a single

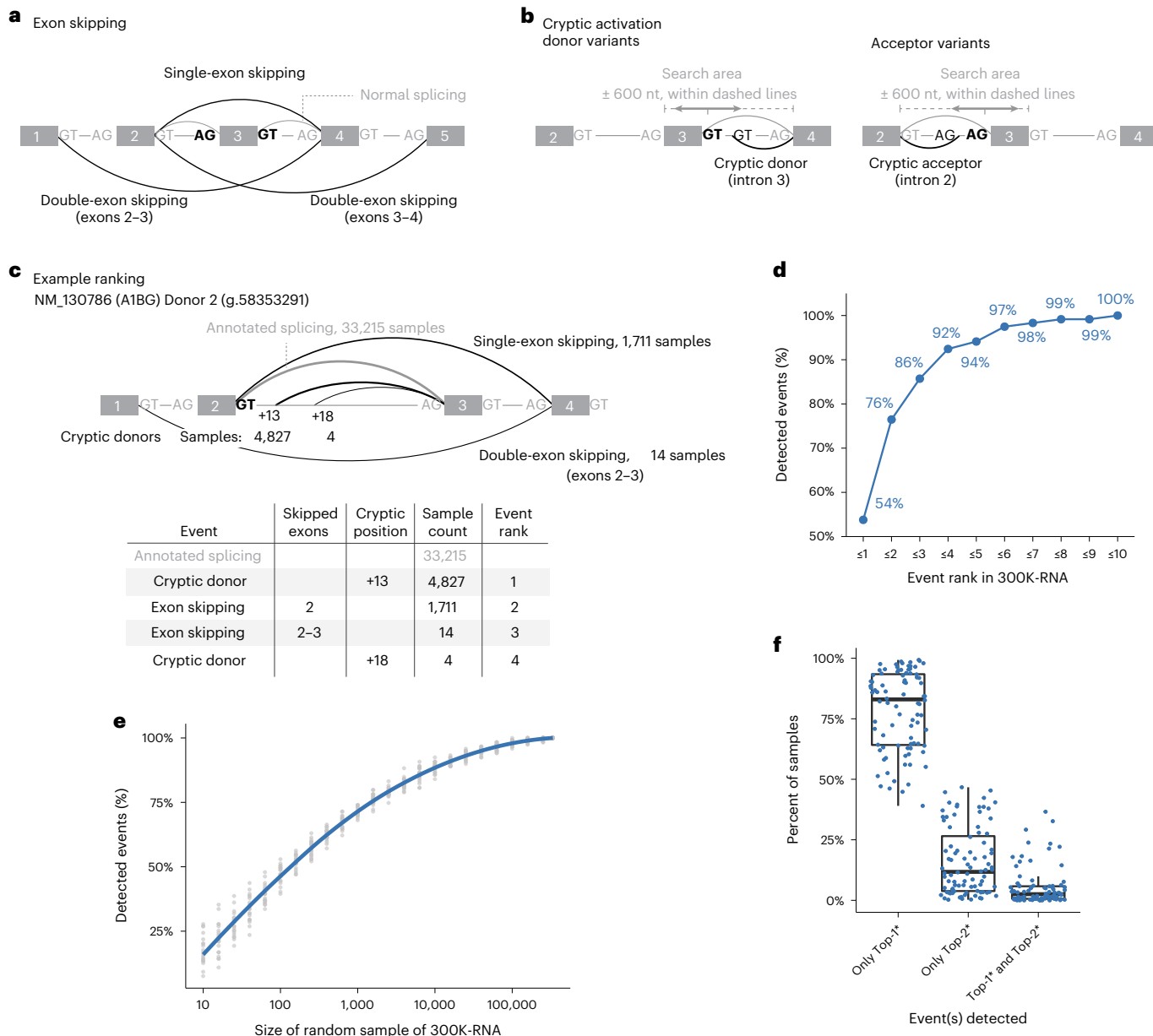

**Fig. 2 | Unannotated splicing events seen in 300K-RNA. a**, Exon-skipping events are evidenced by split-reads spanning nonconsecutive exons within the transcript. Splice sites (GT/AG motifs) shown in bold and black are those for which events are being ranked. **b**, Cryptic activation events are evidenced by split-reads spanning: (i) an annotated acceptor and an unannotated donor or (ii) an annotated donor and an unannotated acceptor. **c**, Example showing the Top-4* events for NM_130786 (A1BG) exon 2 donor (g.58353291). Exon/intron lengths are not drawn to scale. Arc thickness corresponds to event rank. **d**, One hundred percent (119/119) of exon-skipping and cryptic activation events detected across 88 variants are present in 300K-RNA, and 92% are in the Top-4* events for their respective splice site. **e**, Percent of the 119 true-positive events detected within

random subsets of the 335,663 source specimens in 300K-RNA. Gray dots show proportion across 20 random samples; blue line shows mean proportions with LOESS smoothing. **f**, Top-1* and Top-2* events around the splice sites affected by our 88 variants typically occur in mutually exclusive specimens—with both events seen, on average, in only 5% of total samples where either event was detected. Internal lines of boxplot denote the median value, and the lower and upper limits of the boxes represent 25th and 75th percentiles. Whiskers extend to the largest and smallest values at most 1.5IQR. An asterisk indicates our filter for events involving skipping one or two exons and cryptic activation within 600 nt of the annotated splice site.

exon (70/148 total events, 47%), followed by cryptic activation (44/148 events, 30%) and intron retention (29/148 events, 20%), and rarely caused multi-exon skipping (5/148 events, 3%) (Fig. 1c).

### Unannotated splicing events in 300K-RNA
The 300K-RNA database describes natural variation in splicing among 335,663 publicly available RNA-seq samples from GTEx[8] and SRA[15],

collected in the recount3 resource[16] (Methods). For each donor and acceptor in Ensembl[13] and RefSeq[14] transcripts, we collate all unannotated, stochastic splicing events surrounding that splice site (Fig. 2a–c), detected in RNA-seq samples processed in a unified pipeline in the recount3 resource[16]. Wilks et al. used splicing-aware alignment in an annotation-agnostic fashion, preventing bias against detection of unannotated events[16].

These splice junctions provide experimental evidence for an executed splicing reaction using the following: (1) a paired donor and acceptor from different introns, reflecting skipping of one or more consecutive exons normally present in that transcript (Fig. 2a; exon skipping); or (2) an annotated donor or acceptor, paired with an unannotated acceptor or donor, respectively, indicating cryptic splicing (Fig. 2b; cryptic splicing). Mis-splicing events detected at each splice site in 300K-RNA are ranked by the number of samples in which at least one splice junction read was detected. Figure 2c shows the Top-4 ranked events in 300K-RNA for *AIBG* exon 2 donor: at least one read representing annotated splicing was detected in 33,215 samples across 300K-RNA. A cryptic donor 13 nt downstream in the intron was detected in 4,827 samples, representing the Top-1 natural mis-splicing event at this site, and so on for the Top-2 to Top-4 events. The highest sensitivity and positive predictive value (PPV) for 300K-RNA predictions were obtained by applying a filter of a maximum of two exons skipped and cryptic splice sites within 600 nt (Extended Data Fig. 1; filter denoted hereafter by an asterisk). Using this filter, 300K-RNA Top-10* events identified all 119 exon-skipping and cryptic-splicing events induced by the 88 variants (Fig. 2d) with 64 of 119 (54%) the Top-ranked* event for that splice site (Fig. 2d and Extended Data Fig. 2).

Figure 2e shows the importance of sample size in 300K-RNA for the detection of all 119 true-positive exon-skipping and cryptic-splicing events. Taking random subsets within the 335,663 source specimens shows sensitivity only begins to maximize with ~100,000 samples. Deeper scrutiny of the 119 true-positive events shows, on average, each event is detected as a single splice junction read in 78% of samples that contain it—underpinning why all single read events are cataloged in 300K-RNA. Figure 2f reinforces the stochastic nature of these mis-splicing events, showing the Top-1 and Top-2 events around the splice sites affected by our 88 variants typically occur in mutually exclusive specimens—with both events seen, on average, in only 5% of samples where either event was seen.

## Concordance of 300K-RNA events between tissues and datasets

The ranking of the most common, natural mis-splicing events detected around each splice site in 300K-RNA is highly concordant between each tissue within GTEx and between GTEx and SRA (Fig. 3a,b (our 88 variants) and Extended Data Fig. 3a,b (98,810 annotated splice sites in clinically relevant Mendelian disease genes)). This tells us that the spliceosome reproducibly makes the same mistakes across a diverse repertoire of tissues and cell lines. Additionally, the number of samples in which 300K-RNA events are detected is strongly correlated between different GTEx tissues (Fig. 3c) and between GTEx and SRA (Extended Data Fig. 3c; $R = 0.91$ for our 88 variants, $R = 0.84$ for all splice sites in clinically relevant Mendelian disease genes). However, only 20% of 300K-RNA Top-4* events are present in all four clinically accessible tissues in GTEx (Fig. 3d; blood, fibroblasts, Epstein−Barr virus-transformed lymphoblast cell line (EBV-LCL) and muscle)—reflecting low expression of many Mendelian genes in these tissues. Sample size in GTEx (that is, the number of specimens available for that tissue subtype) increases the number of unannotated splice junctions detected ($R = 0.91$, $P < 2.2 \times 10^{-16}$; Extended Data Fig. 3d).

Notably, GTEx Muscle Top-4* did not provide improved sensitivity over 300K-RNA Top-4* for 19/88 of our variants associated with muscle disorders subject to RNA diagnostics on muscle samples (Extended Data Fig. 3e). Therefore, we recommend the use of 300K-RNA Top-4* as prediction of the probable nature of variant-associated mis-splicing until larger sample sizes and/or sequencing depth of specific tissues or cell types allows more thorough evaluation of a tissue-specific approach. We advise caution for genes with known tissue-specific or developmental alternative splicing where RNA-seq from the relevant tissue is not represented, or poorly represented, within the 300K-RNA data sources.

## Assessment of alternate RNA-seq sources

Long-read RNA-seq may prospectively assist the evaluation of new methods to improve the fidelity of short read alignment and reduce splice-junction artifacts in 300K-RNA, although it has its own bioinformatic and technical challenges. Extended Data Fig. 4 shows that long-read RNA-seq (7 M mean read depth and 740 nt average length) of 22 fibroblast specimens in GTEx v9 (ref. [24]) identifies only 15% (180 K) of the 1.16 M total splice junctions detected in 504 GTEx v8 fibroblast specimens (75 bp paired reads, poly A enriched and 80 M read depth). For comparison, in-house RNA-seq data from seven fibroblast specimens (150 bp paired reads, rRNA depleted total RNA and 100–200 M read depth) subject to cycloheximide (CHX) treatment identify 36% (420 K/1.16 M), substantially more than DMSO treated specimens (340 K/1.16 M) or in two randomized sets of seven fibroblast specimens from GTEx v8 (245 K/1.16 M). This preliminary evidence indicates CHX treatment and high read depth of RNA from human cell lines may substantially enhance our detection of unannotated mis-splicing events through inhibition of nonsense-mediated decay (NMD).

## SpliceAI predictions using custom interpretive rules

SpliceAI[19] is a deep-learning algorithm that predicts the location of splice sites, using up to 10,000 nt of mRNA sequence context for each prediction. SpliceAI is trained on the pre-mRNA sequences surrounding 130,796 annotated donor−acceptor pairs[19]. By default, SpliceAI scans ±50 nt of a variant entered and reports four delta scores (Δ-score), the maximum difference in score with and without the variant for acceptor loss, acceptor gain, donor loss and donor gain. The authors recommend a threshold of $\Delta \geq 0.20$ for high sensitivity of splicing alterations, $\Delta \geq 0.50$ for more general purposes and $\Delta \geq 0.80$ for high specificity[19].

Two insertion−deletion variants could not be assessed by SpliceAI (Supplementary Table 1; cases 50 and 66). For the remaining 86 variants, the SpliceAI[19] high sensitivity threshold of $\Delta \geq 0.20$ correctly predicted mis-splicing for 76/86 (88%) variants and for 63/86 (73%) variants with the $\Delta \geq 0.50$ threshold. Preliminary investigations using the default SpliceAI window of ±50 nt and high sensitivity $\Delta \geq 0.20$ threshold identified 13/44 variant-activated cryptic splice sites, noting only 19/44 activated cryptics lie within 50 nt, and 3/73 exon-skipping events (inferred by donor or acceptor loss of $\Delta \geq 0.20$ for both annotated splice sites flanking an exon), while the remaining 70 skipped exons were longer than the 50 nt scanned window. Therefore, we reasoned SpliceAI output could offer improved predictions of mis-splicing outcomes by extending the default scanned window to the maximum length of ±5,000 nt of the variant, developing custom interpretive rules to include predictions of multi-exon skipping, and optimizing the Δ-score threshold for this particular purpose (Fig. 4a−c).

Delta scores < 0.001 were excluded as nominal predictions of neutral impact, and all scores above this threshold were retained for subsequent precision-recall analysis (Methods). The remaining 2,836 Δ-scores returned for the 86 variants were interpreted according to the following rules: Δ-loss scores at the annotated splice site constituted a prediction of mis-splicing. Exon skipping is inferred if both splice sites flanking an exon have a Δ-loss score above threshold (Fig. 4a). Double-exon skipping is inferred if the relevant splice site of the upstream or downstream exon also has a Δ-loss score above threshold (Fig. 4a). Intron retention is inferred if both splice sites flanking an intron have a Δ-loss score above threshold (Fig. 4b). Cryptic activation is predicted by Δ-gain score above threshold for any unannotated donor or acceptor within the bounds of the exon and intron flanking the variant splice site (Fig. 4c). Delta scores that do not fall into these categories are annotated as 'other' (Methods).

According to these custom rules, SpliceAI predicts at least one mis-splicing event for all 86 variants and up to 31 predictions for a single variant (Extended Data Fig. 5). Of 145 mis-splicing events elicited by the 86 variants, 139 lie within the maximum ±5,000 nt

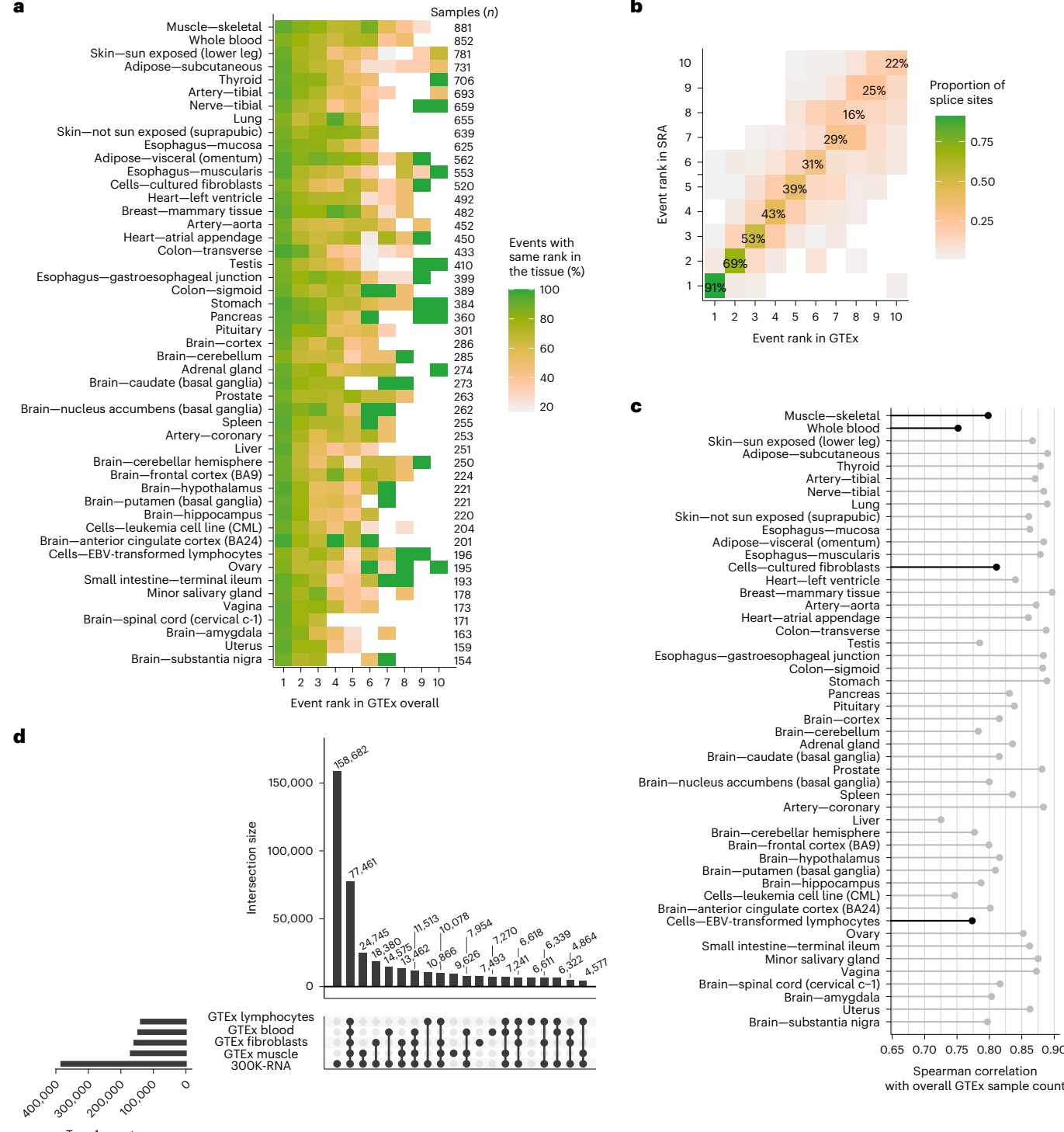

**Fig. 3 | 300K-RNA event rankings across tissues and data sources. a**, Heatmap showing the proportion of mis-splicing events* with the same event rank in each GTEx tissue subtype, as compared to all GTEx tissue subtypes combined—for the 88 splice sites affected by our cohort of variants. Only tissues with ≥100 GTEx samples are shown. The Top-1* event in individual tissues is concordant with the Top-1* event in 'all GTEx tissues' for ≥80% of splice sites. **b**, Concordance of top-ranked mis-splicing events* in GTEx versus SRA. The Top-1* event in GTEx is the Top-1* event in SRA for 80/88 (91%) splice sites in our cohort. **c**, Spearman correlation of all mis-splicing events* across 98,810 annotated splice sites in clinically relevant Mendelian disease genes (Methods) in each GTEx tissue

subtype versus GTEx overall. Only tissues with ≥100 GTEx samples are shown. Black, clinically accessible tissues. **d**, Upset plot[26] showing 300K-RNA Top-4* across clinically relevant Mendelian disease genes (four events per splice site, *n* = 383,677 in total) versus Top-4* specific to four clinically accessible tissues in GTEx. Twenty percent (77,461/383,677) of all 300K-RNA Top-4* across clinically relevant Mendelian disease genes are captured as Top-4* events among all four clinically accessible tissues (blood, fibroblasts, EBV-LCL and muscle). An asterisk indicates filtering to skipping one or two exons and cryptic activation within 600 nt of the annotated splice site.

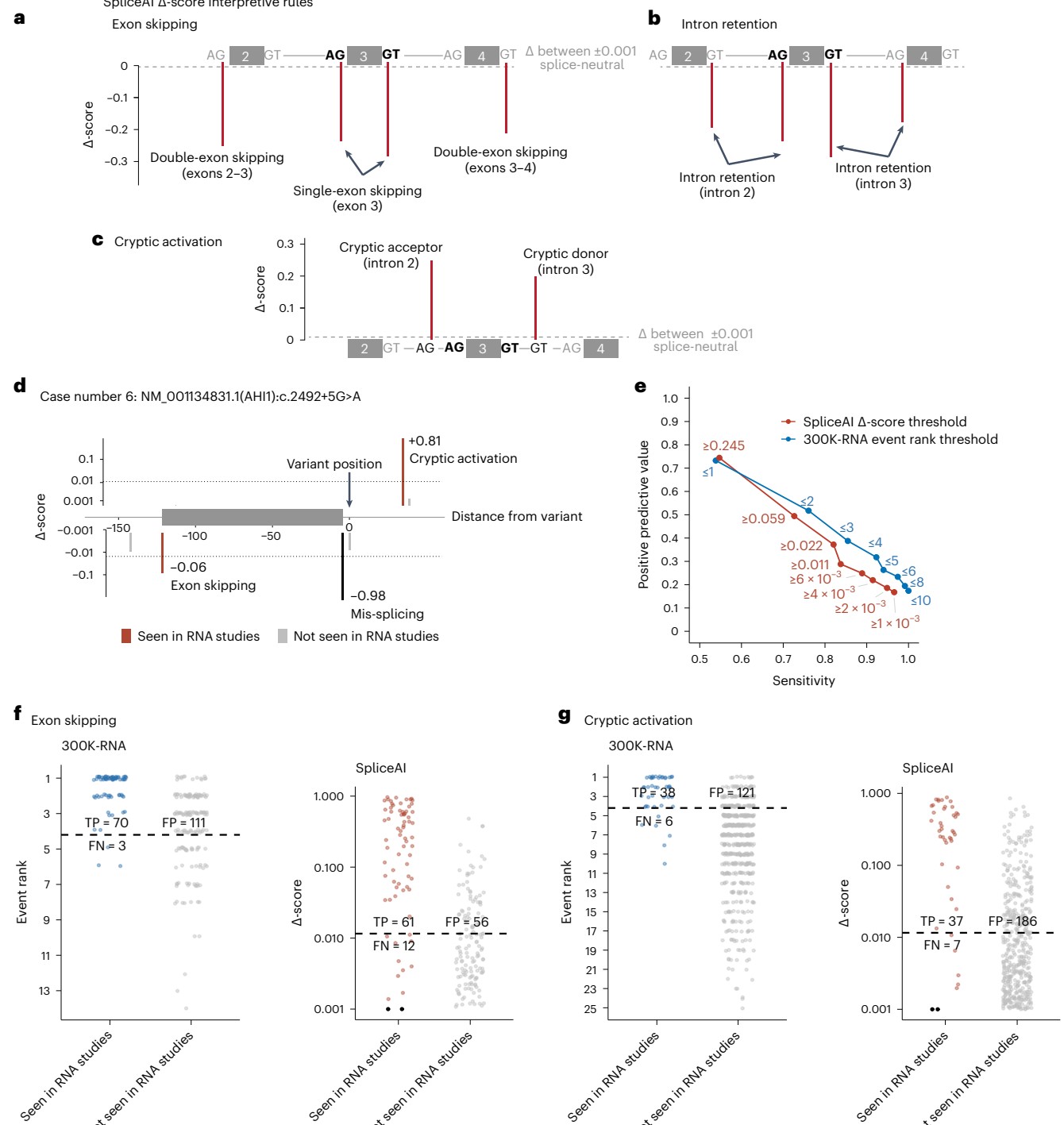

**Fig. 4 | Comparison of 300K-RNA Top-4\* with SpliceAI. a–c**, Custom interpretive rules applied to SpliceAI Δ-scores to predict the nature of mis-splicing. Δ-scores below 0.001 were excluded as our applied threshold for no predicted impact on splicing (threshold shown with gray dashed lines). Heights of red lines denote example Δ-scores that predict mis-splicing events according to our rules. **a**, Single-exon skipping is predicted if both splice sites flanking the exon have donor and acceptor loss Δ-scores above threshold, and double-exon skipping was inferred if the splice site of the upstream or downstream intron also had donor loss or acceptor loss Δ-score above threshold. **b**, Intron retention was predicted if both splice sites flanking an intron had donor loss and acceptor loss Δ-score above threshold. **c**, Cryptic activation was predicted by donor gain or acceptor gain Δ-scores above threshold for any unannotated donor or acceptor. **d**, Example showing SpliceAI predictions of exon skipping and cryptic activation in case number 6. **e**, Sensitivity and PPV of 300K-RNA and SpliceAI for exon-

skipping and cryptic-activation prediction at different thresholds, for the 86/88 variants that can be scored by SpliceAI. Points on the 300K-RNA curve (blue) show metrics when using Top-1\*, Top-2\*, Top-3\*, Top-4\*, etc. events as predictions of the nature of mis-splicing. Points on the SpliceAI curve (red) show metrics at Δ-scores that predict the same number of exon skipping and cryptic activation as 300K-RNA Top-1\*, Top-2\*, Top-3\*, Top-4\* and so on. **f**, 300K-RNA and SpliceAI predictions of exon skipping (seen/not seen in RNA studies across 86 cases). **g**, 300K-RNA and SpliceAI predictions of cryptic splice-site activation (seen/not seen in RNA studies across 86 cases). Dashed lines indicate the threshold of Top-4\* and SpliceAI Δ-score ≥ 0.011 identified in **e**. Black dots, mis-splicing events seen in RNA studies but not meeting the Δ-score threshold of 0.001. An asterisk indicates that filtering to skipping one or two exons and cryptic activation within 600 nt of the annotated splice site. TP, true positives, FN, false negatives, FP, false positives.

SpliceAI window (44/44 cryptics, 68/68 single-exon-skipping events, 4/5 double-exon-skipping events and 7/12 intron retention events). Figure 4d shows Δ-scores for case number 6, with SpliceAI predicting cryptic activation and exon skipping, both of which were seen in RNA studies. More distal Δ-scores above threshold not shown in Fig. 4d predict three multi-exon-skipping events and intron retention, which were not seen in RNA studies.

### Comparing predictive performance of 300K-RNA with SpliceAI

Precision-recall curves show the sensitivity and PPV of 300K-RNA and SpliceAI to predict the 119 exon-skipping and cryptic-activation events induced by 86/88 variants assessable by both methods (Fig. 4e). Top-4* showed higher sensitivity (92%) than SpliceAI (84%) at a Δ ≥ 0.011 threshold—this low threshold was selected because it identifies the same total number of mis-splicing events as Top-4* (Fig. 4e) to compare true/false positive and true/false negative rates.

Top-4* correctly identifies 96% (70/73) of detected exon-skipping events while SpliceAI Δ ≥ 0.011 predicts 85% (61/73) (Fig. 4f). Eighty-six percent (38/44) of activated cryptics are in Top-4*, with SpliceAI Δ ≥ 0.011 predicting 84% (37/44) (Fig. 4g). Both methods show a low PPV as follows: 39% (Top-4*) and 52% (SpliceAI Δ ≥ 0.011) for exon skipping (Fig. 4f) and 24% (Top-4*) and 17% (SpliceAI Δ ≥ 0.011) for cryptic activation (Fig. 4g). For intron retention events, which cannot be predicted using 300K-RNA, SpliceAI shows a sensitivity of 31% (9/29) and PPV of 36% (9/25) at the Δ ≥ 0.011 threshold.

Sensitivity (S) and PPV of Top-4* and SpliceAI (using Δ ≥ 0.011 threshold) were similar across two additional sets of variants curated from literature; 58 variants tested in patient specimens (Top-4* S = 91%, PPV = 26%; SpliceAI S = 94%, PPV = 33%) and 63 variants tested using midi-gene assays (Top-4* S = 92%, PPV = 29%; SpliceAI S = 86%, PPV = 29%) (Extended Data Fig. 6 and Supplementary Tables 2 and 3).

Analysis of features of Top-4* events reveals that relative to false positives, true positives tend to be identified in more samples (Extended Data Fig. 7a), represented by more unannotated splicing reads (Extended Data Fig. 7b,c), with a higher maximum ratio of the unannotated event relative to read depth for annotated splicing in any one sample (Extended Data Fig. 7e). However, there was no significant difference between true and false positives in the mean reads of annotated splicing in samples where events were detected (Extended Data Fig. 7d) or in the mean ratio between unannotated and annotated splicing (Extended Data Fig. 7f). Double-exon skipping is rarely activated by splice-site variants (Extended Data Fig. 7g). Four of five detected double-exon-skipping events were Top-1 (among 14 cases with double skipping ranked Top-1) and one of five were Top-2. In addition, true-positive and false positive Top-4* events show no difference in the length of the spliced-out region for exon-skipping events (Extended Data Fig. 7h) or cryptic distance (Extended Data Fig. 7i). We identified one false positive that may be due to alignment issues in the short read splice junction dataset: the first few nucleotides of two sequential exons are identical and split reads with only a few postjunction nucleotides can map to exon skipping or normal splicing.

We also emphasize that while a SpliceAI threshold of 0.011 was effective for this bespoke application to forecast the likely nature of any variant-elicited mis-splicing, we do not recommend the use of a 0.011 delta score threshold as a prediction for mis-splicing generally, as our evidence from experimentally confirmed splice neutral variants (ref. [7] and cases studied since) indicates this will yield a high false-positive rate of >50%.

### RNA re-analysis uncovers previously undetected Top-4* events

We noted many Top-4* events not detected in our early RNA diagnostics cases (before 40K-RNA or 300K-RNA) involved double-exon-skipping events (42%) or cryptic activation events further than 250 nt from the annotated splice site (11%), which may have been missed on initial analysis. Before the development of 40K-RNA[6], our laboratory practice included critical review of all cryptic splice sites within 250 nt of the annotated donor[7]. Scrutiny of 24/88 variants where the Top-1 event was not detected by RNA studies (Extended Data Fig. 2) showed that 16/24 were detectable via the RNA diagnostic strategy deployed, but not observed, due to the following: (1) the event was not activated by the variant; (2) low expression of the target gene, potentially limiting sensitivity and (3) the variant simultaneously weakened an annotated splice site and spatially overlapping cryptic splice site comprising a Top-1* event. Six of 24 events were observed, although available RNA assay data did not confidently establish elevated levels relative to controls. One of 24 events was detected upon review of Sanger sequencing trace file. We performed RNA re-analysis for the one case with an undetected Top-1* event and three additional cases where multiple Top-4* events were not detected (A024-*OPHN1* c.702+4A>G; A060-*GSDME* c.1183+5G>A; A014-*SPG11* c.2317-13C>G; A205-*EMD* c.266-3A>G)[7]. We identified or clarified variant-associated enhanced use of 1/4 multi-exon-skipping events (Extended Data Fig. 8a,b; *SPG11*, red), 4/4 cryptic splice sites (Extended Data Fig. 8c,d; *GSDME* and *EMD*, red) and one single-exon-skipping event (Extended Data Fig. 8d, *EMD*, red). Skipping of multiple exons associated with *SPG11* c.2317-13C>G was not detected initially by RT-PCR due to primer placement in exons too proximal to the splice variant and undetected by RNA-seq due to low read depth exacerbated by NMD. Activation of two cryptic donors and two cryptic acceptors associated with *GSDME* c.1183+5G>A and *EMD* c.266-3A>G, respectively, was missed initially due to competition inherent with multitemplate PCRs, heteroduplex formation and challenges resolving multitrace chromatograms by Sanger sequencing.

### Half of essential GT-AG variants induce ≥1 in-frame event

Forty-nine percent (27/55) of ES variants across the three variant sets induced at least one in-frame event, with a similar proportion of 51% (45/88) for all variants in our cohort (Extended Data Fig. 8e). When considering Top-4* and intron retention as a prediction of variant-induced mis-splicing, the number of ES variants with ≥1 in-frame event is 80% (45/55) and 81% (71/88) for all variants in our cohort.

## Discussion

Clinical interpretation of splicing variants relies on predicting, or experimentally verifying, the nature of variant-induced mis-splicing to confirm the variant impact on the encoded protein. This is of particular importance when applying the PVS1 (null variant) criterion to ES variants[5]. While the impact of exon skipping and intron retention on protein reading frame can be theorized, it has remained difficult to predict whether exon-skipping or cryptic splice-site activation will occur—and if a cryptic splice site is activated, which one of the many potential sites present in the vicinity will be selected by the spliceosome.

Our empirical method of using 300K-RNA Top-4* accurately predicts the nature of variant-associated mis-splicing with 92% sensitivity for 88 variants across a broad range of genes and disorders, outperforming SpliceAI on average to correctly predict exon-skipping, double-exon skipping and cryptic splice site activation. We emphasize that Top-4* cannot be used for variants creating or modifying the ES motif of a cryptic splice site and recommend use of SpliceAI for this category of variant[6]. In addition, although intron retention cannot yet be quantified and ranked by 300K-RNA, Extended Data Fig. 9 shows that intron retention induces a frameshift or encodes a PTC in all three frames for at least 97% introns in clinically relevant Mendelian disease genes and is therefore consistent with null outcomes in most instances.

It is important to acknowledge the low PPV of Top-4* when used as a prediction of the nature of mis-splicing. However, we feel that prioritizing sensitivity is of the greatest importance, to avoid false-negative predictions. RNA re-analysis of four cases with one or more undetected Top-4* events via our initial RNA diagnostic testing revealed we had missed 6/9 of these events, due to experimental design and/or

technical limitations. A priori knowledge of Top-4* mis-splicing events has been transformative for our research-led clinical RNA diagnostics program, facilitating both variant curation and strategic experimental design of RNA assays to specifically target probable mis-splicing events, expressly important for RT-PCR where primer design and extension times strongly influence which products may be amplified.

Our re-interrogation of early cases, showing we had missed several rare events, raises the possibility that Top-4* PPV could be higher than we currently estimate. It also reinforces clinical benefits of being able to reliably predict probable mis-splicing events to improve the completeness and accuracy of conclusions drawn from RNA diagnostics. Notably, we cross-checked all other early cases (before 300K-RNA) to confirm that interpretation of likely pathogenicity would not be impacted by any undetected Top-4* events that may have resulted (it was neither feasible nor economic to retest all specimens).

Top-ranked unannotated splicing events are highly concordant between tissues and between GTEx and SRA (Fig. 3a,b and Extended Data Fig. 3a,b), in line with previous analyses showing that expression levels rather than alternative splicing patterns underpin most tissue-specific variation in GTEx[25]. We suspect the predictive accuracy of our ranking method may be improved further by the following: (1) higher read depth RNA-seq data across the breadth of manifesting tissues in rare disorders (for example, no data are currently available for cochlear), (2) incorporation of RNA-seq data from human fetal samples to catalog developmental alternative splicing, (3) use of CHX to inhibit nonsense-mediated decay, (4) a better understanding of contexts that influence variant-associated double-exon skipping, (5) an ability to rank likelihood of variant-activated intron retention and (6) improved bioinformatic methods in sequencing read alignment. Consideration of Top-3* events in specific manifesting tissues, if shown to maintain >90% sensitivity, could substantially improve PPV and clinical utility for variant classification.

To our knowledge, 300K-RNA Top-4* is the first evidence-based method for predicting the nature of variant-associated mis-splicing and will assist clinical classification of splice-site variants and/or guide RNA diagnostic testing strategies to make sure no likely mis-splicing outcome is missed due to design flaws or technical limitations. Informed by the current investigation, Extended Data Fig. 10 details our draft guidelines for the possible use of 300K-RNA to assist application of the PVS1 'null variant' criterion to ES variants, aligning closely with revised PVS1 guidelines[5]. Theorized consideration of intron retention and Top-4* events by genetic pathology workforces provides a pragmatic, evidence-based method to reliably assess variant-associated exon skipping and/or cryptic splice site use within a larger distance window of 600 nt. Over 2022–2023, the Australasian Consortium for RNA diagnostics (SpliceACORD)[7] will rigorously evaluate the clinical accuracy and usefulness of Extended Data Fig. 10 draft guidelines for cases prospectively recruited into RNA diagnostic testing pipelines. Extended Data Fig. 8e indicates these draft guidelines will allow the application of PVS1 for ~50% of ES variants (IR and Top-4* ≤ 2 in-frame events).

We provide SpliceVault, a web portal to access 300K-RNA (and 40K-RNA in hg19), which quantifies natural variation in splicing and potently predicts the nature of variant-associated mis-splicing (https://kidsneuro.shinyapps.io/splicevault/). Users require no bioinformatics expertise and can retrieve stochastic mis-splicing events for any splice junction annotated in Ensembl or RefSeq. Default settings display 300K-RNA Top-4* output according to the optimized parameters we describe herein, with the option to return all events, customize the number of events returned, distance scanned for cryptic splice sites, maximum number of exons skipped or list tissue-specific mis-splicing events. We hope SpliceVault will improve the ability to classify and study splicing variants with accuracy and completeness, avoiding the non-actionable diagnostic endpoint of a variant of uncertain significance.

## Online content

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

# Methods

## Ethics declaration
Consent for diagnostic genomic testing was supported by governance infrastructure of the relevant local ethics committees of the participating Australian Public Health Local Area Health Districts. Kids Neuroscience Centre's biobanking and functional genomics human ethics protocol was approved by the Sydney Children's Hospitals Network Human Research Ethics Committee (protocol 10/CHW/45 renewed with protocol 2019/ETH11736 (July 2019–2024)) with informed, written consent for all participants.

## Creating 300K-RNA
The 300K-RNA database consists of splice junctions from 335,663 publicly available RNA-seq samples from GTEx[8] and SRA[15] aligned using a unified Monorail pipeline from ref. [16]. Splice junction read counts derived from 316,449 human RNA-seq samples from SRA and 19,214 human RNA-seq samples from GTEx were downloaded from the public resource recount3 (ref. [16]). We then filtered split reads to those that span at least one annotated splice site, and for each splice junction detected we tallied the number of samples it occurred in across the two data sources. For each splice junction, we filtered associated detected exon-skipping and cryptic-activation events according to the rules in Fig. 2a,b.

Unannotated splicing events were ranked according to the number of samples in which the event was detected. This ranking process was completed with respect to each annotated splice junction in Ensembl transcripts (v104) and Refseq transcripts (GRCh38, downloaded August 2021).

The 300K-RNA enables customized access to ranked splice-junction data from individual GTEx tissue subtypes. SRA metadata precludes breakdown into specimen subtypes. As SRA contains data from cancer specimens (genetically heterogeneous), maximum read-counts output for each splice junction is derived from GTEx data. The R package Snapcount (v1.8.0)[27] was used to retrieve information on individual samples for Fig. 2e,f and Extended Data Fig. 7.

## SpliceAI Δ-score interpretive rules
To adapt SpliceAI to the prediction of mis-splicing, we retrieved all Δ-scores ±5,000 nt of each variant, adapting a script from the SpliceAI GitHub (https://github.com/Illumina/SpliceAI, '3. Can SpliceAI be used to score custom sequences?') into an API to allow us to easily retrieve scores (source code provided here https://gitlab.com/kidsneuro/SpliceAIAPI; ref. [28]). As input, we used the pre-mRNA sequence ±5,000 nt of the variant. Two Δ-scores returned at each base (variant nucleotide versus reference nucleotide) generated up to 20,002 Δ-scores per variant, of which we excluded all Δ-scores ≤ 0.001 as neutral impact.

Across the 86 variants which could be scored by SpliceAI, 2,836 Δ-scores returned were above the 0.001 threshold. Eighty-six of these were donor loss or acceptor loss Δ-scores of the affected annotated splice site, denoting a prediction of mis-splicing. Our custom interpretive rules (see explanation of Fig. 4a–c in the text) applied to any Δ-score > 0.001 yielded predictions of erroneous use of 161 cryptic acceptors, 340 cryptic donors, 215 exon-skipping events and 49 intron retention events. Of the remaining 2,071 predictions, 1,637 were decreases in scores of unannotated splice sites, 33 were increases in scores of annotated splice sites and 315 were increases in the scores of unannotated splice sites outside the bounds of the exon and intron flanking the variant splice site or increases in the scores of unannotated donors for acceptor variants and vice versa—and deemed uninterpretable within our paradigm.

## Clinically relevant Mendelian disease genes
Clinically relevant genes were extracted from the Genomics England PanelApp[29] (September 2021 release) and Online Mendelian Inheritance in Man database (OMIM, https://omim.org/; February 2021 release). Genes were extracted from the Genomics England PanelApp using the Swagger PanelApp API (v1) (panelapp.genomicsengland.co.uk/api/docs/), excluding disease susceptibility panels; all genes were below a confidence level 3 (green; diagnostic grade). OMIM-listed genes were excluded when their only phenotype associations were nondiseases, susceptibilities, provisional links and somatic mutations.

## RNA re-analysis
Whole blood was collected in a PAXgene (PreAnalytiX) blood RNA tube and RNA was isolated according to kit instructions. Peripheral blood mononuclear cells (PBMCs) were isolated using SepMate-15 tubes (StemCell Technologies) and Ficoll Paque Plus (GE Healthcare). PBMCs were cultured in RPMI 1640 Medium (Gibco), 10% fetal bovine serum (GE Healthcare) and penicillin-streptomycin (100 U µg ml$^{-1}$; Gibco). Fibroblasts were cultured in high glucose DMEM (Gibco), 10% FBS (GE healthcare) and Gentamicin (50 µg ml$^{-1}$; Gibco). PBMCs and primary fibroblasts were treated with dimethyl sulfoxide (Sigma-Aldrich) or 100 µg ml$^{-1}$ cycloheximide (Sigma-Aldrich) for 6 h before harvesting in 600 µl RLT buffer (Qiagen) for RNA extraction using the Qiagen RNeasy mini kit. SuperScript IV first-strand synthesis system (Invitrogen) was used to make cDNA from 500 ng of RNA according to kit instructions. Recombinant Taq DNA polymerase (Invitrogen) and MasterAmp 2X PCR PreMix D (Epicentre Biotechnologies) were used for PCRs. Thermocycling conditions were 94 °C for 3 min, 35 cycles 94 °C 30 s, 58 °C 30 s, 72 °C 90 s per kilobase, then 72 °C for 10 min. All PCR products were analyzed on a 1.2% agarose gel. Amplicons were manually excised from an agarose gel with a scalpel and cDNA was purified using GeneJET gel extraction kit (Thermo Fischer Scientific) according to the manufacturer's instructions. Further, 8–75 ng of purified cDNA and 1 pmol of sequencing primer were subject to Sanger sequencing. Sanger sequencing chromatograms were analyzed using Sequencher DNA sequence analysis software (Gene Codes). See Supplementary Table 4 for primers used for re-analysis of RNA by RT-PCR.

## Statistics and reproducibility
No statistical method was used to predetermine the sample size. All splicing variants were included in this study for which robust RNA assay data were available and met the following ascertainment criteria. Variants affecting an annotated splice site and demonstrated to activate exon-skipping or cryptic splice-site use were included in this study. Variants creating or modifying a cryptic splice site were excluded. The experiments were not randomized. The investigators were not blinded to allocation during experiments and outcome assessment.

## Reporting summary
Further information on research design is available in the Nature Portfolio Reporting Summary linked to this article.

# Data availability
Source data for 300K-RNA were downloaded from snaptron (http://snaptron.cs.jhu.edu/data). 300K-RNA can be easily accessed and queried through SpliceVault (https://kidsneuro.shinyapps.io/splicevault/). The data used for the analyses described in this manuscript were obtained from the GTEx portal and dbGaP accession numbers phs000424.v8.p2 and phs000424.v9. Source data are provided with this paper.

# Code availability
All codes required to perform analyses and generate Figs. 1c, 2d–f, 3 and 4e–g and Extended Data Figs. 1–7 and 9 are available at https://github.com/kidsneuro-lab/SpliceVault_figures (ref. [30]). All codes required to create 300K-RNA are available at https://github.com/kidsneuro-lab/300K-RNA (ref. [31]). The code used to create SpliceVault is available at https://github.com/kidsneuro-lab/SpliceVault/ (ref. [32]).

The code used for SpliceAI Δ score retrieval API is available at https://gitlab.com/kidsneuro/SpliceAIAPI (ref. [28]).

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

## Acknowledgements

We thank the families for their participation and invaluable contributions to this research. We also thank the clinicians and healthcare workers involved in their assessment and management. The GTEx Project was supported by the Common Fund of the Office of the Director of the National Institutes of Health, and by NCI, NHGRI, NHLBI, NIDA, NIMH and NINDS. S.T.C. is supported by a National Health and Medical Research Council (NHMRC) of Australia Senior Research Fellowship APP1136197. This project received funding through NHMRC Ideas Grants APP1106084 and APP2002640 and a Medical Research Future Fund Rapid Applied Research Translation Program grant awarded to Sydney Health Partners. Part of this work was supported by Luminesce Alliance Innovation for Children's Health, a not-for-profit joint venture between the Sydney Children's Hospitals Network, the Children's Medical Research Institute and the Children's Cancer Institute, from Lenity Australia, a not-for-profit philanthropic organization, and from the Sydney Children's Hospitals Foundation, a registered charity. R.D. and A.M.B. are supported by a University of Sydney Research Training Scholarship. S.J.B. is supported by a Muscular Dystrophy Association of New South Wales Sue Connor postgraduate training scholarship.

## Author contributions

R.D. conceived the project. R.D., A.M.B., S.J.B., R.G.M. and H.J. curated datasets. R.D., R.G.M. and H.J. performed computational analysis. A.M.B., S.J.B. and S.B. performed experimental validation of splicing variants. R.D., A.M.B. and S.T.C. wrote the original draft manuscript with input and editing from all authors. R.D. and A.M.B. contributed equally. H.J. and S.T.C. jointly supervised this work.

## Competing interests

S.T.C. is director and shareholder of Frontier Genomics Pty (Australia). S.T.C. currently receives no consultancy fees or other remuneration for this role. H.J. offers Technology advice to Frontier Genomics Pty (Australia) and receives no remuneration for this role. All the other authors declare no conflicts of interest.

## Additional information

**Extended data** is available for this paper at https://doi.org/10.1038/s41588-022-01293-8.

**Correspondence and requests for materials** should be addressed to Sandra T. Cooper.

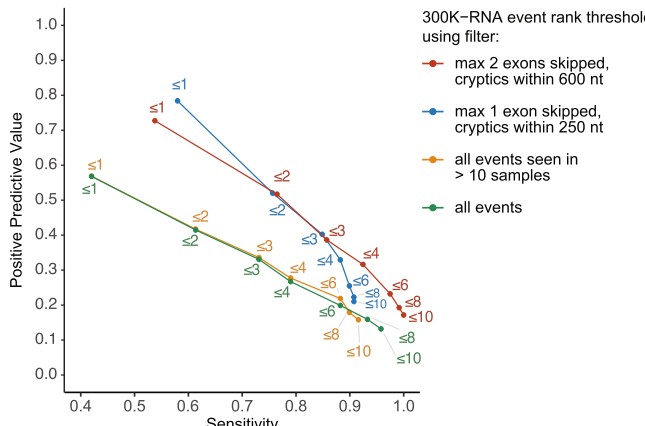

**Extended Data Fig. 1 | Sensitivity and PPV of 300K-RNA using different filtering criteria for ranking mis-splicing events.** Points on the red curve correspond to use of Top-1, 2, 3, 4 etc events as prediction of the nature of mis-splicing. The asterisk i.e 'Top-4*' is used denote application of a filter to limit ranked events to those involving skipping of one or two exons and cryptic activation within 600 nt of the annotated splice-site. Points on the blue, yellow and green curves similarly show metrics when using Top-1, 2, 3, 4 etc events as prediction of the nature of mis-splicing, when the three alternative filters listed in the legend are applied (max 1 exon skipped and cryptics within 250 nt, all events seen in >10 samples, or all events).

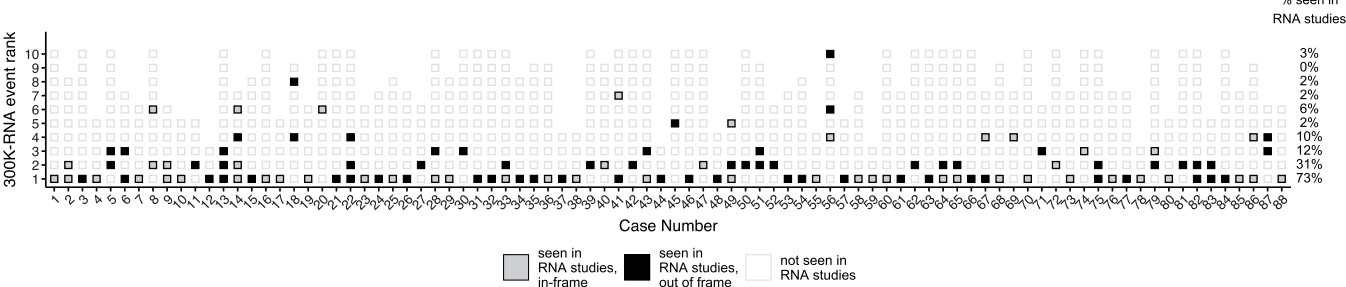

**Extended Data Fig. 2 | Top-10\* mis-splicing events seen in 300K-RNA\* for our cohort of 88 variants, filled if they were seen in RNA studies.** *Gray fill*: multiple of 3 (maintains frame). *Black fill*: not a multiple of 3 (disrupts frame). *No fill*: Event not seen in RNA studies. \* = skipping one or two exons and cryptic activation within 600 nt of the annotated splice-site.

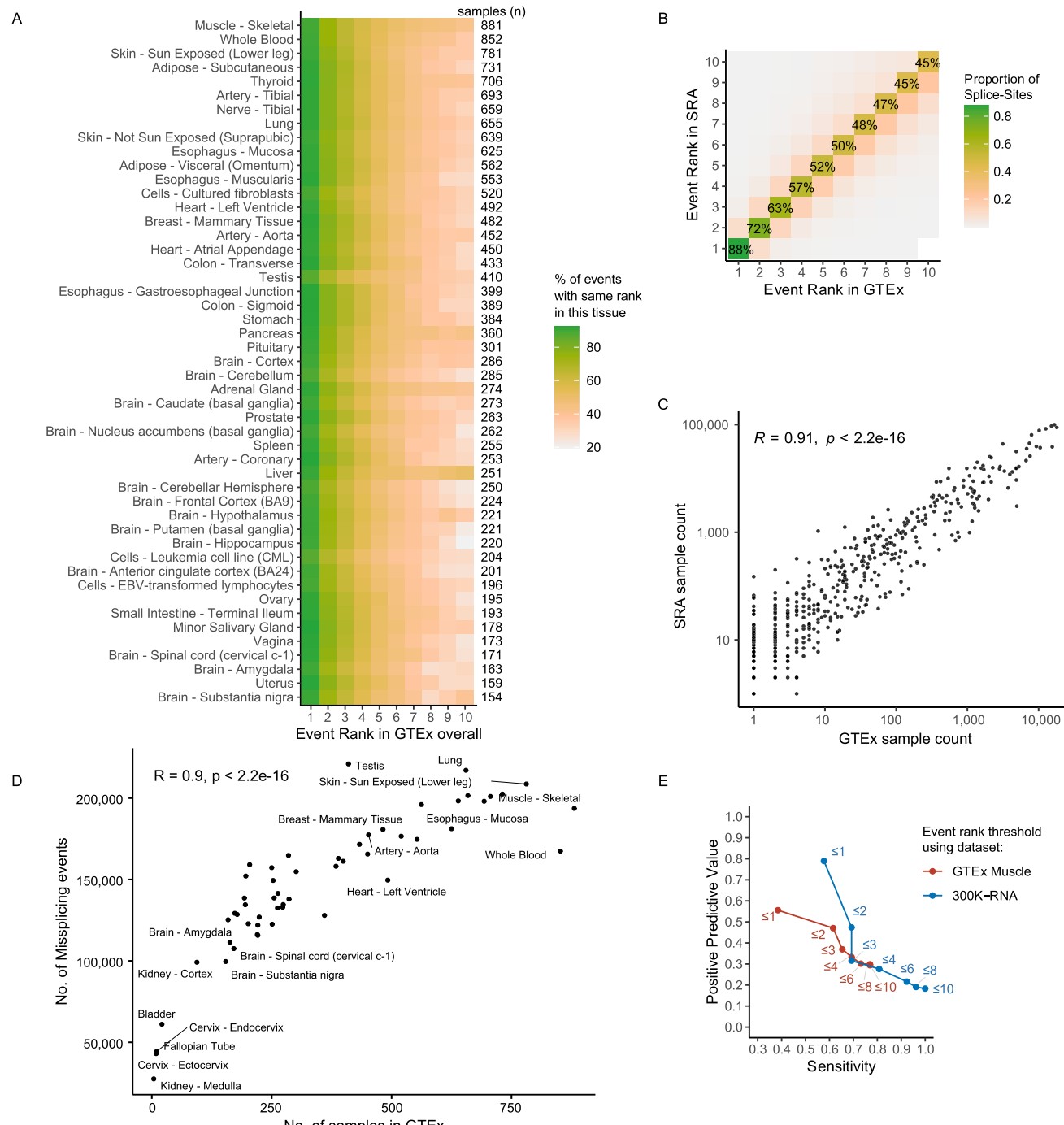

**Extended Data Fig. 3 | 300K-RNA event rankings across tissues and data-sources. a)** Heatmap showing the proportion of mis-splicing events* with the same event rank in each GTEx tissue subtype, as compared to all GTEx tissue subtypes combined, for 98,810 annotated splice-sites in clinically relevant Mendelian disease genes (see methods). The top-1* event is > 86% concordant across all tissues with > 100 samples in GTEx. **b)** Concordance of top-ranked mis-splicing events* in GTEx versus SRA. The top-1* event in GTEx is the top-1* event in SRA for 88% all splice-sites in clinically relevant Mendelian disease genes. **c)** Sample counts are highly correlated between GTEx and SRA for all unannotated splicing events* in 300K-RNA at the splice-sites affected by our cohort of 88 variants (Spearman correlation, R = 0.91, p < 2.2e−16). **d)** Sequencing breadth in GTEx samples increases sensitivity that is the more specimens the greater the proportion of total mis-splicing events detected, with testis being a notable outlier (Spearman correlation, R = 0.91, p < 2.2e−16). **e)** Sensitivity and PPV of Top-ranked* events seen in either 300K-RNA (blue) or GTEx Muscle (red, 881 samples in GTEx), for 19 variants associated with muscle disorders and where RNA testing was performed using muscle RNA. * = skipping one or two exons and cryptic activation within 600 nt of the annotated splice-site.

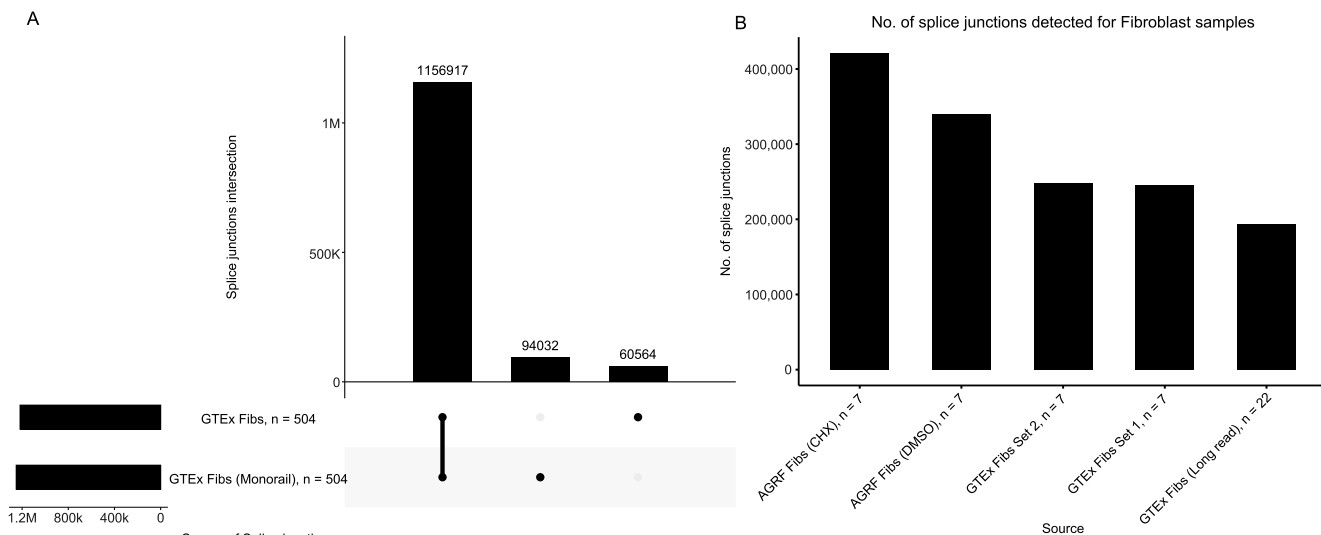

**Extended Data Fig. 4 | a) Upset plot showing splice-junctions concordantly and uniquely detected by GTEx and Monorail processing pipelines for 504 fibroblast specimens. b)** Upset plot showing GTEx V9 long-read RNA-Seq (7 M mean read depth, 740 nt average length) for 22 fibroblast specimens identifies 15% (180 K/1.16 M) of all fibroblast SJ detected in 504 GTEx v8 fibroblast samples (75 bp paired reads, 80 M depth, Poly A enrichment). In-house RNA-Seq data from 7 fibroblast specimens (150 bp paired reads, 100–200 M depth, rRNA depleted total RNA) subject to cycloheximide (CHX) treatment identifies 36% (420 K/1.16 M) of all SJ, substantially more that DMSO treated specimens (340 K/1.16 M) or in two randomized sets of 7 fibroblast specimens from GTEx v8 (245 K/1.16 M). Splice junctions present in GTEx v9 long read RNA-Seq data was reverse-engineered from the transcript count information generated using FLAIR[27]. AGRF, Australian Genome Research Facility; Fibs, fibroblasts.

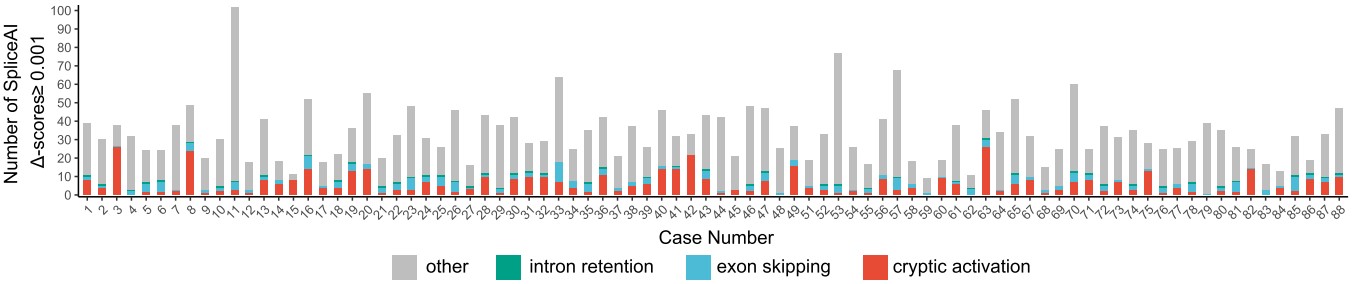

**Extended Data Fig. 5 | SpliceAI Δ-scores above 0.001 for 86/88 variants scored by SpliceAI.** Predictions are colored according to the event type assigned by our interpretive rules.

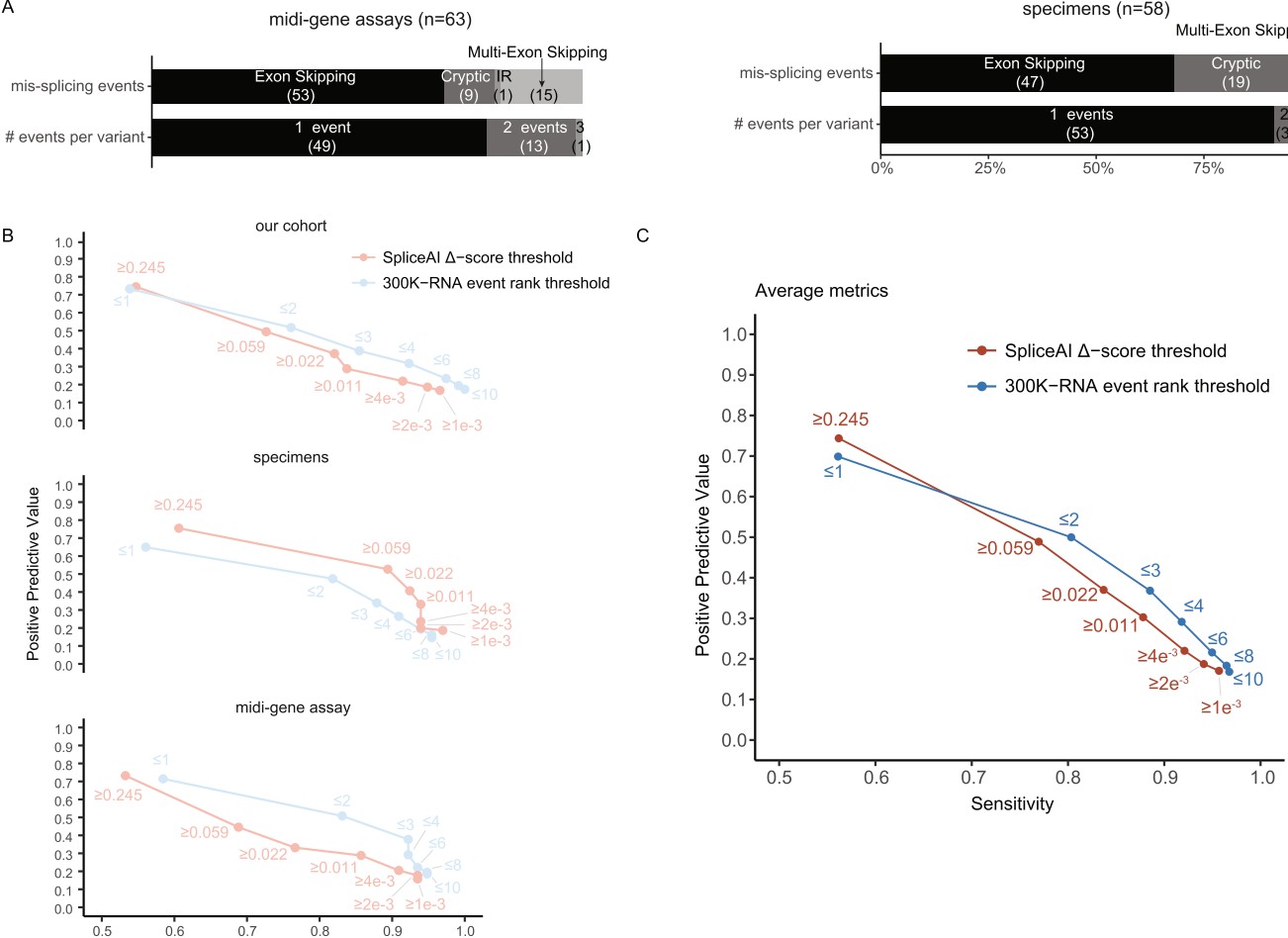

**Extended Data Fig. 6 | Comparison of 300K-RNA Top-4* with SpliceAI using two additional variant cohorts. a)** Mis-splicing events induced by variants in two cohorts curated from literature for additional validation of the Top-4* approach: 58 variants studied in patient specimens and 63 variants studied through midi-gene assays (scrutinized to ensure technical design permitted detection of multi-exon skipping events, see Supplementary Tables 2, 3 for references). **b)** Sensitivity and PPV of 300K-RNA and SpliceAI for exon-skipping and cryptic activation predictions at different thresholds. Points on the

300K-RNA curve (blue) show metrics when using Top-1*, 2*, 3*, 4* etc events as a prediction of the nature of mis-splicing. Points on the SpliceAI curve (red) show metrics at delta-scores that predict the same number of exon-skipping and cryptic-activation as 300K-RNA top-1*,2*,3*,4* etc to enable direct comparison of true/false positive rates with the Top-4 method. **c)** The average sensitivity and PPV of 300K-RNA and SpliceAI across three variant cohorts shown in **(B)**. Points on the curves correspond to thresholds used in panel B. * = skipping one or two exons and cryptic activation within 600 nt of the annotated splice-site.

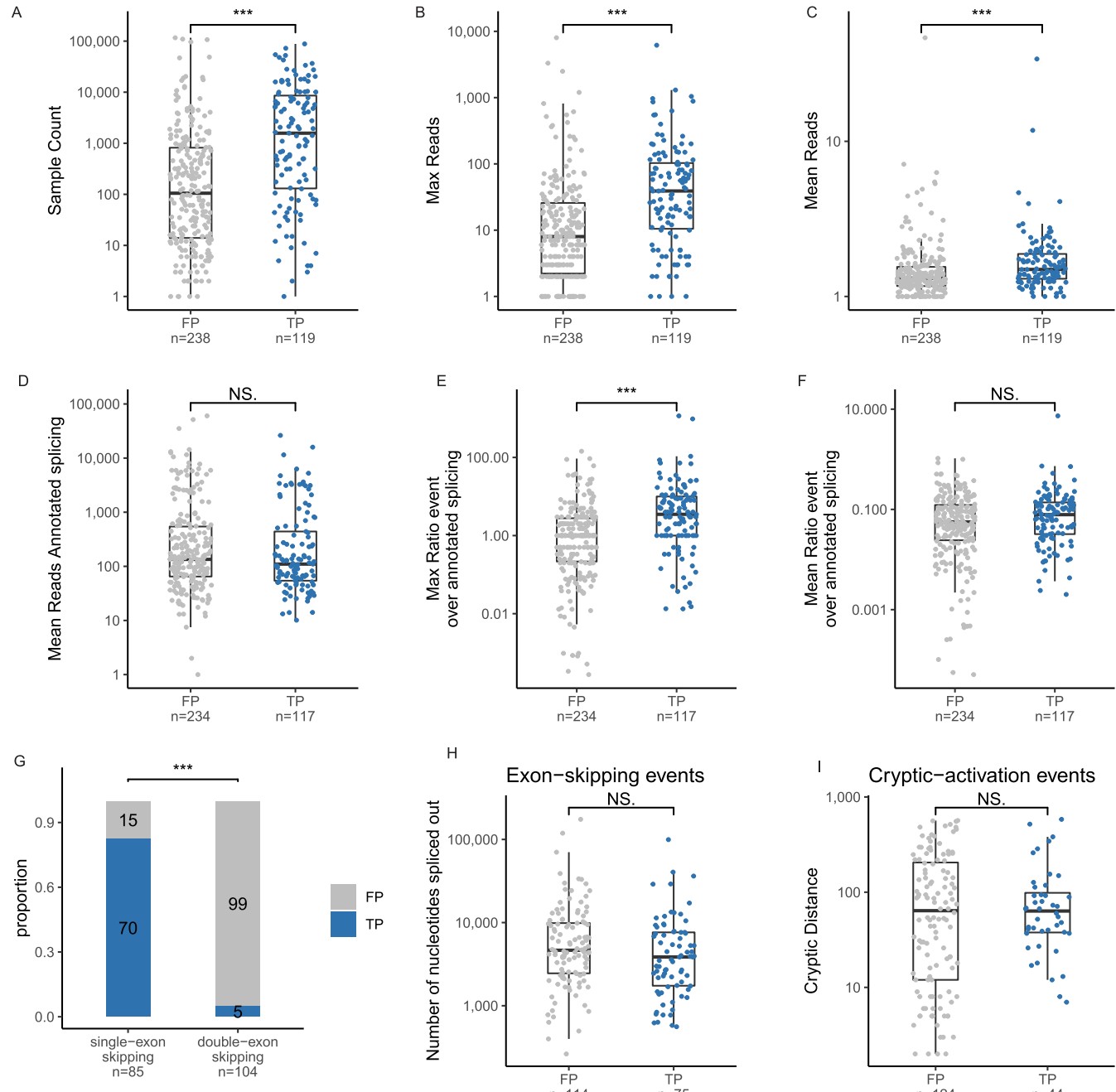

**Extended Data Fig. 7 | Investigating features of events seen/not seen in RNA studies for our 88 variants.** True positive (TP) events seen in RNA studies (n = 119), versus false positives (FP) in the Top-4 (n = 238) have higher: **a)** sample counts (W = 20,222, p = 4e⁻¹¹), **b)** max (W = 20,144, p = 7e⁻¹¹), and **c)** mean reads (W = 18,284, p = 7e⁻6, two-sided Wilcoxon rank sum test). Events are biologically independent in **A-C**. For the annotated splice-junction around which the unannotated event is detected, TP versus FP have **d)** no significant difference in the mean read-depth (W = 12,848, p = 0.3), **e)** higher maximum ratio (W = 19,126, p = 1e⁻⁹) but **f)** no significant difference in the mean ratio of unannotated to annotated reads (W = 15,181, p = 0.1, two-sided Wilcoxon rank sum test). n = 234 Top-4* FP and 117 TP biologically independent events for **(D-F)**: 4/238 FP and 2/119 TP were detected only in samples where no annotated splicing was detected, so are excluded from **d-f**. **g)** Single-exon skipping events* are significantly more

likely to be seen in RNA studies than double-exon skipping events* (Chi-squared test; χ² = 114.29, p = 1.1e⁻²⁶). **h)** Total length (nt) of the fragment excised from the pre-mRNA from single and double exon skipping was not statistically different between TP and FP (two-sided Wilcoxon rank sum test; W = 2,554, p = 0.0502, n = 114 Top-4* FP and 75 TP biologically independent exon-skipping events). **i)** Distance from annotated splice-site to activated cryptic splice-site was not statistically different between TP and FP (two-sided Wilcoxon rank sum test: W = 2,834, p = 0.70, n = 124 Top-4* FP and 44 TP biologically independent cryptic-activation events). A-E and H-I are box-whisker plots, with internal lines denoting the median value, and the lower and upper limits of the boxes representing 25th and 75th percentiles. Whiskers extend to the largest and smallest values at most 1.5IQR. * = skipping of one or two exons or cryptic activation within 600 nt of the annotated splice-site.

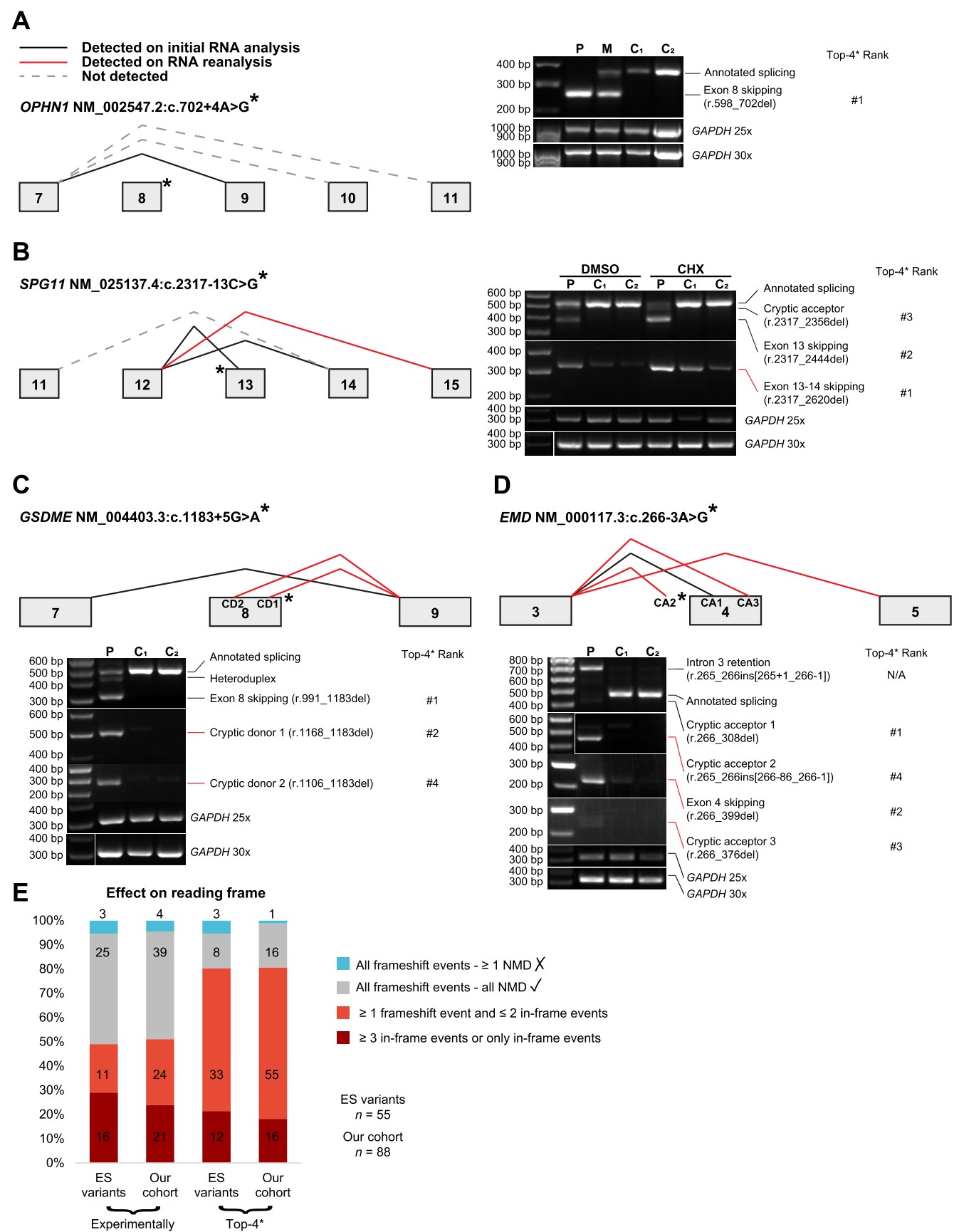

**Extended Data Fig. 8 | See next page for caption.**

**Extended Data Fig. 8 | RNA re-analysis to check for undetected 300K-RNA Top-4 mis-splicing events.** *Black lines:* mis-splicing identified during initial RNA analysis[7]. *Red lines*: Top-4* events detected upon re-analysis. *Gray lines*: Top-4* events undetected upon re-analysis. **a)** No additional Top-4 events were identified for *OPHN1* c.702 + 4 A > G. **b)** For *SPG11* c.2317-13 C > G, Top-1* exon 13 + 14 skipping was detected on re-analysis (missed initially by RT-PCR due to primer positioning and by RNA-seq due to low read depth and NMD), but Top-4* event, exon 12 + 13 skipping, was not detected. **c)** RT-PCR using primers specific for two exonic cryptic donors shows their variant-associated increased use for *GSDME* c.1183 + 5 G > A. These rare events were missed during initial RNA analysis due to PCR biases and challenges resolving Sanger sequencing chromatograms due to heteroduplex formation. **d)** RT-PCR identifies rare use of two cryptic acceptors and exon 4 skipping associated with *EMD* c.266-3 A > G missed during initial RNA analysis due to PCR biases and heteroduplex formation. **e)** 49% (27/55) essential splice-site variants (ES) across the three variant datasets induce ≥1 in-frame events, with a similar proportion of 51% (45/88) in our overall cohort. Use of Intron Retention (IR) and Top-4* as proxy for a prediction of variant-induced mis-splicing increases the relative number of ES variants with ≥1 in-frame event to 80% (45/55) and to 81% (71/88) for our overall cohort. P = proband, C1 = control 1, C2 = control 2, DMSO = dimethyl sulfoxide, CHX = cycloheximide, CD1 = cryptic donor 1, CD2 = cryptic donor 2, CA1 = cryptic acceptor 1, CA2 = cryptic acceptor 2, CA3 = cryptic acceptor 3, ES = essential splice-site, NMD = nonsense mediated decay.

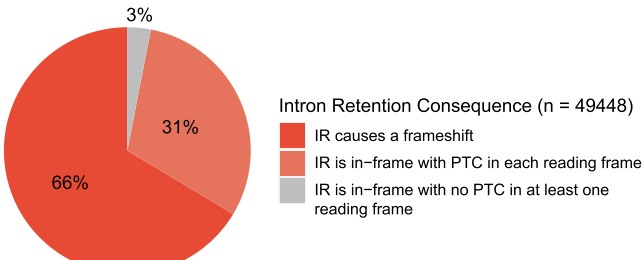

**Extended Data Fig. 9 | The consequence of intron retention upon the open reading frame for 49,448 canonical introns in clinically relevant Mendelian disease genes.** Variant-activated intron retention elicits a frameshift for 66% introns or encodes a premature termination codon (PTC) in all reading frames for 31% introns. In summary, IR will induce a frameshift or encode a PTC for at least 97% cases and is therefore consistent with null outcomes in most instances. Intron coordinates were extracted from Ensembl (104) via the hg38 genome assembly. IR: intron retention; PTC: premature termination codon.

| | PVS1_Very strong | PVS1_Strong | PVS1_Moderate |
|---|---|---|---|
| **Empirical evidence** | **1.** Intron retention (IR) and 300K-RNA Top-4* all result in a frameshift or encode a premature termination codon (PTC); | **1.** Intron retention (IR) and 300K-RNA Top-4* all result in a frameshift or encode a premature termination codon (PTC); **OR** **2.** Intron retention (IR) and 300K-RNA Top-4* include one in-frame event. | **1.** Intron retention (IR) and 300K-RNA Top-4* all result in a frameshift or encode a premature termination codon (PTC); **OR** **2.** Intron retention (IR) and 300K-RNA Top-4* include one or two in-frame events. |
| **Transcript disease association** | **3.** IR and 300K-RNA Top-4 events affect splicing of one or more constitutive exon(s) present in the clinically-relevant isoform(s) expressed by the manifesting tissue(s); **AND/OR** **4.** IR and 300K-RNA Top-4 events activate inclusion of ectopic/intronic sequences into the clinically-relevant isoform(s) expressed by the manifesting tissue(s); | | |
| **Pathogenetic mechanism** | **5.** Loss-of-function variants are a known causal basis for disease; | **5.** Loss-of-function variants are a known causal basis for disease; **AND/OR** **6.** Truncating variants, missense variants and/or in-frame indels are a known causal basis for disease | |
| | **6.** Transcripts with encoded PTCs are predicted to activate nonsense mediated decay | **6.** Transcripts with encoded PTCs are predicted to activate nonsense mediated decay; **AND/OR** **7.** Transcripts with encoded PTCs are in a genetic context that may evade NMD. However, there are one or more analogous truncating variants in the same region of the gene classified likely/pathogenic, **OR**, the truncated region is critical to function of the gene product. **8.** Transcript with an in-frame event altering length of the gene product disrupt a region of the gene with: i) evident clinical importance as shown by presence of one or more causal missense variants or in-frame indels; **AND/OR** ii) evident functional importance via disruption of a known domain critical to function of the gene product. | **6.** Transcripts with encoded PTCs are predicted to activate nonsense mediated decay; **AND/OR** **7.** Transcripts with encoded PTCs are in a genetic context that may evade NMD. However, there are one or more analogous truncating variants in the same region of the gene classified likely/pathogenic, **OR**, the truncated region is critical to function of the gene product. **8.** Transcripts with an in-frame event altering length of the gene product disrupt a region of the gene with: i) evident clinical importance as shown by presence of one or more causal missense variants or in-frame indels; **AND/OR** ii) evident functional importance via disruption of a known domain critical to function of the gene product; **AND/OR** iii) inferred functional importance via disruption of an evolutionarily conserved region intolerant to genetic variation. |

**Extended Data Fig. 10 | Draft Guidelines for potential use of empirical evidence from 300K-RNA to assist application of the PVS1 criterion for essential splice-site variants based on probable mis-splicing outcomes.** (being assessed in a clinical evaluation trial by the Australian Consortium for RNA Diagnostics (SpliceACORD)). We recommend pathology consideration of Intron Retention (IR) and 300K-RNA Top-4* for all disease relevant transcript(s). PVS1 levels of evidence are influenced by the collective nature of probable induced mis-splicing, relative to evidence supporting null outcomes for the encoded gene product. For use of PVS1 at a Very Strong evidence level, IR and Top-4* events should all be consistent with null outcomes. PVS1 applied at Strong or Moderate should be considered when IR and Top-4* events include one or two in-frame events, adjusting the evidence weighting according to; the number and nature of in-frame events, known clinical relevance of the affected region of disease relevant transcript(s), and the established pathogenetic mechanism(s) associated with a given gene and disorder. We favor weighting of known clinical relevance, biological function or evolutionary conservation of the disrupted gene region, over relative length of the in-frame disruption. We recommend additional consideration of abnormal or alternative transcription initiation or termination for first intron and last intron variants, respectively. We recommend use of the PM4 criterion for essential splice-site variants when IR and Top-4* events include three or more in-frame events.

# Reporting Summary

## Statistics

For all statistical analyses, confirm that the following items are present in the figure legend, table legend, main text, or Methods section.

| n/a | Confirmed | |
|---|---|---|
| ☐ | ☒ | The exact sample size (*n*) for each experimental group/condition, given as a discrete number and unit of measurement |
| ☒ | ☐ | A statement on whether measurements were taken from distinct samples or whether the same sample was measured repeatedly |
| ☐ | ☒ | The statistical test(s) used AND whether they are one- or two-sided *Only common tests should be described solely by name; describe more complex techniques in the Methods section.* |
| ☒ | ☐ | A description of all covariates tested |
| ☒ | ☐ | A description of any assumptions or corrections, such as tests of normality and adjustment for multiple comparisons |
| ☐ | ☒ | A full description of the statistical parameters including central tendency (e.g. means) or other basic estimates (e.g. regression coefficient) AND variation (e.g. standard deviation) or associated estimates of uncertainty (e.g. confidence intervals) |
| ☐ | ☒ | For null hypothesis testing, the test statistic (e.g. *F*, *t*, *r*) with confidence intervals, effect sizes, degrees of freedom and *P* value noted *Give P values as exact values whenever suitable.* |
| ☒ | ☐ | For Bayesian analysis, information on the choice of priors and Markov chain Monte Carlo settings |
| ☒ | ☐ | For hierarchical and complex designs, identification of the appropriate level for tests and full reporting of outcomes |
| ☐ | ☒ | Estimates of effect sizes (e.g. Cohen's *d*, Pearson's *r*), indicating how they were calculated |

*Our web collection on statistics for biologists contains articles on many of the points above.*

## Software and code

Policy information about availability of computer code

| Data collection | Custom R code was used to create 300K-RNA and is available at https://github.com/kidsneuro-lab/300K-RNA. Code developed in R version 3.6. Packages used: tidyverse 1.3.2, data.table 1.14.2. Code used to create SpliceVault is available at https://github.com/kidsneuro-lab/SpliceVault/. Code developed in R version 4.2.1. Packages used: data.table v1.14.2, DT v1.26, rintrojs v0.3.2, shinyBS v0.61.1, shinycssloaders v1.0.0, shinydashboard v0.7.2, shinyjs v2.1.0, shinyWidgets v0.7.4, DBI v1.1.3, tidyverse v1.3.2, DBI v1.1.3, odbc v1.3.3, faq v0.1.1, scales v1.1.1.  Code used for SpliceAI delta score retrieval API is available at https://gitlab.com/kidsneuro/SpliceAIAPI |
|---|---|
| Data analysis | Custom R code was used to perform data  analysis relevant to the paper and is available in the following repository; https://github.com/kidsneuro-lab/SpliceVault_figures. Code developed in R version 4.2.1. Packages used: data.table v1.14.2, tidyverse v1.3.2, UpSetR v1.4.0, cowplot v1.1.1, ggsci v2.9, ggpubr v0.4.0, scales v1.2.1, metR v0.12.0, readxl v1.4.1, stringr v1.4.1, jsonlite v1.8.2, TxDb.Hsapiens.UCSC.hg38.knownGene v3.15.0, openxlsx v4.2.5.1 |

For manuscripts utilizing custom algorithms or software that are central to the research but not yet described in published literature, software must be made available to editors and reviewers. We strongly encourage code deposition in a community repository (e.g. GitHub). See the Nature Portfolio guidelines for submitting code & software for further information.

# Data

Policy information about availability of data

All manuscripts must include a data availability statement. This statement should provide the following information, where applicable:
- Accession codes, unique identifiers, or web links for publicly available datasets
- A description of any restrictions on data availability
- For clinical datasets or third party data, please ensure that the statement adheres to our policy

Source files (i.e. SRA and GTEx splice junctions processed in recount3) were downloaded from snaptron; http://snaptron.cs.jhu.edu/data/srav3h/junctions.bgz and http://snaptron.cs.jhu.edu/data/gtexv2/junctions.bgz respectively. 300K-RNA can be easily accessed and queried through SpliceVault (https://kidsneuro.shinyapps.io/splicevault/). The data used for the analyses described in this manuscript were obtained from the GTEx portal and dbGaP accession number phs000424.v8.p2 and phs000424.v9.

## Human research participants

Policy information about studies involving human research participants and Sex and Gender in Research.

| | |
|---|---|
| Reporting on sex and gender | Not relevant to this study. This was a retrospective analysis where all conceivable splice outcomes were investigated for each variant. Sex was only relevant to ascertainment "segregates with disease" criteria detailed in Bournazos et al. Sex and gender were not reported for this study. |
| Population characteristics | There were no covariate-relevant population characteristics of the human research participants as the experimental results of RNA diagnostic studies was not compared between patients. Variant-induced mis-splicing events were established relative to age and sex matched controls for each patient independently. |
| Recruitment | Ascertainment criteria detailed in Bournazos et al. Ascertain variants with high clinical suspicion of causality were (1) a high likelihood of a monogenic Mendelian disorder, (2) variant allele frequency consistent with disease incidence, (3) putative splicing variant in a clinician-defined, phenotypically concordant gene, and (4) preferably the variant segregates with disease . |
| Ethics oversight | Consent for diagnostic genomic testing was supported by governance infrastructure of the relevant local ethics committees of the participating Australian Public Health Local Area Health Districts. Kids Neuroscience Centre's biobanking and functional genomics human ethics protocol was approved by the Sydney Children's Hospitals Network Human Research Ethics Committee (protocol 10/CHW/45 renewed with protocol 2019/ETH11736 (July 2019 – 2024)) with informed, written consent for all participants. |

Note that full information on the approval of the study protocol must also be provided in the manuscript.

# Field-specific reporting

Please select the one below that is the best fit for your research. If you are not sure, read the appropriate sections before making your selection.

☒ Life sciences   ☐ Behavioural & social sciences   ☐ Ecological, evolutionary & environmental sciences

For a reference copy of the document with all sections, see nature.com/documents/nr-reporting-summary-flat.pdf

# Life sciences study design

All studies must disclose on these points even when the disclosure is negative.

| | |
|---|---|
| Sample size | Determined by the number of variants for which we had experimental evidence to show were splice-altering through our RNA diagnostics program (Bournazos et al). |
| Data exclusions | Variants that did not alter splicing and variants creating or modifying the essential splice site motif of a cryptic splice site were excluded. |
| Replication | All splicing outcomes were experimentally confirmed by at least 2 primer pairs as per our standardized RNA diagnostics practices outlined in Bounazos et al. Some variants were confirmed by both RNAseq and RT-PCR. |
| Randomization | We did not randomise in our analysis of genetic splice-altering variants. All splicing variants were included in this study for which robust RNA assay data was available and met our inclusion criteria. Ascertainment criteria detailed in Bournazos et al. Ascertain variants with high clinical suspicion of causality were (1) a high likelihood of a monogenic Mendelian disorder, (2) variant allele frequency consistent with disease incidence, (3) putative splicing variant in a clinician-defined, phenotypically concordant gene, and (4) preferably the variant segregates with disease . For certain analyses we took random samples of RNA-seq samples (i.e. figure 2E and extended data figure 9). |
| Blinding | N/A.  The was a retrospective analysis were all conceivable splicing outcomes were investigated for each variant. |

# Reporting for specific materials, systems and methods

We require information from authors about some types of materials, experimental systems and methods used in many studies. Here, indicate whether each material, system or method listed is relevant to your study. If you are not sure if a list item applies to your research, read the appropriate section before selecting a response.

## Materials & experimental systems

| n/a | Involved in the study |
|-----|----------------------|
| ☒ ☐ | Antibodies |
| ☒ ☐ | Eukaryotic cell lines |
| ☒ ☐ | Palaeontology and archaeology |
| ☒ ☐ | Animals and other organisms |
| ☒ ☐ | Clinical data |
| ☒ ☐ | Dual use research of concern |

## Methods

| n/a | Involved in the study |
|-----|----------------------|
| ☒ ☐ | ChIP-seq |
| ☒ ☐ | Flow cytometry |
| ☒ ☐ | MRI-based neuroimaging |

