## [Peer Review File · Nature Genetics]

Peer Review Information

Manuscript Title: SpliceVault predicts the precise nature of variant-associated mis-splicing.

Corresponding author name(s): Professor Sandra Cooper

Reviewer Comments & Decisions:

Decision Letter, initial version:
--

31st Jan 2022

Dear Professor Cooper,

Your Analysis entitled "SpliceVault: predicting the precise nature of variant-associated mis-splicing." has now been seen by 3 referees, whose comments are attached. While they find your work of potential interest, they have raised serious concerns which in our view are sufficiently important that they preclude publication of the work in Nature Genetics, at least in its present form.

While the referees find your work of some interest, they raise concerns about the strength of the novel conclusions that can be drawn at this stage.

As you will see, Reviewer #1 thinks that the idea is of interest, but highlights some major limitations. Reviewer #2 also thinks that this has the potential to be a valuable contribution, but also has multiple major concerns. Reviewer #3 thinks that the advance here would be in the clinical diagnostic space, but that there are major shortcomings in the current analysis.

Importantly, all reviewers note that there needs to be the use of receiver operating characteristic curve or precision-recall curve to calculate the area under the curve for performance comparisons, not just PPV.

Reviewer #1 and Reviewer #2 have concerns about batch effects. Reviewer #2 also thinks that the data need to be harmonized through a single pipeline (which we agree is an important suggestion, but we recognize that this would entail a great deal of work).

Reviewer #1 thinks that there needs to be further (larger) validation datasets, Reviewer #2 suggests that the data should be sub-divided by source sample, and Reviewer #3 thinks that more needs to be done to show that this will be clinically useful.

We think that these issues are substantial enough to preclude publication at Nature Genetics.

Should further experimental data or analyses allow you to fully address these criticisms we would be willing to consider an appeal of our decision (unless, of course, something similar has by then been

accepted at Nature Genetics or appeared elsewhere). This includes submission or publication of a portion of this work someplace else.

If you are interested in attempting to revise this manuscript for submission to Nature Genetics in the future, please contact me to discuss a potential appeal. Otherwise, we hope that you find our referees' comments helpful when preparing your manuscript for resubmission elsewhere.

Thank you very much.

All the best,

Catherine

Catherine Potenski, PhD
Chief Editor
Nature Genetics
1 NY Plaza, 47th Fl.
New York, NY 10004
catherine.potenski@us.nature.com
<https://orcid.org/0000-0002-4843-7071>

Referee expertise:

Referee #1: bioinformatics, computational biology

Referee #2: genomics, transcriptomics, rare genetic disease

Referee #3: genetics, genomics, transcriptomics

Reviewers' Comments:

Reviewer #1:

Remarks to the Author:

Remarks to the Author:

Alternative splicing is one of the most important posttranscriptional regulations, while its misregulation is closely relevant to human disease. A large proportion of disease-related Mis-splicing is caused by genetic variations. In this study, Dawes et al proposed a hypothesis that genetic variant-associated Mis-spliced transcripts could be also detected in control RNA-seq samples from healthy individuals. This idea is very interesting, but also rational as it might share a similar philosophy as the previous finding, which reported that many mutations in the key cancer driver genes could also be observed in physiologically normal tissues. Based on this hypothesis, the author collates all unannotated splicing events surrounding each splice site by analyzing splicing junctions from 335,301 RNA-seq samples (300K-RNA). For each splicing site, they ranked all unannotated splicing events based on the number

of samples, in which the event is detected/present. They used the four most common ones (300K-RNA Top4) to identify Mis-splicing caused by genetic variants, in which they found 96% of exon-skipping events and 82% of cryptic splice-sites induced by 86 clinically reported variants could be identified. By comparing with the predictions from the other two methods, SpliceAI and MMSplice, they claimed that 300K-RNA Top4 outperforms these two to correctly identify multiple Mis-splicing events. Finally, they developed SpliceVault, a web portal to access unannotated splicing events in 300K-RNA Top4. In general, the idea is interesting, the flow of the manuscript is clear and straightforward. However, the limitations are also obvious, and some points should be addressed.

1, Some sentence in the abstract is very unfriendly to readers without background knowledge. For instance, Line 16-17, 'In comparison, applying interpretative rules to SpliceAI D-scores correctly identifies 55% of ...'. It will make people confused about what's the SpliceAI and what's the delta-scores? In addition, the authors did comparisons between 300K-RNA Top4 and SpliceAI and MMSplice. It would be rather weird to only mention SpliceAI in the abstract but did not mention MMSplice at all.

2, The author analyzed 86 variants across 72 genes to demonstrate the capacity of 300K-RNA Top4 for the identification of variant-induced Mis-splicing events. They should give a clear explanation about how they selected these variants. Do they include all current experimentally confirmed, splicing-related genetic variants? The author should provide a table to show details of these variants and genes. In addition, the author classified these variants into different categories (Fig. 1A), what's the purpose of this analysis? Is there any expectation that variants from different categories show different validation rates by 300K-RNA Top 4?

3, The largest limitation of 300K-RNA Top4 is that the positive predictive value (PPV) is very low compared to other methods, which means FDR is very high in the other words. For instance, the FDR of exon skipping prediction by 300K-RNA is 67%. The author also claimed that '300K-RNA Top4 outperforms SpliceAI (and MMSplice)' in Line 232. Based on Fig. 2E, under similar PPV/FDR, for instance (69% for 300K-RNA, 65% for SpliceAI), sensitivity/recall of two methods (50% and 59% respectively) are very closed and 300K-RNA performs even slightly worse than SpliceAI. A fair comparison should use the receiver operating characteristic (ROC) curve or precision-recall curve to calculate the area under the curve (AUC).

4, As the precision of 300K-RNA Top 4 is relatively low, even only using the Top 1 splicing events, 40% of the prediction could be false positives, the author should provide a deeper investigation regarding what's the difference between those 'Seen in RNA studies' and those not. For instance, do those skipped exons Seen in RNA studies are shorter than those not? Do these events have relative stronger/weaker splicing strength? Are they more GC/AT enriched et al?

5, The authors showed the performance on Mis-splicing prediction with three methods, respectively. What's the overlap between each other? What're the characteristics of the events that could be identified by all three methods or could only be identified by each of them?

6, In Line 153, the author claimed that intron retention cannot be predicted by 300K-RNA. It's not correct as there are no significant differences between the detection of intron retention and other types of splicing events from RNA-seq data. Theoretically, detection of reads spanning across the exon-intron boundary, if it's not easier, it's quite similar to the detection of splicing junctions spanning two exons. Almost all well-known computational tools for splicing analysis based on RNA-seq data have the capacity of intron retention identification, such as MISO (Katz, Yarden, et al. 2010), rMATS (Shen, Shihao, et al.2014), and PSI-Sigma (Lin, Kuan-Ting, and Adrian R. Krainer. 2019). Moreover,

some tools specifically for the detection of intron retention have also been developed, for instance, IRFinder (Wong, Justin J-L., et al. 2013). As 20% of the Mis-splicing events induced by the 86 variants are intron retention, which is not a negligible proportion. The author should characterize the intron retention events, at least for these 29 events (Line 89), in RNA-seq samples with a relatively large sample size to get robust Top-ranked events.

7, The authors analyzed 335,301 RNA-seq samples from GTEx and SRA. They should clarify if any batch effect correction should be considered? They should also include an ablation study to show the robustness of those top-ranked splicing events. For instance, using 5K, 10K, 50K, 100K, and 300K samples, what's the overlap of the top identified unannotated splicing events. Especially, for the 147 variants induced Mis-splicing events, what's the PPV and sensitivity if only using 10K, 50K, 100K samples?

8, As the key finding of this study, is that 300K-RNA Top 4 could accurately predict 90% of the variants-associated Mis-splicing events. Only using a single dataset (147 events) might be not sufficient to prove its true sensitivity. The author should either apply the analysis on another independent dataset (if it's possible) or experimentally validate the predictions beyond the 86 variants. For instance, they can use CRISPR to mutate several splice sites and check whether the mutation-induced splicing variants indeed could be identified by 300K-RNA Top4.

Minor Points

1, Line 80, the authors mentioned that multiple experimental approaches have been performed to characterize aberrant splicing events arising from 86 genetic variants. It would be better to make a table to show how many of them were confirmed by each approach, including RT-PCR, RNA-seq, and/or minigene assay.

2, Line 84, '77 SNVs were included in the dataset' is contradict to 78 SNVs (59 + 19) shown in Figure 1B. The author should clarify which number is correct. In addition, it'd be better to also show the relative position of "2 insertions, 5 deletions, and 2 deletion-insertion variants" to the splice sites in the figure.

3, Line 223, the authors claimed that 'no previous method can predict the nature of variant-induced mis-splicing'. However, both SpliceAI and MMSplice are public methods. Did the authors want to conclude that the prediction from SpliceAI and MMSplice is not 'the nature of variant-induced mis-splicing'?

4, Since the authors performed RNA reanalysis for 4 cases, it would be interesting to characterize full-length of those mis-spliced transcripts using the third-generation sequencing technology. The results could be more convincing compared to bands in the gel which only amplified parts of the mis-spliced transcripts.

Reviewer #2:

Remarks to the Author:

I read the paper by Dawes et al with great interest. The notion that mis-splicing events caused by genomic variant can be predicted through reference to known mis-splicing events in the general population is intuitive. Whilst data resources to complete these types of analyses are publicly available, I am not aware of an existing resource of the scale described in this manuscript.

The authors utilized RNA-seq datasets from GTEx and SRA to identify mis-splicing events in the

general population, and present an analysis of 40,000 samples to show the capability of this acquired knowledge of mis-splicing events to identify cryptic splice sites (Nat Comms paper), and an analysis of 300,000 samples in the context of cryptic splice sites and exon skipping (Nat Genetics paper). I have read both papers, but have restricted this review to encompass only the information presented in the 300K splice vault (Nature Genetics) paper. I note some areas of natural complementarity and overlap between the papers, but feel that the 300K SpliceVault lacks significant detail in areas, some of which is covered to some extent in the 40K paper, but when considered as a standalone manuscript isn't acceptable in its current state for publication. I believe this resource and manuscript could be of broad interest and an informative resource for variant interpretation, but have noted major issues with the methodologies which should be addressed before further consideration:

1. The data utilized to generate the 300K resource will be diverse, with a number of different tissues/cell-types/biosamples used and created through different wet laboratory and bioinformatics pipelines. The authors state that the 'splice-junction read counts were summarised' using a program called Datamash. I am not familiar with Datamash but believe it to be a simple command-line unix system programme capable of generating summary statistics. This is the extent of the description of data curation, but this is the most important part of the manuscript in my opinion. There is no explanation of the data curation and/or attempts to harmonize raw datasets through a single uniformed bioinformatics pipeline. Drawing analogies with human genomic resources, e.g. gnomAD (<https://gnomad.broadinstitute.org/>), the power of these databases lies to some extent in the unified data processing and analysis approaches applied to the data and to thereby identify and filter for artefactual findings. Such approaches for data harmonization will make the 300K resource more powerful, and whilst practically difficult, is to some extent why such a resource is yet to be realised. My suggestions are two-fold: (1) a summary of the different biosamples and wet laboratory approaches used in the analysis should be added; and (2) harmonization of the data through a single pipeline, e.g. the GTEx RNA-seq pipeline (<https://github.com/broadinstitute/gtex-pipeline>) or a harmonised use of algorithms to statistically identify mis-spliced events from RNA-seq, e.g. FRASER (<https://www.nature.com/articles/s41467-020-20573-7>) or DROP (<https://www.nature.com/articles/s41596-020-00462-5>). The authors should also expand their description of mis-splicing events to include quality metrics and appropriate thresholds (e.g. read depth, quality and mapping thresholds).

2. In addition to comment #1, there is no attempt to sub-divide mis-splicing events in the 300K resource by biosample sources. Such an analysis could lead to interesting and important observations in the data, and contribute to how to expand the resource going forward, e.g. does the use of tissue-shared Vs tissue-specific mis-splicing events impact the accuracy of the Top-4 approach? Do certain biosamples exhibit higher levels of mis-splicing than others? Is the expression of isoform X in tissue A associated with higher levels of mis-splicing than in isoform Y in tissue B? Whilst the composite resource presented here provides power by numbers, an analysis enabling the calculation of mis-splicing events by biosample type may enable further improvements to the algorithm and I think this is an area the authors should explore. In addition, the authors remark that their RNA analyses performed in blood and fibroblasts were informed by the diverse biosample resource, can the authors also comment on the suitability of their investigated genes in clinically accessible tissues? e.g. through isoform similarity evaluated through MAJIQ-CAT? (<https://pubmed.ncbi.nlm.nih.gov/32225167/>) – can the authors comment on the utility of the 300K resource for genes that aren't appropriate for surveillance from clinically accessible tissues?

3. The 300K manuscript uses specific thresholds for each of the approaches for prioritization (Top-4, 0.05, ≤ -2), but makes no attempt to explain why the Top-4 approach is selected. Whilst there is presentation of sensitivity and PPV information in Figure 2E (which is assumed to be the 86 experimentally validated splice-altering variants – this isn't declared in the figure gene), a formal analysis of optimal thresholds could be performed using the Precision-Recall or Receiver Operator Curve. The authors already declare that PPV isn't their primary objective, and they place more emphasis on sensitivity (lines 236-239), so why isn't a Top-7 approach used with 97% sensitivity and ~25% PPV (Figure 2E)? In addition to comment #2, consideration of optimal thresholds for tissue-shared Vs tissue-specific mis-splicing events may be informative, and an interesting analysis to add here.
4. The interpretative rules for SpliceAI require greater explanation in the text section of the methods. While figure 1 and explanation in the results section enables greater clarity, the methods are not clear as a standalone piece of text. Some explicit justifications for the use of 0.05 thresholds are also required.
5. There are two fairly recent algorithms that the authors should also consider in their comparative analysis (CADD-Splice, <https://genomemedicine.biomedcentral.com/articles/10.1186/s13073-021-00835-9> & SQUIRLS, [https://www.cell.com/ajhg/pdfExtended/S0002-9297\(21\)00238-X](https://www.cell.com/ajhg/pdfExtended/S0002-9297(21)00238-X)).
6. The authors note that over half of the variants included two or more mis-splicing events (line 87). This is an interesting observation which would benefit from further elaboration and discussion. Can the authors demonstrate the relative dosage of each aberrant junction to the canonical junction? How does the relative dosage of the mis-splicing events impact the accuracy of the algorithms to predict multiple mis-splicing events? E.g. Rowlands et al (<https://doi.org/10.1016/j.ajhg.2021.12.014>) describe a normalized read count applied during the Cummings et al pipeline (10.1126/scitranslmed.aal5209) to rank mis-splicing events. Recent updates to ACMG guidelines have suggested that partial impacts are an important aspect of variant interpretation (<https://doi.org/10.1101/2021.12.28.21267792>), and the authors have an opportunity to quantify these important observations in over 40 cases and to put this in greater context with the existing literature around variant interpretation.
7. Figure 4 suggests use of PVS1 ACMG codes in the context of the presented evidence. Whilst this is a natural and relevant extension of the presented work, I would suggest that these recommendations are more relevant for incorporation through ongoing work in the ClinGen community. This will enable wider panels of experts to evaluate and comment on guidance before it is made available in literature, and to link these seamlessly with other updates to guidance on PVS1 for splicing variants, which can result in rapid uptake of guidance across diagnostic genomic centres: <https://clinicalgenome.org/working-groups/sequence-variant-interpretation/>.

Reviewer #3:

Remarks to the Author:

Overview

Dawes et al propose a new method addressing an unmet need: predicting the splice isoform outcome of essential splice-site variants. To this end, the authors suggest that mis-splicing events are typically

resolved into an alternative splice isoform that can be detected, though at a low level, in unaffected samples. As a corollary, variants affecting splicing of genes that do not express non-functional splicing isoforms are unlikely to affect gene function.

The idea is original and goes conceptually beyond what sequence-based variant predictors (SpliceAi, MMSplice, etc.) have been addressing so far. However, the presented results lack important analyses, leading to concerns about the actual relevance for rare disease diagnostics. At this stage, the relevance of the methods and of the endeavor itself is not substantiated enough.

Major points

The groundwork of extensively mapping weak splice junctions was done previously by the authors (Dawes et al. 2021). The added value of this paper should be to provide a justification of the new diagnostic guidelines proposed in Fig. 4. However:

- The paper should provide estimates of how frequently essential splicing variants do not result in IR, PTC or frameshift. Unless this represents a substantial fraction of essential splicing variants, the need for a new guideline is not so impactful. It may well be large, but this number is lacking.

- Assuming this fraction is indeed substantial, the authors should then provide estimates as to how frequent it is that the proposed very strong evidence criteria does not apply (i.e. that not all 300 K-RNA top-4 result in IR, PTC or frameshift). Introducing the new guideline is relevant only if this fraction is substantial. Here too, the fraction may well be large, but this number is lacking.

- The analysis focuses on sensitivity alone using a benchmark dataset consisting of pathogenic variants only. Pathogenic variants are biased for variants resulting in loss-of-function splice isoforms. While sensitivity may matter most, the study needs to also assess a substantial set of splice variants that do not resolve in loss-function isoforms to justify the utility of the tool and demonstrate that those would be often appropriately discarded.

- The authors should discuss whether, according to their guidelines, very strong predictions would not need an RNA-seq or RT-PCR validation anyways. If so, then the added value is limited.

Going now through the manuscript:

- Fig 2A-C. There are striking mismatches between the reported numbers used to compute sensitivity and PPV, and the actual number of points displayed on the figure. An obvious example is the number "5" below cutoff for SpliceAi in Fig 2A. That is pervasive (are there 128 data points not seen in RNA studies and below cutoff for top-K RNA in Fig 2A?). There are probably reasons for this, e.g. what is plotted are splice-site levels scores, and what is counted are splicing events but this should be explained. I would suggest to show on the plot what is actually counted and to provide a clear rationale about how "things" are counted for each method. The same "philosophy" should be applied to each method.

- Fig 2A is based on fixed cutoffs. Instead, precision-recall curves should be shown. This is a major concern because the authors have adjusted the cutoff of their method to increase sensitivity but did not investigate changes of cutoff of other methods. The authors argue that sensitivity matters and should make use of the precision-recall curve to compare the performance of all tools at same recall (i.e. sensitivity).

- it is incorrect to state that "All approaches have limited or no ability to predict intron retention". The MMSplice software can be used to predict splicing efficiency which models intron retention. This should be tried.

- It is excessive to state "Whereas SpliceAI predicted only a single mis-splicing event for 17/20 variants". It would be more appropriate to state that the algorithm the authors proposed on top of SpliceAI "predicted only a single mis-splicing event....". Technically, SpliceAI outputs scores for creation or loss of splice sites. One could propose several ways of predicting isoform outcomes from these scores. Here, the authors propose a greedy approach considering every site with score > 0.2 to be used. It is not expected that the authors over-engineer a method on top of SpliceAI. However, since they value sensitivity so high, they should use SpliceAI in a way that maximizes the number of isoform predictions. For instance, one could consider drawing independently sites using the Splice scores and obtaining thereby a distribution of isoforms from which one could retain the most probable ones. This would be more comparable to the "top-4" approach proposed by the authors.

- 300K Top-4 relies on a genome annotation but no detail is given about this. The choice of the genome annotation could be critical. If the genome annotation of splice isoforms is too stringent, say, limited to a single major isoform per gene, then the most frequent non-annotated isoform (top-1) could often be a genuine, biologically functional splice isoform. At the other extreme, if the genome annotation is too inclusive then it can contain rare, non-functional splice isoforms, exactly those one would instead like to retain. On what basis did the authors decide which isoforms were "annotated"? Some assessment would be insightful.

Minor points

- Line 50, the observation that aberrant splicing junctions are typically detected at weak levels in other samples was already reported by Kremer et al., Nat Commun, 2017, with a quantification of the prevalence of this phenomenon consistent with the much more extensive and recent analyses by Dawes et al., 2021. Also, Kremer et al. also formulated the hypothesis of the present study described l. 64-66. This prior work should be cited in this respect.

- Figure 2 is entitled "Accuracy" but what is assessed is sensitivity.

- Designing targeted RT-PCR follow-ups guided by the predictions have moderate value as they require specific primer designs and are blind to not-predicted isoforms. Would not the authors suggest RNA-seq, potentially with cycloheximide, in any case? This should be discussed.

Author Appeal

Reviewer Comments:

Author's Response to all reviewers:

All reviewers acknowledged the clinical importance and our discovery and offered several constructive suggestions how we could improve evidence supporting the rigour and robustness of the 300K-RNA Top 4 method. All three reviewers unanimously highlighted; 1) precision-recall analyses, 2) evidence for concordance among the ranking of unannotated splicing events between tissues and sources, and 3) additional validation and identification of sources of bias; as suggested areas for improvement.

Thus, we now include additional Figures and Supplemental Figures:

1. Precision-recall analyses
2. Updated SpliceVault and 300K-RNA source data to utilise RNA-Seq splice-junction data processed using the unified Monorail pipeline.
3. Updated the SpliceVault portal to enable users the flexibility to access ranked, unannotated splicing events *within* different tissue specimen subtypes within GTEx.
4. Further validation of the Top-4 method using two additional, independent cohorts of clinical variants curated from literature (61 variants studied in patient specimens and 63 variants studied through midi-gene assays).
5. Concordance analyses of the ranking of unannotated splicing events for the 88 splice-sites affected by our cohort of 88 splice-altering clinical variants, and for all unannotated splicing events detected for 98,810 annotated splice-sites across 3,668 clinically relevant Mendelian disease genes (the Mendeliome) – between different biospecimen subtypes within GTEx, and, between GTEx and SRA datasets.
6. For 19/88 variants associated with muscle disorders and where RNA testing was performed using muscle, we showed that GTEx Muscle Top-4* (n = 881 muscle samples) results in lower sensitivity and PPV than using 300K-RNA Top-4* (though discuss that higher sequencing depth of human tissues may ultimately result in improved accuracy).
7. Conducted deep analysis of features that help to discriminate ‘true-positive’ from ‘false-positive’ events to inform prospective improvement of the ranking method.
8. Quantified that ~50% of essential splice site variants are associated with one or more in-frame event (across all three validation cohorts) – confirming the clinical importance of using SpliceVault to accurately consider the probable nature of variant-associated mis-splicing for informed use of PVS1, PM4 and PP3 criteria of the ACMG-AMP guidelines.

With a large body of additional data, though we conducted side-by-side comparative analyses with MMSplice and SPANR as recommended, it became very cumbersome to include and discuss comparisons for multiple algorithms. As an additional note to R3, CADD-Splice does not use an independent algorithmic approach, rather, it integrates SpliceAI and MMSplice along with other features.

As this manuscript that seeks to validate the diagnostic utility and predictive accuracy of the 300K-RNA Top-4 approach, we remove all comparative reference to MMSplice and SPANR. We maintain comparative evaluation between 300K-RNA and Splice AI, due to their mutual ability to predict both cryptic activation and exon skipping to enable precision-recall analyses. We also feel the comparison of an evidence-based approach with a sophisticated, deep learning approach, is of broad general interest to the Nature Genetics readership. Our point-by-point rebuttal to your comments follows below. We are very appreciative of the constructive suggestions offered, which have improved the body of evidence supporting the rigour and accuracy of our Top-4* method.

Reviewer #1:

1. Some sentence in the abstract is very unfriendly to readers without background knowledge. For instance, Line 16-17, 'In comparison, applying interpretative rules to SpliceAI D-scores correctly identifies 55% of ...'. It will make people confused about what's the SpliceAI and what's the delta-scores? In addition, the authors did comparisons between 300K-RNA Top4 and SpliceAI and MMSplice. It would be rather weird to only mention SpliceAI in the abstract but did not mention MMSplice at all.

Author's Response

We have edited the abstract to be friendlier to readers not familiar with the terminology of SpliceAI scores, simply saying "300K-RNA showed higher sensitivity and positive predictive value than SpliceAI in predicting exon-skipping and cryptic-activation events".

Please see "*Response to all Reviewers*" above for an explanation of our reasoning to remove reference to MMSplice within this manuscript.

2. The author analyzed 86 variants across 72 genes to demonstrate the capacity of 300K-RNA Top4 for the identification of variant-induced Mis-splicing events. They should give a clear explanation about how they selected these variants. Do they include all current experimentally confirmed, splicing-related genetic variants? The author should provide a table to show details of these variants and genes. In addition, the author classified these variants into different categories (Fig. 1A), what's the purpose of this analysis? Is there any expectation that variants from different categories show different validation rates by 300K-RNA Top 4?

Author's Response

Ascertainment criteria are clearly defined in the first paragraph of results (see manuscript text lines 87-90). We add Table S1 that details all variants confirmed to disrupt splicing and the detected splicing outcomes. The purpose of Figure 1A is to show that our cohort of 88 variants encompasses a wide range of disorders with different tissue-specific or multi-system involvement.

3. The largest limitation of 300K-RNA Top4 is that the positive predictive value (PPV) is very low compared to other methods, which means FDR is very high in the other words. For instance, the FDR of exon skipping prediction by 300K-RNA is 67%. The author also claimed that ‘300K-RNA Top4 outperforms SpliceAI (and MMSplice)’ in Line 232. Based on Fig. 2E, under similar PPV/FDR, for instance (69% for 300K-RNA, 65% for SpliceAI), sensitivity/recall of two methods (50% and 59% respectively) are very close and 300K-RNA performs even slightly worse than SpliceAI. A fair comparison should use the receiver operating characteristic (ROC) curve or precision-recall curve to calculate the area under the curve (AUC).

Author’s Response

We agree. New Figure 2B shows precision-recall curves for 300K-RNA and SpliceAI at Δ -score thresholds that offer the same, total number of predictions as offered by 300K-RNA at each rank cut-off – such that true/false positive rates can be compared. This demonstrates 300K-RNA has higher sensitivity and comparable positive predictive value at every cut-off except 300K-RNA top-1, where the scores are similar. Further analyses of precision and recall are provided in Figures 2C, 2D, S1C, S1E, S1F, S4E.

4. As the precision of 300K-RNA Top 4 is relatively low, even only using the Top 1 splicing events, 40% of the prediction could be false positives, the author should provide a deeper investigation regarding what’s the difference between those ‘Seen in RNA studies’ and those not. For instance, do those skipped exons Seen in RNA studies are shorter than those not? Do these events have relative stronger/weaker splicing strength? Are they more GC/AT enriched et al?

Author’s Response

We conducted evaluation of Top-4* true/false positives in new Figures S1B and S2A-D and manuscript commentary in Lines 164 – 168 and 175 – 184. RNA re-analyses detecting Top-4 events suggest precision may be a little higher than we currently report. We determined in Figure S2 that true positives tend to be associated with:

- Higher sample count i.e. seeing an event in more samples. We don’t wish to over emphasize this point, given so many variant-activated events are only seen in a few samples.
- Double-exon skipping is substantially less likely to occur than single exon skipping.

- Double versus single exon skipping is not related to the overall length (in nt) of the excised region (or exon length, not shown).
 - We do not compare or present analyses of splice-site strength or use of GC/AT enrichment among unannotated splicing events as this was investigated in detail in Dawes et al. 2022 NCOMMS21-28137 (proofs returned to NCOMMS. Online publication imminent).
5. The authors showed the performance on Mis-splicing prediction with three methods, respectively. What's the overlap between each other? What're the characteristics of the events that could be identified by all three methods or could only be identified by each of them?

Author's Response

Please see “*Response to all Reviewers*” above for an explanation of our reasoning to remove reference to additional algorithms. We agree this is an interesting question and feel comparative evaluation of multiple contemporary splicing prediction tools warrants investigation as an entire publication in its own right. Our focus here is the diagnostic utility and predictive accuracy of the 300K-RNA Top-4 evidence-based approach.

6. In Line 153, the author claimed that intron retention cannot be predicted by 300K-RNA. It's not correct as there are no significant differences between the detection of intron retention and other types of splicing events from RNA-seq data. Theoretically, detection of reads spanning across the exon-intron boundary, if it's not easier, it's quite similar to the detection of splicing junctions spanning two exons. Almost all well-known computational tools for splicing analysis based on RNA-seq data have the capacity of intron retention identification, such as MISO (Katz, Yarden, et al. 2010), rMATS (Shen, Shihao, et al. 2014), and PSI-Sigma (Lin, Kuan-Ting, and Adrian R. Krainer. 2019). Moreover, some tools specifically for the detection of intron retention have also been developed, for instance, IRFinder (Wong, Justin J-L., et al. 2013). As 20% of the Mis-splicing events induced by the 86 variants are intron retention, which is not a negligible proportion. The author should characterize the intron retention events, at least for these 29 events (Line 89), in RNA-seq samples with a relatively large sample size to get robust Top-ranked events.

Author's Response

300K-RNA is a database of detected splice-junctions from split-reads in RNA-seq data. Intron retention is inferred by non-split reads spanning the exon-intron boundary; absence of a splice-junction cannot be calculated from the 300K-RNA data sources. Quantification of intron retention is confounded by the presence of non-split reads corresponding to pre-mRNA and splicing intermediates. We suspect intron

retention quantification may differ between RNA-Seq data derived using ribosomal depleted total RNA versus polyA depletion, because:

- a) transcripts with retained introns can be non-adenylated and retained in the nucleus
- b) introns can be long. Transcripts with retained introns can be particularly impacted by 3' bias associated with PolyA enrichment protocols.

Thus, herein we recommend pathology consideration of intron retention in addition to Top-4. But we do not advise or recommend how it may be used in a Top-4 style ranking method to predict the nature of probable mis-splicing.

7. The authors analyzed 335,301 RNA-seq samples from GTEx and SRA. They should clarify if any batch effect correction should be considered? They should also include an ablation study to show the robustness of those top-ranked splicing events. For instance, using 5K, 10K, 50K, 100K, and 300K samples, what's the overlap of the top identified unannotated splicing events. Especially, for the 147 variants induced Mis-splicing events, what's the PPV and sensitivity if only using 10K, 50K, 100K samples?

Author's Response

We adapted SpliceVault to source data processed using the monorail approach of Wilks et al.¹ Inherent robustness and reproducibility of the 300K-RNA Top-4* method is shown by highly similar statistical metrics using both sets of source data. We further conducted correlative analyses of the ranking of unannotated splicing events around: i) the 88 splice sites impacted by our cohort of variants (Figure 3A-B) and ii) 98,810 annotated splice-sites in 3,668 clinically relevant Mendelian disease genes (the Mendeliome) (Figure S4A-B), comparing:

- 1) biospecimen subtypes within GTEx
- 2) GTEx and SRA datasets

including overall correlation of sample counts for each unannotated splicing event.

- We confirm the number of mis-splicing events detected in each individual tissue in GTEx is correlated with the number of samples present for that tissue (Figure S4D).
- We show the overlap of the top-4 300K-RNA events with the top-4 events present in four clinically accessible tissues in GTEx (Figure 3D)
- We show that using GTEx muscle Top-4 shows lower sensitivity than using 300K-RNA Top-4 for 19 variants associated with muscle disorders where RNA testing was conducted using muscle RNA (Figure S4E).

However, our overall opinion is the best predictive accuracy of a ranking approach may be garnered via a balance of breadth and depth i.e. high sequencing read depth of a reasonable number of specimens (e.g. hundreds but not thousands). We are keenly investigating this, as well as potential benefits of using human primary cell lines treated with cycloheximide to inhibit NMD, which may influence event ranking.

8. As the key finding of this study, is that 300K-RNA Top 4 could accurately predict 90% of the variants-associated Mis-splicing events. Only using a single dataset (147 events) might be not sufficient to prove its true sensitivity. The author should either apply the analysis on another independent dataset (if it's possible) or experimentally validate the predictions beyond the 86 variants. For instance, they can use CRISPR to mutate several splice sites and check whether the mutation-induced splicing variants indeed could be identified by 300K-RNA Top4.

Author's Response

We agree and conducted validation of the sensitivity and positive predictive value of 300K-RNA Top-4* predictions on two additional and independent datasets: 121 variants from published patient-derived RNA (n=58) and midi-gene (n=63) studies are also now included in Table S1. We scrutinised the technical design of the midi-genes, to ensure multi-exon skipping and distal cryptics, if activated, could be detected. Importantly, we obtained a similar average sensitivity of 92% and average PPV of 29%.

Minor Points

9. Line 80, the authors mentioned that multiple experimental approaches have been performed to characterize aberrant splicing events arising from 86 genetic variants. It would be better to make a table to show how many of them were confirmed by each approach, including RT-PCR, RNA-seq, and/or minigene assay.

Author's Response

We provide Table S1 that details all variants evaluated in this study, the experimental approaches used, RNA source and splicing outcomes.

10. Line 84, '77 SNVs were included in the dataset' is contradict to 78 SNVs (59 + 19) shown in Figure 1B. The author should clarify which number is correct. In addition, it'd be better to also show the relative position of "2 insertions, 5 deletions, and 2 deletion-insertion variants" to the splice sites in the figure.

Author's Response

Figure 1B has been corrected and now represents an updated list of experimentally-verified variants, details provided in Table S1.

11. Line 223, the authors claimed that 'no previous method can predict the nature of variant-induced mis-splicing'. However, both SpliceAI and MMSplice are public methods. Did the authors want to conclude that the prediction from SpliceAI and MMSplice is not 'the nature of variant-induced mis-splicing'?

Author's Response

We have removed this sentence and reference to MMSplice.

12. Since the authors performed RNA reanalysis for 4 cases, it would be interesting to characterize full-length of those mis-spliced transcripts using the third-generation sequencing technology. The results could be more convincing compared to bands in the gel which only amplified parts of the mis-spliced transcripts.

Author's Response

We can only redress this comment by advising that we agree entirely - and are currently conducting a collaborative (via SpliceACORD) clinical evaluation trial of PCR followed by long-read sequencing of amplicons to see if it: 1) improves sequencing clarity of multi-template PCRs, which are inherent in a subject with a heterozygous splice-altering variant, and common for homozygous or hemizygous variants that elicit more than one mis-splicing outcome; 2) presents cost-savings over Sanger sequencing.

Reviewer #2:

1. The data utilized to generate the 300K resource will be diverse, with a number of different tissues/cell-types/biosamples used and created through different wet laboratory and bioinformatics pipelines. The authors state that the ‘splice-junction read counts were summarised’ using a program called Datamash. I am not familiar with Datamash but believe it to be a simple command-line unix system programme capable of generating summary statistics. This is the extent of the description of data curation, but this is the most important part of the manuscript in my opinion. There is no explanation of the data curation and/or attempts to harmonize raw datasets through a single uniformed bioinformatics pipeline. Drawing analogies with human genomic resources, e.g. gnomAD (<https://gnomad.broadinstitute.org/>), the power of these databases lies to some extent in the unified data processing and analysis approaches applied to the data and to thereby identify and filter for artefactual findings. Such approaches for data harmonization will make the 300K resource more powerful, and whilst practically difficult, is to some extent why such a resource is yet to be realised. My suggestions are two-fold: (1) a summary of the different biosamples and wet laboratory approaches used in the analysis should be added; and (2) harmonization of the data through a single pipeline, e.g. the GTEx RNA-seq pipeline (<https://github.com/broadinstitute/gtex-pipeline>) or a harmonised use of algorithms to statistically identify mis-spliced events from RNA-seq, e.g. FRASER (<https://www.nature.com/articles/s41467-020-20573-7>) or DROP (<https://www.nature.com/articles/s41596-020-00462-5>). The authors should also expand their description of mis-splicing events to include quality metrics and appropriate thresholds (e.g. read depth, quality and mapping thresholds).

Author’s Response

We appreciate this constructive suggestion and contemporised SpliceVault to source data processed identically using the monorail processing pipeline¹. In line with suggestions by R1 and R2, we provide Table S1 that details all variants, the wet-lab approaches and the source of the RNA. We have also provided improved clarity in Methods and Results regarding specific procedural steps to ascertain variants, rank the events etc. See response to R1 Q7 regarding our extensive validation of method sensitivity and positive predictive value, which is not substantially influenced by RNA-Seq method or sample sub-type for exon-skipping and cryptic splice site junctional reads. We do feel there will be influence of technical pipeline for quantification of intron retention – underpinning our decision to not comment at this early stage on use of IR non-split reads in our Top-4 ranking method for clinical use.

2. In addition to comment #1, there is no attempt to sub-divide mis-splicing events in the 300K resource by biosample sources. Such an analysis could lead to interesting and important observations in the data, and contribute to how to expand the resource going forward, e.g.
 - a. does the use of tissue-shared Vs tissue-specific mis-splicing events impact the accuracy of the Top-4 approach?
 - b. Do certain biosamples exhibit higher levels of mis-splicing than others?

- c. Is the expression of isoform X in tissue A associated with higher levels of mis-splicing than in isoform Y in tissue B?

Whilst the composite resource presented here provides power by numbers, an analysis enabling the calculation of mis-splicing events by biosample type may enable further improvements to the algorithm and I think this is an area the authors should explore.

3. In addition, the authors remark that their RNA analyses performed in blood and fibroblasts were informed by the diverse biosample resource, can the authors also comment on the suitability of their investigated genes in clinically accessible tissues? e.g. through isoform similarity evaluated through MAJIQ-CAT? (<https://pubmed.ncbi.nlm.nih.gov/32225167/>) can the authors comment on the utility of the 300K resource for genes that aren't appropriate for surveillance from clinically accessible tissues?

Author's Response

We have increased the functionality of the SpliceVault portal to enable users customized access to “all of 300K-RNA” or any one of the 56 specific specimen sub-types within the GTEx data source.

a) Please see our response to R1 Q7, with respect to our comprehensive evaluation of concordance in ranked unannotated splice events between GTEx tissues, between GTEx and SRA, for both our cohort of 88 splice-sites and across the Mendeliome. We showed the same concordance across every annotated human splice site, though felt with our clinical focus the Mendeliome was more relevant.

Figure S4 provides precision-recall analyses showing 300K-RNA Top-4* outperforms GTEx Skeletal muscle Top-4* versus for 19 variants subject to RNA testing of muscle. However, we feel that tissue-specific SpliceVault databases that encompasses deeper sequencing of tissue specimens from individuals of different ages could improve predictive accuracy. We are currently pursuing this hypothesis for blood, fibroblasts and muscle that we are studying in our clinical program of RNA Diagnostics (typically conducting ~150 million read depth +/- cycloheximide to inhibit NMD for primary cells, as compared to 20 million read depth for GTEx specimens).

b) The number of mis-splicing events detected correlates strongly with the number of samples present in GTEx for each biosample type (Figure S4D). Testis are a notable outlier – well known to have ‘leaky’ higher expression of many genes that show tissue-restricted expression.

c) Unfortunately, this important point requires long-read RNA sequencing from a clinical cohort of affected individuals with splice-altering variants, compared to controls or disease controls. This is a current area of our investigation, though falls outside the scope of this manuscript.

Q3) For RNA diagnostic testing performed in our laboratory, review of alternative splicing patterns in the target gene was performed using GTEx RNA-seq data derived from clinically accessible tissues and manifesting tissues to establish that testing in clinically accessible tissues was appropriate (see Bournazos et al., 2022)². In regards to the suitability of 300K-RNA resource for genes not expressed in clinically relevant tissues, See lines 216-220: *“Therefore, we recommend use of 300K-RNA Top-4* as prediction of the probable nature of variant associated mis-splicing until we have great enough breadth and/or depth of RNA-Seq data to evaluate a tissue-specific approach. We advise caution for genes with known tissue-specific or developmental alternative splicing, where RNA-Seq from the relevant tissue is not represented, or poorly represented, within 300K-RNA data sources.”*

4. The 300K manuscript uses specific thresholds for each of the approaches for prioritization (Top-4, 0.05, ϵ -2), but makes no attempt to explain why the Top-4 approach is selected. Whilst there is presentation of sensitivity and PPV information in Figure 2E (which is assumed to be the 86 experimentally validated splice-altering variants – this isn’t declared in the figure gene), a formal analysis of optimal thresholds could be performed using the Precision-Recall or Receiver Operator Curve. The authors already declare that PPV isn’t their primary objective, and they place more emphasis on sensitivity (lines 236-239), so why isn’t a Top-7 approach used with 97% sensitivity and ~25% PPV (Figure 2E)? In addition to comment #2, consideration of optimal thresholds for tissue-shared Vs tissue-specific mis-splicing events may be informative, and an interesting analysis to add here.

Author’s Response

We present our critical evaluation of precision-recall curves that provide rationale for our selection of the Top-4* (Figure 2B, and Figure S1C,E and F). The challenge of increasing sensitivity via consideration of e.g. Top-7* events relates to complexity for diagnostic pathology workforces. Acknowledging that application of PVS1 currently involves uninformed guesswork, informed consideration of 5 events (IR and Top-4) is feasible – with consideration of every additional event creating further interpretative complexity.

5. The interpretative rules for SpliceAI require greater explanation in the text section of the methods. While figure 1 and explanation in the results section enables greater clarity, the methods are not clear as a standalone piece of text. Some explicit justifications for the use of 0.05 thresholds are also required.

Author’s Response

We had originally taken an agnostic approach analysing all SAI delta scores. We thought use of the author’s high sensitivity and high specificity thresholds may be more useful for our readership. However, we inherently agree with this comment and therefore present our precision-recall analyses of all returned SAI delta scores, excluding only those < 0.001 as effectively neutral impact (Figure 2B, and

Figure S1C,E and F). This was requisite, in order to create an interpretable number of Δ -scores for statistical and graphical analysis (2,836/1,720,000 of delta score returned). We bias sensitivity and specificity toward SAI by considering only the 86/88 variants SAI could assess, and removing all ‘uninterpretable’ delta scores. We have improved clarity in precisely how we adapted and evaluated SAI delta scores in Methods and Results text.

6. There are two fairly recent algorithms that the authors should also consider in their comparative analysis (CADD-Splice, <https://genomemedicine.biomedcentral.com/articles/10.1186/s13073-021-00835-9> & SQUIRLS, [https://www.cell.com/ajhg/pdfExtended/S0002-9297\(21\)00238-X](https://www.cell.com/ajhg/pdfExtended/S0002-9297(21)00238-X)).

Author’s Response

Please see “*Response to all Reviewers*” above for an explanation of our reasoning to remove reference to MMSplice within this manuscript that seeks to validate the diagnostic and predictive accuracy of the 300K-RNA Top 4 approach.

7. The authors note that over half of the variants included two or more mis-splicing events (line 87). This is an interesting observation which would benefit from further elaboration and discussion. Can the authors demonstrate the relative dosage of each aberrant junction to the canonical junction? How does the relative dosage of the mis-splicing events impact the accuracy of the algorithms to predict multiple mis-splicing events? E.g. Rowlands et al (<https://doi.org/10.1016/j.ajhg.2021.12.014>) describe a normalized read count applied during the Cummings et al pipeline (10.1126/scitranslmed.aal5209) to rank mis-splicing events. Recent updates to ACMG guidelines have suggested that partial impacts are an important aspect of variant interpretation (<https://doi.org/10.1101/2021.12.28.21267792>), and the authors have an opportunity to quantify these important observations in over 40 cases and to put this in greater context with the existing literature around variant interpretation.

Author’s Response

We agree that determination of partial mis-splicing from a hypomorphic allele is important for variant interpretation – with interpretive parallels to somatic mosaic variants in rare disorders and minor allele frequencies of ‘second hits’ in cancer predisposition genes associated with tumorigenesis.

In our experience (Bournazos et al., 2022)², quantitation of mis-spliced transcripts is not as clinically useful as the determining levels of residual canonical splicing. In our cohort of 88 variants evaluated in this manuscript; 39/88 variants resulted in mis-spliced transcripts targeted by NMD and a further 25/88 variants that produced a mix of in-frame and frameshift variants targeted by NMD. Furthermore, 20

variants were associated with autosomal recessive disease in a context where NMD is acting on the allele *in trans* due to nonsense/splicing variation.

With NMD acting with variable efficiency on transcripts encoding a premature termination codon, and no ability to comparatively measure rates of splicing reactions for canonical splicing versus variant-associated mis-splicing events – this is a very complex problem. Though an area of great interest to the authors, this issue lies outside the framework of this manuscript. To at least partly redress this important comment, we detail information of complete versus partial mis-splicing for 41 variants for which this evidence was obtainable in Table S1.

8. Figure 4 suggests use of PVS1 ACMG codes in the context of the presented evidence. Whilst this is a natural and relevant extension of the presented work, I would suggest that these recommendations are more relevant for incorporation through ongoing work in the ClinGen community. This will enable wider panels of experts to evaluate and comment on guidance before it is made available in literature, and to link these seamlessly with other updates to guidance on PVS1 for splicing variants, which can result in rapid uptake of guidance across diagnostic genomic centres: <https://clinicalgenome.org/working-groups/sequence-variant-interpretation/>.

Author's Response

We agree. Our Figure S5 draft guidelines have been renamed thus, further refined, and will be critically evaluated by: **a)** a multi-disciplinary group of > 150 genetic experts over 2022 (the Australasian Consortium for RNA Diagnostics, SpliceACORD), and **b)** in collaboration with the NIH Clinical Genome Resource (Cooper leads a “splicing variant sub-group” within her expert panels). However, until Australian and International clinical guidelines are devised and endorsed - it is important in the meanwhile to provide clinical workforces, who may wish to use the predictive, evidence-based insight garnered via SpliceVault and 300K-RNA, with a framework evaluated by expert users that guides potential use of this novel approach in a clinical setting. SpliceVault and 300K-RNA is transformative for interpretation of splicing variants. It does not yet mitigate all caveats, but it is vastly better than guesswork.

Reviewer #3:

1. The paper should provide estimates of how frequently essential splicing variants do not result in IR, PTC or frameshift. Unless this represents a substantial fraction of essential splicing variants, the need for a new guideline is not so impactful. It may well be large, but this number is lacking.

Author's Response

We agree. We collate experimentally determine outcomes from ES variants (and all variants) in three variant cohorts: 1) our 88 variants; 2) 61 variants studied in patient specimens curated from the literature; and 3) 63 variants studied through midi-gene assays (scrutinized to ensure technical design permitted detection of multi-exon skipping events). All variants, references and published splicing outcomes detected are detailed in Table S1. Figure S3E shows, 27/55 (49%) essential splice site variants across the three variant sets resulted in at least one in-frame event, 45/88 (51%) variants from our cohort resulted in at least one in-frame event.

2. Assuming this fraction is indeed substantial, the authors should then provide estimates as to how frequent it is that the proposed very strong evidence criteria does not apply (i.e. that not all 300 K- RNA top-4 result in IR, PTC or frameshift). Introducing the new guideline is relevant only if this fraction is substantial. Here too, the fraction may well be large, but this number is lacking.

Author's Response

This is another good point and the fraction *is* large @ 50% and now shown in Figure S3. Many challenges associated with application of PVS1 are mitigated by an accurate, evidence-based method to predict the probable nature of mis-splicing, to improve our ability to cater for the:

- ~50% splicing variants that induce more than one mis-splicing outcome;
- ~30% splicing variants that activate one or more cryptic splice-sites;
- ~50% splicing variants that induce one-or-more in-frame events.
-

We tried to align our draft guidelines in Figure S5 very closely with efforts made in ³ to provide greater flexibility in PVS1 to cater for the complexity of splicing variants, by including joint consideration of rationale underpinning null outcomes, and PM4 (change in protein length) - to cater for in-frame events or truncating variants predicted to evade NMD. Importantly, Figure S3 shows that our rubric will allow application of PVS1 for ~50% of ES variants (considering IR and Top-4*, max two in-frame events). We feel this provides a good balance, informed by evidence, not so conservative that the clinical utility of PVS1 is compromised by consideration of 5 events, though appropriately reduces level of confidence/evidence for variants likely to induce in-frame or truncating events. We suspect that once our Top-4* ranking method is disseminated, worldwide research efforts will innovate advances to improve predictive accuracy. Simple steps like deeper sequencing of human tissues, use of cycloheximide to protect events being degraded by NMD, improved methods to assess and quantify intron retention – are very likely to iteratively improve the predictive accuracy of this approach.

3. The analysis focuses on sensitivity alone using a benchmark dataset consisting of pathogenic variants only. Pathogenic variants are biased for variants resulting in loss-of-function splice isoforms. While sensitivity may matter most, the study needs to also assess a substantial set of splice variants that do not resolve in loss-function isoforms to justify the utility of the tool and demonstrate that those would be often appropriately discarded.

Author's Response

Our evidence refutes this assumption. Our ClinVar extraction from 2020 (n = 759,924 variants) shows that splicing variants account for 15% of LP/P variants, 22% of LB/B variants, and 7% of VUS. Many splicing variants are either non detected, or are detected and ignored or not reported due to complexity with interpretation.

Specifically, ~50% variants in our cohort, and ~50% variants in the two additional cohorts collated in this revision for additional validation, produce at least one transcript with an in-frame event or PTC predicted to evade NMD resulting in a truncated protein isoform. Further, in some disorders, 'loss-of-function' is clinically inferred for use of the PVS1 criterion only with in-frame consequences and probable expression of a dominant-negative mutant (e.g. filamentous proteins like actin, tubulin, myosin, collagen). For example the *MYH7* c.5655+4A>T variant causes in-frame exon skipping consistent with an autosomal dominant pathogenic mechanism in *MYH7* myopathy; or the *COL4A3* c.2881+1G>A variant consistent with an autosomal dominant pathogenetic mechanism in Alport Syndrome. Both *MYH7* and *COL4A3* have gnomAD pLI scores of 0 consistent with loss of function tolerance. Variants details and splicing outcomes are listed in Supplementary Table S1

4. The authors should discuss whether, according to their guidelines, very strong predictions would not need an RNA-seq or RT-PCR validation anyways. If so, then the added value is limited.

Author's Response

We agree this is very important. Over 2022, the authors are leading a consultative process akin to that described in ³ to: **1)** Clinically evaluate and refine the Draft Guidelines in Figure S5 and **2)** Devise a diagnostic algorithm for clinical application of PVS1, PM4, PP3 or BP4 criteria, OR, clinical recommendation of RNA Diagnostic testing. This very large body of consultative research falls outside the scope of this manuscript. One may further acknowledge that it is not appropriate to seek or obtain clinical endorsement of an unpublished methodology not yet subject to the rigour of peer review.

5. Fig 2A-C. There are striking mismatches between the reported numbers used to compute sensitivity and PPV, and the actual number of points displayed on the figure. An obvious example is the number "5" below cutoff for SpliceAi in Fig 2A. That is pervasive (are there 128 data points not seen in RNA studies and below cutoff for top-K RNA in Fig 2A?). There are probably reasons for this, e.g. what is plotted are splice-site levels scores, and what is counted are splicing events but this should be explained. I would suggest to show on the plot what is actually counted and to provide a clear rationale about how "things" are counted for each method. The same "philosophy" should be applied to each method.

Author's Response

Emended. Figure 2C-D shows all events seen in RNA studies: those with a SAI Δ -score > 0.001 are shown in red, occasional events seen in RNA studies with Δ -score < 0.001 are shown in black. Now false-negatives (FN) numbers annotated align with the number of dots.

6. Fig 2A is based on fixed cutoffs. Instead, precision-recall curves should be shown. This is a major concern because the authors have adjusted the cutoff of their method to increase sensitivity but did not investigate changes of cutoff of other methods. The authors argue that sensitivity matters and should make use of the precision-recall curve to compare the performance of all tools at same recall (i.e. sensitivity).

Author's Response

We agree and this point was raised by all reviewers. Comprehensive assessment of precision-recall curves for 300K-RNA and SAI are shown in Figure 2B, and S1B, E-F.

7. it is incorrect to state that "All approaches have limited or no ability to predict intron retention". The MMSplice software can be used to predict splicing efficiency which models intron retention. This should be tried.

Author's Response

We agree and removed this sentence from discussion. For simplicity and clarity, we focus intently upon additional validation of the sensitivity and positive predictive value of 300K-RNA Top-4* evidence-based approach. We elect to maintain comparison with deep learning Splice AI, as it can be adapted to predict both exon-skipping and cryptic splice-site activation.

8. It is excessive to state "Whereas SpliceAI predicted only a single mis-splicing event for 17/20 variants". It would be more appropriate to state that the algorithm the authors proposed on top of SpliceAI "predicted only a single mis-splicing event...". Technically, SpliceAI outputs scores for creation or loss of splice sites. One could propose several ways of predicting isoform outcomes from these scores. Here, the authors propose a greedy approach considering every site with score > 0.2 to be used. It is not expected that the authors over-engineer a method on top of SpliceAI. However, since they value sensitivity so high, they should use SpliceAI in a way that maximizes the number of isoform predictions. For instance, one could consider drawing independently sites using the Splice scores and obtaining thereby a distribution of isoforms from which one could retain the most probable ones. This would be more comparable to the "top-4" approach proposed by the authors.

Author's Response

See response to R2 Q5. We now employ an exclusion threshold of 0.001, which was essential to remove 99% of the >1.7 million scores returned scanning +/- 5000 nt of the 86/88 variants able to be assessed by Splice AI, in order to generate informative predictions and statistical metrics.

To clarify, our method enables SpliceAI to make multiple predictions via an analogous Top-4 approach. Splice AI often predicts activation of multiple cryptic splice-sites (Figure S1A).

We transparently declare the rules we apply are a customised interpretation of SpliceAI output.

9. 300K Top-4 relies on a genome annotation but no detail is given about this. The choice of the genome annotation could be critical. If the genome annotation of splice isoforms is too stringent, say, limited to a single major isoform per gene, then the most frequent non-annotated isoform (top-1) could often be a genuine, biologically functional splice isoform. At the other extreme, if the genome annotation is too inclusive then it can contain rare, non-functional splice isoforms, exactly those one would instead like to retain. On what basis did the authors decide which isoforms were "annotated"? Some assessment would be insightful.

Author's Response

This is an important point. Unfortunately, there is no national or international consensus of the transcript to be used for diagnostic reporting for each gene and condition – acknowledging different conditions can be associated with different transcripts arising from the same gene. Each diagnostic laboratory has their own reference list of transcript numbers they use for diagnostic reporting.

To cater for this complexity, 300K-RNA enables users to select the annotated transcript. All unannotated exon-skipping and cryptic-activation events, relative to this transcript, are returned in the Table.

We additionally specify within SpliceVault output when any of the top-ranked events are annotated events in alternate transcripts. It is important these annotated splicing events are encompassed within SpliceVault 300K-RNA output, so that clinical users can assess the likely pathogenicity of predominant or enhanced expression of an alternate annotated isoform.

Minor points

10. Line 50, the observation that aberrant splicing junctions are typically detected at weak levels in other samples was already reported by Kremer et al., Nat Commun, 2017, with a quantification of the prevalence of this phenomenon consistent with the much more extensive and recent

analyses by Dawes et al., 2021. Also, Kremer et al. also formulated the hypothesis of the present study described I. 64-66. This prior work should be cited in this respect.

Author's Response

This insight has been added and cited within our introduction. See lines 50-55.

11. Figure 2 is entitled "Accuracy" but what is assessed is sensitivity.

Author's Response

Emended.

12. Designing targeted RT-PCR follow-ups guided by the predictions have moderate value as they require specific primer designs and are blind to not-predicted isoforms. Would not the authors suggest RNA-seq, potentially with cycloheximide, in any case? This should be discussed.

Author's Response

Cycloheximide treatment was performed for all re-analysed variants, however no additional isoforms were detected. Mis-spliced *GSDME* and *OPHN1* transcripts were not compliant with NMD and *OPHN1* expression was too low in blood for RNA-seq analysis for agnostic interrogation. RNA sequencing was performed for the *SPG11* c.2317-13C>G variant, with expression levels insufficient to detect the double exon-skipping identified by RT-PCR (Bournazos et al., 2022)². For *EMD*, patient fibroblast-derived RNA is currently being used to compare long-read transcriptomics versus targeted sequencing of RT-PCR amplicons, plus and minus cycloheximide, and is part of a larger clinical evaluation of the diagnostic utility of long-read sequencing for RNA and DNA diagnostics.

1. Wilks, C. *et al.* recount3: summaries and queries for large-scale RNA-seq expression and splicing.

bioRxiv 2021.05.21.445138 (2021) doi:10.1101/2021.05.21.445138.

2. Bournazos, A. M. *et al.* Standardized practices for RNA diagnostics using clinically accessible

specimens reclassifies 75% of putative splicing variants. *Genetics in Medicine* (2021)

doi:10.1016/j.gim.2021.09.001.

3. Abou Tayoun, A. N. *et al.* Recommendations for interpreting the loss of function PVS1 ACMG/AMP

variant criterion. *Hum Mutat* **39**, 1517–1524 (2018).

Decision Letter, Appeal

19th Apr 2022

Dear Dr. Cooper,

Thank you for your message of 19th Apr 2022, asking us to reconsider our decision on your manuscript "SpliceVault: predicting the precise nature of variant-associated mis-splicing.". I have now discussed the points of your letter with my colleagues, and while we are not sure that the reviewers will be fully satisfied with these efforts, we are willing to send the revised manuscript back to reviewers for comment.

When preparing a revision, please ensure that it fully complies with our editorial requirements for format and style; details can be found in the Guide to Authors on our website (<http://www.nature.com/ng/>).

Please be sure that your manuscript is accompanied by a separate letter detailing the changes you have made and your response to the points raised. At this stage we will need you to upload:

1) a copy of the manuscript in MS Word .docx format.

**2) The Editorial Policy Checklist:

<https://www.nature.com/documents/nr-editorial-policy-checklist.pdf>
[THIS CURRENTLY IS MISSING]

**3) The Reporting Summary:

<https://www.nature.com/documents/nr-reporting-summary.pdf>
(Here you can read about the role of the Reporting Summary in reproducible science:
<https://www.nature.com/news/announcement-towards-greater-reproducibility-for-life-sciences-research-in-nature-1.22062>)
[THIS CURRENTLY IS MISSING]

Please use the link below to be taken directly to the site and view and revise your manuscript:

[redacted]

With kind wishes,

Catherine

Catherine Potenski, PhD
Chief Editor
Nature Genetics
1 NY Plaza, 47th Fl.

New York, NY 10004
catherine.potenski@us.nature.com
<https://orcid.org/0000-0002-4843-7071>

Author Rebuttal to Initial comments

Reviewer Comments:

Author's Response to all reviewers:

All reviewers acknowledged the clinical importance and our discovery and offered several constructive suggestions how we could improve evidence supporting the rigour and robustness of the 300K-RNA Top 4 method. All three reviewers unanimously highlighted; 1) precision-recall analyses, 2) evidence for concordance among the ranking of unannotated splicing events between tissues and sources, and 3) additional validation and identification of sources of bias; as suggested areas for improvement.

Thus, we now include additional Figures and Supplemental Figures:

9. Precision-recall analyses
10. Updated SpliceVault and 300K-RNA source data to utilise RNA-Seq splice-junction data processed using the unified Monorail pipeline.
11. Updated the SpliceVault portal to enable users the flexibility to access ranked, unannotated splicing events *within* different tissue specimen subtypes within GTEx.
12. Further validation of the Top-4 method using two additional, independent cohorts of clinical variants curated from literature (61 variants studied in patient specimens and 63 variants studied through midi-gene assays).
13. Concordance analyses of the ranking of unannotated splicing events for the 88 splice-sites affected by our cohort of 88 splice-altering clinical variants, and for all unannotated splicing events detected for 98,810 annotated splice-sites across 3,668 clinically relevant Mendelian disease genes (the Mendeliome) – between different biospecimen subtypes within GTEx, and, between GTEx and SRA datasets.
14. For 19/88 variants associated with muscle disorders and where RNA testing was performed using muscle, we showed that GTEx Muscle Top-4* (n = 881 muscle samples) results in lower sensitivity and PPV than using 300K-RNA Top-4* (though discuss that higher sequencing depth of human tissues may ultimately result in improved accuracy).
15. Conducted deep analysis of features that help to discriminate 'true-positive' from 'false-positive' events to inform prospective improvement of the ranking method.
16. Quantified that ~50% of essential splice site variants are associated with one or more in-frame event (across all three validation cohorts) – confirming the clinical importance of using SpliceVault to accurately consider the probable nature of variant-associated mis-splicing for informed use of PVS1, PM4 and PP3 criteria of the ACMG-AMP guidelines.

With a large body of additional data, though we conducted side-by-side comparative analyses with MMSplice and SPANR as recommended, it became very cumbersome to include and discuss comparisons for multiple algorithms. As an additional note to R3, CADD-Splice does not use an independent algorithmic approach, rather, it integrates SpliceAI and MMSplice along with other features.

As this manuscript that seeks to validate the diagnostic utility and predictive accuracy of the 300K-RNA Top-4 approach, we remove all comparative reference to MMSplice and SPANR. We maintain comparative evaluation between 300K-RNA and Splice AI, due to their mutual ability to predict both cryptic activation and exon skipping to enable precision-recall analyses. We also feel the comparison of an evidence-based approach with a sophisticated, deep learning approach, is of broad general interest to the Nature Genetics readership. Our point-by-point rebuttal to your comments follows below. We are very appreciative of the constructive suggestions offered, which have improved the body of evidence supporting the rigour and accuracy of our Top-4* method.

Reviewer #1:

13. Some sentence in the abstract is very unfriendly to readers without background knowledge. For instance, Line 16-17, 'In comparison, applying interpretative rules to SpliceAI D-scores correctly identifies 55% of ...'. It will make people confused about what's the SpliceAI and what's the delta-scores? In addition, the authors did comparisons between 300K-RNA Top4 and SpliceAI and MMSplice. It would be rather weird to only mention SpliceAI in the abstract but did not mention MMSplice at all.

Author's Response

We have edited the abstract to be friendlier to readers not familiar with the terminology of SpliceAI scores, simply saying "300K-RNA showed higher sensitivity and positive predictive value than SpliceAI in predicting exon-skipping and cryptic-activation events".

Please see "*Response to all Reviewers*" above for an explanation of our reasoning to remove reference to MMSplice within this manuscript.

14. The author analyzed 86 variants across 72 genes to demonstrate the capacity of 300K-RNA Top4 for the identification of variant-induced Mis-splicing events. They should give a clear explanation about how they selected these variants. Do they include all current experimentally confirmed, splicing-related genetic variants? The author should provide a table to show details of these variants and genes. In addition, the author classified these variants into different categories (Fig. 1A), what's the purpose of this analysis? Is there any expectation that variants from different categories show different validation rates by 300K-RNA Top 4?

Author's Response

Ascertainment criteria are clearly defined in the first paragraph of results (see manuscript text lines 87-90). We add Table S1 that details all variants confirmed to disrupt splicing and the detected splicing

outcomes. The purpose of Figure 1A is to show that our cohort of 88 variants encompasses a wide range of disorders with different tissue-specific or multi-system involvement.

15. The largest limitation of 300K-RNA Top4 is that the positive predictive value (PPV) is very low compared to other methods, which means FDR is very high in the other words. For instance, the FDR of exon skipping prediction by 300K-RNA is 67%. The author also claimed that ‘300K-RNA Top4 outperforms SpliceAI (and MMSplice)’ in Line 232. Based on Fig. 2E, under similar PPV/FDR, for instance (69% for 300K-RNA, 65% for SpliceAI), sensitivity/recall of two methods (50% and 59% respectively) are very close and 300K-RNA performs even slightly worse than SpliceAI. A fair comparison should use the receiver operating characteristic (ROC) curve or precision-recall curve to calculate the area under the curve (AUC).

Author’s Response

We agree. New Figure 2B shows precision-recall curves for 300K-RNA and SpliceAI at Δ -score thresholds that offer the same, total number of predictions as offered by 300K-RNA at each rank cut-off – such that true/false positive rates can be compared. This demonstrates 300K-RNA has higher sensitivity and comparable positive predictive value at every cut-off except 300K-RNA top-1, where the scores are similar. Further analyses of precision and recall are provided in Figures 2C, 2D, S1C, S1E, S1F, S4E.

16. As the precision of 300K-RNA Top 4 is relatively low, even only using the Top 1 splicing events, 40% of the prediction could be false positives, the author should provide a deeper investigation regarding what’s the difference between those ‘Seen in RNA studies’ and those not. For instance, do those skipped exons Seen in RNA studies are shorter than those not? Do these events have relative stronger/weaker splicing strength? Are they more GC/AT enriched et al?

Author’s Response

We conducted evaluation of Top-4* true/false positives in new Figures S1B and S2A-D and manuscript commentary in Lines 164 – 168 and 175 – 184. RNA re-analyses detecting Top-4 events suggest precision may be a little higher than we currently report. We determined in Figure S2 that true positives tend to be associated with:

- Higher sample count i.e. seeing an event in more samples. We don’t wish to over emphasize this point, given so many variant-activated events are only seen in a few samples.
- Double-exon skipping is substantially less likely to occur than single exon skipping.
- Double versus single exon skipping is not related to the overall length (in nt) of the excised region (or exon length, not shown).
- We do not compare or present analyses of splice-site strength or use of GC/AT enrichment among unannotated splicing events as this was investigated in detail in Dawes et al. 2022 NCOMMS21-28137 (proofs returned to NCOMMS. Online publication imminent).

17. The authors showed the performance on Mis-splicing prediction with three methods, respectively. What’s the overlap between each other? What’re the characteristics of the events that could be identified by all three methods or could only be identified by each of them?

Author's Response

Please see “*Response to all Reviewers*” above for an explanation of our reasoning to remove reference to additional algorithms. We agree this is an interesting question and feel comparative evaluation of multiple contemporary splicing prediction tools warrants investigation as an entire publication in its own right. Our focus here is the diagnostic utility and predictive accuracy of the 300K-RNA Top-4 evidence-based approach.

18. In Line 153, the author claimed that intron retention cannot be predicted by 300K-RNA. It's not correct as there are no significant differences between the detection of intron retention and other types of splicing events from RNA-seq data. Theoretically, detection of reads spanning across the exon-intron boundary, if it's not easier, it's quite similar to the detection of splicing junctions spanning two exons. Almost all well-known computational tools for splicing analysis based on RNA-seq data have the capacity of intron retention identification, such as MISO (Katz, Yarden, et al. 2010), rMATS (Shen, Shihao, et al. 2014), and PSI-Sigma (Lin, Kuan-Ting, and Adrian R. Krainer. 2019). Moreover, some tools specifically for the detection of intron retention have also been developed, for instance, IRFinder (Wong, Justin J-L., et al. 2013). As 20% of the Mis-splicing events induced by the 86 variants are intron retention, which is not a negligible proportion. The author should characterize the intron retention events, at least for these 29 events (Line 89), in RNA-seq samples with a relatively large sample size to get robust Top-ranked events.

Author's Response

300K-RNA is a database of detected splice-junctions from split-reads in RNA-seq data. Intron retention is inferred by non-split reads spanning the exon-intron boundary; absence of a splice-junction cannot be calculated from the 300K-RNA data sources. Quantification of intron retention is confounded by the presence of non-split reads corresponding to pre-mRNA and splicing intermediates. We suspect intron retention quantification may differ between RNA-Seq data derived using ribosomal depleted total RNA versus polyA depletion, because:

- a) transcripts with retained introns can be non-adenylated and retained in the nucleus
- b) introns can be long. Transcripts with retained introns can be particularly impacted by 3' bias associated with PolyA enrichment protocols.

Thus, herein we recommend pathology consideration of intron retention in addition to Top-4. But we do not advise or recommend how it may be used in a Top-4 style ranking method to predict the nature of probable mis-splicing.

19. The authors analyzed 335,301 RNA-seq samples from GTEx and SRA. They should clarify if any batch effect correction should be considered? They should also include an ablation study to show the robustness of those top-ranked splicing events. For instance, using 5K, 10K, 50K, 100K, and 300K samples, what's the overlap of the top identified unannotated splicing events.

Especially, for the 147 variants induced Mis-splicing events, what's the PPV and sensitivity if only using 10K, 50K, 100K samples?

Author's Response

We adapted SpliceVault to source data processed using the monorail approach of Wilks et al.¹ Inherent robustness and reproducibility of the 300K-RNA Top-4* method is shown by highly similar statistical metrics using both sets of source data. We further conducted correlative analyses of the ranking of unannotated splicing events around: i) the 88 splice sites impacted by our cohort of variants (Figure 3A-B) and ii) 98,810 annotated splice-sites in 3,668 clinically relevant Mendelian disease genes (the Mendeliome) (Figure S4A-B), comparing:

- 3) biospecimen subtypes within GTEx
- 4) GTEx and SRA datasets

including overall correlation of sample counts for each unannotated splicing event.

- We confirm the number of mis-splicing events detected in each individual tissue in GTEx is correlated with the number of samples present for that tissue (Figure S4D).
- We show the overlap of the top-4 300K-RNA events with the top-4 events present in four clinically accessible tissues in GTEx (Figure 3D)
- We show that using GTEx muscle Top-4 shows lower sensitivity than using 300K-RNA Top-4 for 19 variants associated with muscle disorders where RNA testing was conducted using muscle RNA (Figure S4E).

However, our overall opinion is the best predictive accuracy of a ranking approach may be garnered via a balance of breadth and depth i.e. high sequencing read depth of a reasonable number of specimens (e.g. hundreds but not thousands). We are keenly investigating this, as well as potential benefits of using human primary cell lines treated with cycloheximide to inhibit NMD, which may influence event ranking.

20. As the key finding of this study, is that 300K-RNA Top 4 could accurately predict 90% of the variants-associated Mis-splicing events. Only using a single dataset (147 events) might be not sufficient to prove its true sensitivity. The author should either apply the analysis on another independent dataset (if it's possible) or experimentally validate the predictions beyond the 86 variants. For instance, they can use CRISPR to mutate several splice sites and check whether the mutation-induced splicing variants indeed could be identified by 300K-RNA Top4.

Author's Response

We agree and conducted validation of the sensitivity and positive predictive value of 300K-RNA Top-4* predictions on two additional and independent datasets: 121 variants from published patient-derived RNA (n=58) and midi-gene (n=63) studies are also now included in Table S1. We scrutinised the technical design of the midi-genes, to ensure multi-exon skipping and distal cryptics, if activated, could be detected. Importantly, we obtained a similar average sensitivity of 92% and average PPV of 29%.

Minor Points

21. Line 80, the authors mentioned that multiple experimental approaches have been performed to characterize aberrant splicing events arising from 86 genetic variants. It would be better to make a table to show how many of them were confirmed by each approach, including RT-PCR, RNA-seq, and/or minigene assay.

Author's Response

We provide Table S1 that details all variants evaluated in this study, the experimental approaches used, RNA source and splicing outcomes.

22. Line 84, '77 SNVs were included in the dataset' is contradict to 78 SNVs (59 + 19) shown in Figure 1B. The author should clarify which number is correct. In addition, it'd be better to also show the relative position of "2 insertions, 5 deletions, and 2 deletion-insertion variants" to the splice sites in the figure.

Author's Response

Figure 1B has been corrected and now represents an updated list of experimentally-verified variants, details provided in Table S1.

23. Line 223, the authors claimed that 'no previous method can predict the nature of variant-induced mis-splicing'. However, both SpliceAI and MMSplice are public methods. Did the authors want to conclude that the prediction from SpliceAI and MMSplice is not 'the nature of variant-induced mis-splicing'?

Author's Response

We have removed this sentence and reference to MMSplice.

24. Since the authors performed RNA reanalysis for 4 cases, it would be interesting to characterize full-length of those mis-spliced transcripts using the third-generation sequencing technology. The results could be more convincing compared to bands in the gel which only amplified parts of the mis-spliced transcripts.

Author's Response

We can only redress this comment by advising that we agree entirely - and are currently conducting a collaborative (via SpliceACORD) clinical evaluation trial of PCR followed by long-read sequencing of amplicons to see if it: 1) improves sequencing clarity of multi-template PCRs, which are inherent in a subject with a heterozygous splice-altering variant, and common for homozygous or hemizygous variants that elicit more than one mis-splicing outcome; 2) presents cost-savings over Sanger sequencing.

Reviewer #2:

9. The data utilized to generate the 300K resource will be diverse, with a number of different tissues/cell-types/biosamples used and created through different wet laboratory and bioinformatics pipelines. The authors state that the 'splice-junction read counts were summarised' using a program called Datamash. I am not familiar with Datamash but believe it to be a simple command-line unix system programme capable of generating summary statistics. This is the extent of the description of data curation, but this is the most important part of the manuscript in my opinion. There is no explanation of the data curation and/or attempts to harmonize raw datasets through a single uniformed bioinformatics pipeline. Drawing analogies with human genomic resources, e.g. gnomAD (<https://gnomad.broadinstitute.org/>), the power of these databases lies to some extent in the unified data processing and analysis approaches applied to the data and to thereby identify and filter for artefactual findings. Such approaches for data harmonization will make the 300K resource more powerful, and whilst practically difficult, is to some extent why such a resource is yet to be realised. My suggestions are two-fold: (1) a summary of the different biosamples and wet laboratory approaches used in the analysis should be added; and (2) harmonization of the data through a single pipeline, e.g. the GTEx RNA-seq pipeline (<https://github.com/broadinstitute/gtex-pipeline>) or a harmonised use of algorithms to statistically identify mis-spliced events from RNA-seq, e.g. FRASER (<https://www.nature.com/articles/s41467-020-20573-7>) or DROP (<https://www.nature.com/articles/s41596-020-00462-5>). The authors should also expand their description of mis-splicing events to include quality metrics and appropriate thresholds (e.g. read depth, quality and mapping thresholds).

Author's Response

We appreciate this constructive suggestion and contemporised SpliceVault to source data processed identically using the monorail processing pipeline¹. In line with suggestions by R1 and R2, we provide Table S1 that details all variants, the wet-lab approaches and the source of the RNA. We have also provided improved clarity in Methods and Results regarding specific procedural steps to ascertain variants, rank the events etc. See response to R1 Q7 regarding our extensive validation of method sensitivity and positive predictive value, which is not substantially influenced by RNA-Seq method or sample sub-type for exon-skipping and cryptic splice site junctional reads. We do feel there will be influence of technical pipeline for quantification of intron retention – underpinning our decision to not comment at this early stage on use of IR non-split reads in our Top-4 ranking method for clinical use.

10. In addition to comment #1, there is no attempt to sub-divide mis-splicing events in the 300K resource by biosample sources. Such an analysis could lead to interesting and important observations in the data, and contribute to how to expand the resource going forward, e.g.
 - a. does the use of tissue-shared Vs tissue-specific mis-splicing events impact the accuracy of the Top-4 approach?

- b. Do certain biosamples exhibit higher levels of mis-splicing than others?
- c. Is the expression of isoform X in tissue A associated with higher levels of mis-splicing than in isoform Y in tissue B?

Whilst the composite resource presented here provides power by numbers, an analysis enabling the calculation of mis-splicing events by biosample type may enable further improvements to the algorithm and I think this is an area the authors should explore.

11. In addition, the authors remark that their RNA analyses performed in blood and fibroblasts were informed by the diverse biosample resource, can the authors also comment on the suitability of their investigated genes in clinically accessible tissues? e.g. through isoform similarity evaluated through MAJIQ-CAT? (<https://pubmed.ncbi.nlm.nih.gov/32225167/>) can the authors comment on the utility of the 300K resource for genes that aren't appropriate for surveillance from clinically accessible tissues?

Author's Response

We have increased the functionality of the SpliceVault portal to enable users customized access to “all of 300K-RNA” or any one of the 56 specific specimen sub-types within the GTEx data source.

a) Please see our response to R1 Q7, with respect to our comprehensive evaluation of concordance in ranked unannotated splice events between GTEx tissues, between GTEx and SRA, for both our cohort of 88 splice-sites and across the Mendeliome. We showed the same concordance across every annotated human splice site, though felt with our clinical focus the Mendeliome was more relevant.

Figure S4 provides precision-recall analyses showing 300K-RNA Top-4* outperforms GTEx Skeletal muscle Top-4* versus for 19 variants subject to RNA testing of muscle. However, we feel that tissue-specific SpliceVault databases that encompasses deeper sequencing of tissue specimens from individuals of different ages could improve predictive accuracy. We are currently pursuing this hypothesis for blood, fibroblasts and muscle that we are studying in our clinical program of RNA Diagnostics (typically conducting ~150 million read depth +/- cycloheximide to inhibit NMD for primary cells, as compared to 20 million read depth for GTEx specimens).

b) The number of mis-splicing events detected correlates strongly with the number of samples present in GTEx for each biosample type (Figure S4D). Testis are a notable outlier – well known to have ‘leaky’ higher expression of many genes that show tissue-restricted expression.

c) Unfortunately, this important point requires long-read RNA sequencing from a clinical cohort of affected individuals with splice-altering variants, compared to controls or disease controls. This is a current area of our investigation, though falls outside the scope of this manuscript.

Q3) For RNA diagnostic testing performed in our laboratory, review of alternative splicing patterns in the target gene was performed using GTEx RNA-seq data derived from clinically accessible tissues and manifesting tissues to establish that testing in clinically accessible tissues was appropriate (see

Bournazos et al., 2022)². In regards to the suitability of 300K-RNA resource for genes not expressed in clinically relevant tissues, See lines 216-220: *“Therefore, we recommend use of 300K-RNA Top-4* as prediction of the probable nature of variant associated mis-splicing until we have great enough breadth and/or depth of RNA-Seq data to evaluate a tissue-specific approach. We advise caution for genes with known tissue-specific or developmental alternative splicing, where RNA-Seq from the relevant tissue is not represented, or poorly represented, within 300K-RNA data sources.”*

12. The 300K manuscript uses specific thresholds for each of the approaches for prioritization (Top-4, 0.05, ϵ -2), but makes no attempt to explain why the Top-4 approach is selected. Whilst there is presentation of sensitivity and PPV information in Figure 2E (which is assumed to be the 86 experimentally validated splice-altering variants – this isn’t declared in the figure gene), a formal analysis of optimal thresholds could be performed using the Precision-Recall or Receiver Operator Curve. The authors already declare that PPV isn’t their primary objective, and they place more emphasis on sensitivity (lines 236-239), so why isn’t a Top-7 approach used with 97% sensitivity and ~25% PPV (Figure 2E)? In addition to comment #2, consideration of optimal thresholds for tissue-shared Vs tissue-specific mis-splicing events may be informative, and an interesting analysis to add here.

Author’s Response

We present our critical evaluation of precision-recall curves that provide rationale for our selection of the Top-4* (Figure 2B, and Figure S1C,E and F). The challenge of increasing sensitivity via consideration of e.g. Top-7* events relates to complexity for diagnostic pathology workforces. Acknowledging that application of PVS1 currently involves uninformed guesswork, informed consideration of 5 events (IR and Top-4) is feasible – with consideration of every additional event creating further interpretative complexity.

13. The interpretative rules for SpliceAI require greater explanation in the text section of the methods. While figure 1 and explanation in the results section enables greater clarity, the methods are not clear as a standalone piece of text. Some explicit justifications for the use of 0.05 thresholds are also required.

Author’s Response

We had originally taken an agnostic approach analysing all SAI delta scores. We thought use of the author’s high sensitivity and high specificity thresholds may be more useful for our readership. However, we inherently agree with this comment and therefore present our precision-recall analyses of all returned SAI delta scores, excluding only those < 0.001 as effectively neutral impact (Figure 2B, and Figure S1C,E and F). This was requisite, in order to create an interpretable number of Δ -scores for statistical and graphical analysis (2,836/1,720,000 of delta score returned). We bias sensitivity and specificity toward SAI by considering only the 86/88 variants SAI could assess, and removing all ‘uninterpretable’ delta scores. We have improved clarity in precisely how we adapted and evaluated SAI delta scores in Methods and Results text.

14. There are two fairly recent algorithms that the authors should also consider in their comparative analysis (CADD-Splice, <https://genomemedicine.biomedcentral.com/articles/10.1186/s13073-021-00835-9> & SQUIRLS, [https://www.cell.com/ajhg/pdfExtended/S0002-9297\(21\)00238-X](https://www.cell.com/ajhg/pdfExtended/S0002-9297(21)00238-X)).

Author's Response

Please see “*Response to all Reviewers*” above for an explanation of our reasoning to remove reference to MMSplice within this manuscript that seeks to validate the diagnostic and predictive accuracy of the 300K-RNA Top 4 approach.

15. The authors note that over half of the variants included two or more mis-splicing events (line 87). This is an interesting observation which would benefit from further elaboration and discussion. Can the authors demonstrate the relative dosage of each aberrant junction to the canonical junction? How does the relative dosage of the mis-splicing events impact the accuracy of the algorithms to predict multiple mis-splicing events? E.g. Rowlands et al (<https://doi.org/10.1016/j.ajhg.2021.12.014>) describe a normalized read count applied during the Cummings et al pipeline (10.1126/scitranslmed.aal5209) to rank mis-splicing events. Recent updates to ACMG guidelines have suggested that partial impacts are an important aspect of variant interpretation (<https://doi.org/10.1101/2021.12.28.21267792>), and the authors have an opportunity to quantify these important observations in over 40 cases and to put this in greater context with the existing literature around variant interpretation.

Author's Response

We agree that determination of partial mis-splicing from a hypomorphic allele is important for variant interpretation – with interpretive parallels to somatic mosaic variants in rare disorders and minor allele frequencies of ‘second hits’ in cancer predisposition genes associated with tumorigenesis.

In our experience (Bournazos et al., 2022)², quantitation of mis-spliced transcripts is not as clinically useful as the determining levels of residual canonical splicing. In our cohort of 88 variants evaluated in this manuscript; 39/88 variants resulted in mis-spliced transcripts targeted by NMD and a further 25/88 variants that produced a mix of in-frame and frameshift variants targeted by NMD. Furthermore, 20 variants were associated with autosomal recessive disease in a context where NMD is acting on the allele *in trans* due to nonsense/splicing variation.

With NMD acting with variable efficiency on transcripts encoding a premature termination codon, and no ability to comparatively measure rates of splicing reactions for canonical splicing versus variant-associated mis-splicing events – this is a very complex problem. Though an area of great interest to the authors, this issue lies outside the framework of this manuscript. To at least partly redress this important comment, we detail information of complete versus partial mis-splicing for 41 variants for which this evidence was obtainable in Table S1.

16. Figure 4 suggests use of PVS1 ACMG codes in the context of the presented evidence. Whilst this is a natural and relevant extension of the presented work, I would suggest that these recommendations are more relevant for incorporation through ongoing work in the ClinGen community. This will enable wider panels of experts to evaluate and comment on guidance before it is made available in literature, and to link these seamlessly with other updates to guidance on PVS1 for splicing variants, which can result in rapid uptake of guidance across diagnostic genomic centres: <https://clinicalgenome.org/working-groups/sequence-variant-interpretation/>.

Author's Response

We agree. Our Figure S5 draft guidelines have been renamed thus, further refined, and will be critically evaluated by: **a)** a multi-disciplinary group of > 150 genetic experts over 2022 (the Australasian Consortium for RNA Diagnostics, SpliceACORD), and **b)** in collaboration with the NIH Clinical Genome Resource (Cooper leads a "splicing variant sub-group" within her expert panels). However, until Australian and International clinical guidelines are devised and endorsed - it is important in the meanwhile to provide clinical workforces, who may wish to use the predictive, evidence-based insight garnered via SpliceVault and 300K-RNA, with a framework evaluated by expert users that guides potential use of this novel approach in a clinical setting. SpliceVault and 300K-RNA is transformative for interpretation of splicing variants. It does not yet mitigate all caveats, but it is vastly better than guesswork.

Reviewer #3:

13. The paper should provide estimates of how frequently essential splicing variants do not result in IR, PTC or frameshift. Unless this represents a substantial fraction of essential splicing variants, the need for a new guideline is not so impactful. It may well be large, but this number is lacking.

Author's Response

We agree. We collate experimentally determine outcomes from ES variants (and all variants) in three variant cohorts: 1) our 88 variants; 2) 61 variants studied in patient specimens curated from the literature; and 3) 63 variants studied through midi-gene assays (scrutinized to ensure technical design permitted detection of multi-exon skipping events). All variants, references and published splicing outcomes detected are detailed in Table S1. Figure S3E shows, 27/55 (49%) essential splice site variants across the three variant sets resulted in at least one in-frame event, 45/88 (51%) variants from our cohort resulted in at least one in-frame event.

14. Assuming this fraction is indeed substantial, the authors should then provide estimates as to how frequent it is that the proposed very strong evidence criteria does not apply (i.e. that not all 300 K- RNA top-4 result in IR, PTC or frameshift). Introducing the new guideline is relevant only if this fraction is substantial. Here too, the fraction may well be large, but this number is lacking.

Author's Response

This is another good point and the fraction *is* large @ 50% and now shown in Figure S3. Many challenges associated with application of PVS1 are mitigated by an accurate, evidence-based method to predict the probable nature of mis-splicing, to improve our ability to cater for the:

- ~50% splicing variants that induce more than one mis-splicing outcome;
- ~30% splicing variants that activate one or more cryptic splice-sites;
- ~50% splicing variants that induce one-or-more in-frame events.

We tried to align our draft guidelines in Figure S5 very closely with efforts made in ³ to provide greater flexibility in PVS1 to cater for the complexity of splicing variants, by including joint consideration of rationale underpinning null outcomes, and PM4 (change in protein length) - to cater for in-frame events or truncating variants predicted to evade NMD. Importantly, Figure S3 shows that our rubric will allow application of PVS1 for ~50% of ES variants (considering IR and Top-4*, max two in-frame events). We feel this provides a good balance, informed by evidence, not so conservative that the clinical utility of PVS1 is compromised by consideration of 5 events, though appropriately reduces level of confidence/evidence for variants likely to induce in-frame or truncating events. We suspect that once our Top-4* ranking method is disseminated, worldwide research efforts will innovate advances to improve predictive accuracy. Simple steps like deeper sequencing of human tissues, use of cycloheximide to protect events being degraded by NMD, improved methods to assess and quantify intron retention – are very likely to iteratively improve the predictive accuracy of this approach.

15. The analysis focuses on sensitivity alone using a benchmark dataset consisting of pathogenic variants only. Pathogenic variants are biased for variants resulting in loss-of-function splice isoforms. While sensitivity may matter most, the study needs to also assess a substantial set of splice variants that do not resolve in loss-function isoforms to justify the utility of the tool and demonstrate that those would be often appropriately discarded.

Author's Response

Our evidence refutes this assumption. Our ClinVar extraction from 2020 (n = 759,924 variants) shows that splicing variants account for 15% of LP/P variants, 22% of LB/B variants, and 7% of VUS. Many splicing variants are either non detected, or are detected and ignored or not reported due to complexity with interpretation.

Specifically, ~50% variants in our cohort, and ~50% variants in the two additional cohorts collated in this revision for additional validation, produce at least one transcript with an in-frame event or PTC predicted to evade NMD resulting in a truncated protein isoform. Further, in some disorders, 'loss-of-function' is clinically inferred for use of the PVS1 criterion only with in-frame consequences and probable expression of a dominant-negative mutant (e.g. filamentous proteins like actin, tubulin, myosin, collagen). For example the MYH7 c.5655+4A>T variant causes in-frame exon skipping consistent with an autosomal dominant pathogenic mechanism in MYH7 myopathy; or the COL4A3 c.2881+1G>A variant consistent with an autosomal dominant pathogenic mechanism in Alport Syndrome. Both

MYH7 and *COL4A3* have gnomAD pLI scores of 0 consistent with loss of function tolerance. Variants details and splicing outcomes are listed in Supplementary Table S1.

16. The authors should discuss whether, according to their guidelines, very strong predictions would not need an RNA-seq or RT-PCR validation anyways. If so, then the added value is limited.

Author's Response

We agree this is very important. Over 2022, the authors are leading a consultative process akin to that described in ³ to: **1)** Clinically evaluate and refine the Draft Guidelines in Figure S5 and **2)** Devise a diagnostic algorithm for clinical application of PVS1, PM4, PP3 or BP4 criteria, OR, clinical recommendation of RNA Diagnostic testing. This very large body of consultative research falls outside the scope of this manuscript. One may further acknowledge that it is not appropriate to seek or obtain clinical endorsement of an unpublished methodology not yet subject to the rigour of peer review.

17. Fig 2A-C. There are striking mismatches between the reported numbers used to compute sensitivity and PPV, and the actual number of points displayed on the figure. An obvious example is the number "5" below cutoff for SpliceAi in Fig 2A. That is pervasive (are there 128 data points not seen in RNA studies and below cutoff for top-K RNA in Fig 2A?). There are probably reasons for this, e.g. what is plotted are splice-site levels scores, and what is counted are splicing events but this should be explained. I would suggest to show on the plot what is actually counted and to provide a clear rationale about how "things" are counted for each method. The same "philosophy" should be applied to each method.

Author's Response

Emended. Figure 2C-D shows all events seen in RNA studies: those with a SAI Δ -score > 0.001 are shown in red, occasional events seen in RNA studies with Δ -score < 0.001 are shown in black. Now false-negatives (FN) numbers annotated align with the number of dots.

18. Fig 2A is based on fixed cutoffs. Instead, precision-recall curves should be shown. This is a major concern because the authors have adjusted the cutoff of their method to increase sensitivity but did not investigate changes of cutoff of other methods. The authors argue that sensitivity matters and should make use of the precision-recall curve to compare the performance of all tools at same recall (i.e. sensitivity).

Author's Response

We agree and this point was raised by all reviewers. Comprehensive assessment of precision-recall curves for 300K-RNA and SAI are shown in Figure 2B, and S1B, E-F.

19. it is incorrect to state that "All approaches have limited or no ability to predict intron retention". The MMSplice software can be used to predict splicing efficiency which models intron retention. This should be tried.

Author's Response

We agree and removed this sentence from discussion. For simplicity and clarity, we focus intently upon additional validation of the sensitivity and positive predictive value of 300K-RNA Top-4* evidence-based approach. We elect to maintain comparison with deep learning Splice AI, as it can be adapted to predict both exon-skipping and cryptic splice-site activation.

20. It is excessive to state “Whereas SpliceAI predicted only a single mis-splicing event for 17/20 variants”. It would be more appropriate to state that the algorithm the authors proposed on top of SpliceAI “predicted only a single mis-splicing event....”. Technically, SpliceAI outputs scores for creation or loss of splice sites. One could propose several ways of predicting isoform outcomes from these scores. Here, the authors propose a greedy approach considering every site with score>0.2 to be used. It is not expected that the authors over-engineer a method on top of SpliceAI. However, since they value sensitivity so high, they should use SpliceAI in a way that maximizes the number of isoform predictions. For instance, one could consider drawing independently sites using the Splice scores and obtaining thereby a distribution of isoforms from which one could retain the most probable ones. This would be more comparable to the “top-4” approach proposed by the authors.

Author's Response

See response to R2 Q5. We now employ an exclusion threshold of 0.001, which was essential to remove 99% of the >1.7 million scores returned scanning +/- 5000 nt of the 86/88 variants able to assessed by Splice AI, in order to generate informative predictions and statistical metrics.

To clarify, our method enables SpliceAI to make multiple predictions via an analogous Top-4 approach. Splice AI often predicts activation of multiple cryptic splice-sites (Figure S1A).

We transparently declare the rules we apply are a customised interpretation of SpliceAI output.

21. 300K Top-4 relies on a genome annotation but no detail is given about this. The choice of the genome annotation could be critical. If the genome annotation of splice isoforms is too stringent, say, limited to a single major isoform per gene, then the most frequent non-annotated isoform (top-1) could often be a genuine, biologically functional splice isoform. At the other extreme, if the genome annotation is too inclusive then it can contain rare, non-functional splice isoforms, exactly those one would instead like to retain. On what basis did the authors decide which isoforms were “annotated”? Some assessment would be insightful.

Author's Response

This is an important point. Unfortunately, there is no national or international consensus of the transcript to be used for diagnostic reporting for each gene and condition – acknowledging different conditions can be associated with different transcripts arising from the same gene. Each diagnostic laboratory has their own reference list of transcript numbers they use for diagnostic reporting.

To cater for this complexity, 300K-RNA enables users to select the annotated transcript. All unannotated exon-skipping and cryptic-activation events, relative to this transcript, are returned in the Table.

We additionally specify within SpliceVault output when any of the top-ranked events are annotated events in alternate transcripts. It is important these annotated splicing events are encompassed within SpliceVault 300K-RNA output, so that clinical users can assess the likely pathogenicity of predominant or enhanced expression of an alternate annotated isoform.

Minor points

22. Line 50, the observation that aberrant splicing junctions are typically detected at weak levels in other samples was already reported by Kremer et al., Nat commun, 2017, with a quantification of the prevalence of this phenomenon consistent with the much more extensive and recent analyses by Dawes et al., 2021. Also, Kremer et al. also formulated the hypothesis of the present study described I. 64-66. This prior work should be cited in this respect.

Author's Response

This insight has been added and cited within our introduction. See lines 50-55.

23. Figure 2 is entitled "Accuracy" but what is assessed is sensitivity.

Author's Response

Emended.

24. Designing targeted RT-PCR follow-ups guided by the predictions have moderate value as they require specific primer designs and are blind to not-predicted isoforms. Would not the authors suggest RNA-seq, potentially with cycloheximide, in any case? This should be discussed.

Author's Response

Cycloheximide treatment was performed for all re-analysed variants, however no additional isoforms were detected. Mis-spliced *GSDME* and *OPHN1* transcripts were not compliant with NMD and *OPHN1* expression was too low in blood for RNA-seq analysis for agnostic interrogation. RNA sequencing was performed for the *SPG11* c.2317-13C>G variant, with expression levels insufficient to detect the double exon-skipping identified by RT-PCR (Bournazos et al., 2022)². For *EMD*, patient fibroblast-derived RNA is currently being used to compare long-read transcriptomics versus targeted sequencing of RT-PCR amplicons, plus and minus cycloheximide, and is part of a larger clinical evaluation of the diagnostic utility of long-read sequencing for RNA and DNA diagnostics.

1. Wilks, C. *et al.* recount3: summaries and queries for large-scale RNA-seq expression and splicing.

bioRxiv 2021.05.21.445138 (2021) doi:10.1101/2021.05.21.445138.

2. Bournazos, A. M. *et al.* Standardized practices for RNA diagnostics using clinically accessible specimens reclassifies 75% of putative splicing variants. *Genetics in Medicine* (2021)

doi:10.1016/j.gim.2021.09.001.

3. Abou Tayoun, A. N. *et al.* Recommendations for interpreting the loss of function PVS1 ACMG/AMP variant criterion. *Hum Mutat* **39**, 1517–1524 (2018).

Decision Letter, first revision:

15th Jun 2022

Dear Professor Cooper,

Your Analysis, "SpliceVault: predicting the precise nature of variant-associated mis-splicing." has now been seen by 2 referees. You will see from their comments below that while they find your work of interest, some important points are raised. We are interested in the possibility of publishing your study in *Nature Genetics*, but would like to consider your response to these concerns in the form of a revised manuscript before we make a final decision on publication.

As you will see, Reviewer #1 is mainly satisfied; there are a few other questions and requests for more comparisons.

Reviewer #2 thinks that there is potential here and that many of the technical concerns have been addressed. However, this reviewer is not fully satisfied and there are several main concerns. Overall, the comparisons and descriptions are found to be lacking. The reviewer continues to think that including clinical PVS1 ACMG codes is not a good idea.

We therefore invite you to revise your manuscript taking into account all reviewer and editor comments. Please highlight all changes in the manuscript text file. At this stage we will need you to upload a copy of the manuscript in MS Word .docx or similar editable format.

*2) If you have not done so already please begin to revise your manuscript so that it conforms to our Analysis format instructions, available [here](http://www.nature.com/ng/authors/article_types/index.html). Refer also to any guidelines provided in this letter.

[redacted]

We hope to receive your revised manuscript within four to eight weeks. If you cannot send it within this time, please let us know.

Sincerely,

Catherine Potenski, PhD
Chief Editor

Nature Genetics
1 NY Plaza, 47th Fl.
New York, NY 10004
catherine.potenski@us.nature.com
<https://orcid.org/0000-0002-4843-7071>

Referee expertise:

Referee #1: bioinformatics, computational biology

Referee #2: genomics, transcriptomics, rare genetic disease

Reviewers' Comments:

Reviewer #1:

Remarks to the Author:

The author conducted a series of data analyses, which addressed most of my concerns. However, based on the updated figures that used the ROC curve to evaluate their method and SpliceAI (Fig. 2B, Supplementary Fig. S1E-F), 300k-RNA indeed showed better performance than SpliceAI, while the advantage is not very evident. Especially for the second cohort 'specimens', SpliceAI even outperforms 300K-RNA. According to the results shown in Supplementary Fig. S2B, I believe multi-exon (double or even more exons) skipping is the main source of false positives of 300K-RNA Top4, while it only contributes minor to the sensitivity. As multi-exon skipping only accounts for 3% of the mis-splicing events (Line 103), I suggest that the author may show the comparison between 300K-RNA and SpliceAI by only using single exon-skipping and cryptic splice site. Anyway, even intron retention which accounts for 20% of the mis-splicing events (Line 102) was not included (Authors' response to Comments 6 of Reviewer #1). In addition, the author may need to reorganize different sections of the results to make the logic more fluent. i.e., Usually, people show the QC of the dataset first. The author may need to switch Fig. 2 and Fig. 3 and their corresponding sections of the results.

Reviewer #2:

Remarks to the Author:

Additional and extended comments:

1. Thank you for clarifying the pipeline used to process RNA-seq datasets. It is still not clear to me what impact an alternative processing pipeline would have on the results. In Dawes et al, the authors state that events with <3 samples supporting the event were discounted, and state that 44% of events were supported by 4 or fewer reads. There is no such method proposed here, can the authors clarify what methods are being used to reduce noise in the unannotated splice junctions, and in so doing, what impact this has on performance? In addition, can the authors show the impact of alternative pipelines for unannotated splice junctions on the 300K-RNA database and what impact this confers to accuracy of their methods. My suggestion would be Cummings (<https://www.ncbi.nlm.nih.gov/pmc/articles/PMC5548421/>), FRASER (<https://www.nature.com/articles/s41467-020-20573-7>) and LeafCutter

(<https://www.nature.com/articles/s41588-017-0004-9>).

2. SpliceAI uses a splice junction reference map during score calculation. The authors should merge unannotated and annotated junctions to calculate scores through spliceAI in order to understand how this impacts performance relative to (i) spliceAI alone, and (ii) 300K-RNA. In addition, the authors state that they use a window of +/-5000nts for SpliceAI score calculation – can they confirm that all aberrant splicing events would be captured within this window in both directions?

3. Long-read RNA sequencing datasets are available for GTEx samples – does incorporation of this data into 300K-RNA confer improvements to the detection of unannotated events?

4. The authors state in their rebuttal – ‘We confirm the number of mis-splicing events detected in each individual tissue in GTEx is correlated with the number of samples present for that tissue (Figure S4D)’ – how does this change with sequencing depth? How does this change by individual donor?

5. The authors have kept the usage of PVS1. Given the profile and wide readership of Nature Genetics I strongly suggest that such recommendations go through the appropriate groups highlighted in response to this issue (SpliceACORD and others) before they are published. Whilst this is a natural extension of the work presented here it is incredibly important that rule refinement is highly curated before uptaken by the diagnostic medicine community.

Author Rebuttal, first revision:

Response to Reviewer’s Comments:

We again appreciate the constructive suggestions of our Reviewers. Redressing their concerns has greatly improved the persuasiveness of our evidence by making us innovate better ways to present our data and support our arguments. Reviewer suggestions have also improved logic, flow and interpretability of the manuscript. In this second revision (R2), we genuinely try to meaningfully redress as many comments or concerns as possible that are technically and practically feasible. We would like to explain that we simply do not have the resources or datasets to do much of what is suggested.

We emphasize that the primary finding of this study is that the 300K-RNA Top-4* ranking method can reliably inform upon the probable nature of variant-associated mis-splicing. That our Reviewers are suggesting ways to improve the 300K-RNA Top-4* ranking method infers the evidence provided is persuasive, such that it can now be conceived how accuracy may be improved. We agree – and many suggestions offered mirror our own critical thinking. Once one accepts the dogma that *the spliceosome reproducibly makes the same mistakes* - a natural extension is to ask: how do we improve our cataloguing and ranking of these splicing mistakes? Can we quantify and incorporate intron retention

events into the ranking method? How can we improve short-read alignment to improve the fidelity of the splice junction dataset? Can we use long-read RNA-Seq to help us do this? Can ML methods improve short-read alignment? How can we validate which alignment method can best discriminate a bonafide from an artifactual splice-junction read?

Each of these are pertinent hypotheses that extend from the primary discovery of the predictive utility of 300K-RNA Top-4*ranking. We respectfully argue that *each* question above represents a substantial research project. It is impossible to do all of this for one paper.

Further refinement of the 300K-RNA Top-4* ranking method and predictive accuracy will require collective energies across the globe. We provide a point-by-point redress of your comments below.

Reviewer #1:

Remarks to the Author:

The author conducted a series of data analyses, which addressed most of my concerns. However, based on the updated figures that used the ROC curve to evaluate their method and SpliceAI (Fig. 2B, Supplementary Fig. S1E-F), 300k-RNA indeed showed better performance than SpliceAI, while the advantage is not very evident. Especially for the second cohort 'specimens', SpliceAI even outperforms 300K-RNA. According to the results shown in Supplementary Fig. S2B, I believe multi-exon (double or even more exons) skipping is the main source of false positives of 300K-RNA Top4, while it only contributes minor to the sensitivity. As multi-exon skipping only accounts for 3% of the mis-splicing events (Line 103), **I suggest that the author may show the comparison between 300K-RNA and SpliceAI by only using single exon-skipping and cryptic splice site.** Anyway, even intron retention which accounts for 20% of the mis-splicing events (Line 102) was not included (Authors' response to Comments 6 of Reviewer #1). In addition, **the author may need to reorganize different sections of the results to make the logic more fluent.** i.e., Usually, people show the QC of the dataset first. **The author may need to switch Fig. 2 and Fig. 3** and their corresponding sections of the results.

Author's response:

Overall Comments: Based on caveats and bias we define below; we do not feel comparing the sensitivity and PPV of Top-4* and SpliceAI after exclusion of double exon-skipping events is valid and we do not include these data within this R2 revision. However, to redress this point in a meaningful way, we emend the manuscript text to better explain the complexity related to double-exon skipping events (Lines 187-195), potential methods 300K-RNA Top-4* ranking and predictive accuracy may be improved (Lines 281-293) and an explanation why IR is not as great of a problem for consideration of PVS1 as being able to predict exon-skipping and cryptic splice-site activation with accuracy (Lines 278-282).

Revision R2. Results section. Lines 187-195. Red font = new text. Green font = moved text.

Analysis of features of false positives versus true positives in Top-4* revealed that false positives: 1) are identified in fewer samples in 300K-RNA (Figure S6A); 2) involve double exon-skipping, **which appears to be a mis-splicing outcome activated only rarely by splice-site variants** (Figure S6B). **4/5 detected double exon skipping events were Top-1 (among 14 cases with double skipping ranked Top-1) and 1/5 Top-2;** 3) show no difference in the length of the spliced-out region for exon-skipping events (Figure S6C) or cryptic distance (Figure S6D). **We identified one false positive that may be due to alignment issues in the short-read splice junction dataset: the first few nucleotides of two sequential exons are identical and split reads with only a few post-junction nts can map to exon-skipping or normal splicing.**

Revision R2. Discussion Section. Lines 281-293. Red font = new text. Green font = moved text.

Our current research is exploring whether high read depth RNA-Seq of manifesting tissues, or **alternative bioinformatic mapping strategies**, has the potential to further improve the precision of our Top-4* methodology. We suspect **the predictive accuracy of our ranking method may be improved further by:** 1) **a better understanding of contexts that influence variant-associated double exon skipping,** 2) **higher read depth RNA-Seq data** across the breadth of manifesting tissues in rare disorders (e.g. there is no data currently available for cochlear), 3) **incorporation of RNA-Seq data from human fetal samples to assess developmental alternative splicing,** 4) **improved bioinformatic methods to align short sequencing reads more accurately,** 5) **long-read RNA sequencing data, which will help validate (4) and also identify unannotated, deep-intronic pseudoexons,** 6) **use of cycloheximide to inhibit nonsense-mediate decay,** 7) **an ability to rank likelihood of variant-activated intron retention.** Consideration of Top-3 events **in specific manifesting tissues**, if shown to maintain > 90% sensitivity, could substantially improve PPV and clinical utility for variant classification.

Revision R2. Discussion Section. Lines 278-282. Red font = new text.

In addition, though intron retention cannot yet be quantified and ranked by 300K-RNA, Figure S8 shows that intron retention induces a frameshift or encodes a premature termination codon in all three frames for at least 97% introns in the Mendeliome and is therefore consistent with null outcomes in most instances.

Redress to specific points:

Point 1. Exclude double exon skipping and compare SpliceAI and Top-4.*

Though double exon-skipping occurs infrequently in this cohort – in all instances, there was diagnostic importance in identifying this event. Moreover, in several cases, double-skipping was the predominant variant-induced outcome. As we hope 300K-RNA will be used in a clinical setting, it is not sensible to exclude double skipping from the analyses as it introduces unnecessary bias. Further, as with any rare event, we are currently not sufficiently powered to hold certainty of the true incidence of variant-activated double exon skipping. We estimate that it will be between 2-10% and need a larger cohort of tested variants to show this. We cannot curate this information from other published studies, because unfortunately, many/most other studies do not deploy RNA testing strategies that are tailored to ensure detection of double-skipping or use of distal cryptic splice sites. It is exactly this deficiency that we hope dissemination of SpliceVault may mitigate.

- a) We clarify that our existing approach favours SpliceAI by excluding the 2 variants that SpliceAI cannot assess in precision-recall analyses (rather than offering the incorrect prediction – an approach also accepted to compare algorithms to mitigate instances when one algorithm cannot offer a prediction), as well as excluding all delta scores that do not align with any of our defined rules (the ‘other’ category).
- b) As we do not wish for our readers to infer that we recommend 0.011 as a threshold to offer a prediction of splice-altering outcomes, we add an extra sentence to clarify that this low threshold was selected only for this bespoke application; namely, to explore if SpliceAI’s deep learning can perceive the broader genomic features that may predict the probable nature of any elicited mis-splicing.

Revision R2. Results section. Lines 206-210

We also emphasise that while a SpliceAI threshold of 0.011 was effective in this specific context, to forecast the likely nature of any variant-elicited mis-splicing, we do not recommend use of a 0.011 delta score threshold as a prediction for mis-splicing as our evidence from experimentally confirmed splice neutral variants (³ and cases studied since) indicates this will yield a high false positive rate of > 50%.

Point 2. Switch order of Figure 2 and Figure 3 to improve logic

We appreciate this helpful suggestion. We have reversed the order and agree that the logic and flow is improved.

Reviewer #2:

Remarks to the Author:

Additional and extended comments:

1. Thank you for clarifying the pipeline used to process RNA-seq datasets. It is still not clear to me what impact an alternative processing pipeline would have on the results. In Dawes et al, the authors state that events with <3 samples supporting the event were discounted, and state that 44% of events were supported by 4 or fewer reads. There is no such method proposed here, **can the authors clarify what methods are being used to reduce noise in the unannotated splice junctions**, and in so doing, **what impact this has on performance?** In addition, can the authors show the **impact of alternative pipelines for unannotated splice junctions on the 300K-RNA database and what impact this confers to accuracy of their methods**. My suggestion would be Cummings

(<https://www.ncbi.nlm.nih.gov/pmc/articles/PMC5548421/>), FRASER

(<https://www.nature.com/articles/s41467-020-20573-7>) and LeafCutter

(<https://www.nature.com/articles/s41588-017-0004-9>).

Author's response:

Overall Comment: To clarify, FRASER, Leafcutter and Cummings et al. are not methods that differentiate between noise (splice junction artifact) and genuine splice junctions. FRASER, Leafcutter and Cummings et al. seek to identify what is different in a test specimen relative to a reference dataset. All three methods leverage splice junction information from aligned data and thus are similarly confounded by alignment artifacts. Their focus is not to catalogue and rank commonly occurring unannotated splicing events – this is what SpliceVault does. It is impossible to apply these tools to a dataset of 335,000 samples and we are not looking for splicing outliers.

FRASER and Leafcutter inspect annotated and unannotated splicing events to identify statistically significant splicing outliers (e.g. novel splice junction(s), higher or lower levels of an annotated or unannotated splice junction, evidence for allele bias.) These data can then be used to investigate potential causative variants within proximity of the splicing outliers. Cummings et al. focuses on

highlighting the clinical utility of RNA sequencing to identify the genetic basis for disease in a cohort of exome-negative individuals affected with Neuromuscular disorders. The Cooper team contributed 28/50 cases and are authors on this study.

Specific points raised.

1. *Methods to reduce noise in unannotated splice junctions?*
2. *Improve accuracy with incorporation of long-read RNA sequencing datasets?*
3. *Could use of other bioinformatic pipelines improve Top-4 ranking?*

We appreciate the previous suggestion by Reviewer 2 to switch 300K-RNA source data to that processed via the unified monorail pipeline. This was extremely helpful to minimise bias against unannotated splice junctions in the best way currently possible. We acknowledge there remains unannotated 'splicing noise' in 300K-RNA due to artifactual unannotated splice junctions due to errors or inherent ambiguity in alignment of short sequencing reads.

Neither FRASER nor Leafcutter are tools to reduce noise in unannotated splice junctions. Further, it is technically impossible for us to conduct the analyses suggested for 335,000 raw sequencing files.

- We do not have the raw sequencing files, only the publicly available splice-junction files.
- Even if we enquired with the Broad Institute if we can access the entire GTEx raw sequencing dataset, each RNA-Seq file is several gigabytes. We do not have means to transfer or store a dataset of this size, nor the computational processing power to analyse it via one of these methods.
- As a specific example, running FRASER for 46 RNA-Seq samples took **2 days** on the University of Sydney's high performance computing (HPC) network. If one extrapolates this to 43,000 GTEx RNA-Seq files or the entire 335,000 files in 300K-RNA dataset – this is **several years** of HPC time, notwithstanding the subsequent analysis time.

Challenges in aligning short RNA-sequencing reads is an intractable problem that continues to confound the short-read transcriptomics field (and has done for the last decade). Long-read sequencing data may ultimately provide the best means to validate alternate informatic alignment strategies for short-read RNA-Seq data - and thus also improve 300K-RNA splice junction Top-4* ranking. **We stress that there are currently only 88 long-read RNA-Seq files available via GTEx** for only a few tissue types, and substantial bioinformatic issues with long-read RNA sequencing data are yet to be resolved. Incorporation of long-read data into a 300K-RNA type ranking method holds enormous potential but is not yet ready.

In Dawes et al, we were able to rigorously evaluate precision-recall with additional rules such as “only events seen in > 3 samples” – because we had a dataset of ~5,000 variants activating cryptic donor splice sites in thousands of genes. Herein we have only 88 variants in 74 genes where many empirically detected events were represented by only a few splice-junction reads, and/or, detected in only a few samples. Therefore, at this time, application of rules such as ‘only events seen in > 3 samples’ or ‘only events detected by 3 or more split reads’ limits sensitivity. However, as our cohort of clinical cases with empirical splicing assay data grows larger and the sequencing depth and breadth of transcriptomic datasets informing a Top-4* style ranking method increases – it will become possible to critically evaluate any diagnostic benefit of applying such rules.

We agree inherently with the *essence* behind Reviewer 2’s suggestions: that alternate bioinformatic alignment strategies may improve event ranking and accuracy of the Top-4* approach. We try to meaningfully redress this suggestion by emending our results to note a specific example of a limitation with alignment of short reads (Revision R2. Results section. Lines 190-197) and we emend our discussion to state exactly how we believe the 300K-RNA approach may be improved into the future (Revision R2. Discussion Section. Lines 283-295). These excerpts are shown in response to our redress to Reviewer 1 comments.

2. SpliceAI uses a splice junction reference map during score calculation. The authors should merge unannotated and annotated junctions to calculate scores through spliceAI in order to understand how this impacts performance relative to (i) spliceAI alone, and (ii) 300K-RNA. In addition, the authors state that they use a window of +/-5000nts for SpliceAI score calculation – **can they confirm that all aberrant splicing events would be captured within this window in both directions?**

Author’s Response:

We acknowledge SpliceAI ‘out of the box’ can be influenced by the input transcript. However, in this study, delta scores +/- 5000 nt of the variant were derived using the raw pre-mRNA sequence as input and is not influenced by annotated versus unannotated splice junctions. Scores offered are not relative to, or influenced by, a splice junction reference map.

We emend our Methods to clarify these details. We are uncertain whether this comment may relate to the masking setting: therefore we clarify the masking setting applies only when the input is a variant - and is not relevant in our context where the input is the raw pre-mRNA sequence.

Revision R2. Results section. Lines 352-354. Red font = new text. Green font = moved text.

To adapt SpliceAI to the prediction of mis-splicing, we instead retrieved all Δ -scores +/- 5000 nt of each variant, using a **python script adapted from the SpliceAI GitHub (<https://github.com/Illumina/SpliceAI>)**. **As input, we used the pre-mRNA sequence either +/- 5000 nt of the variant or to the boundaries of the transcript, whichever was closer.** Two Δ -scores returned at each base (variant nucleotide versus reference nucleotide) generated **up to 20,002 Δ -scores per variant**, of which we excluded all Δ -scores ≤ 0.001 as neutral impact.

We emend our Results to 1) clarify how our usage of SpliceAI differs from default, 'out of the box' usage and 2) clarify how many events lie within the maximum SpliceAI window of +/- 5000 nt.

Revision R2. Results section. Lines 149-155

SpliceAI author's ¹⁹ high sensitivity threshold of $\Delta \geq 0.20$ predicted mis-splicing for 76/86 (88%) variants and for 63/86 (73%) variants with the high specificity threshold of $\Delta \geq 0.50$. **Preliminary investigations using the default SpliceAI window of 50 nt and high sensitivity threshold of $\Delta \geq 0.20$ identified 13/44 variant-activated cryptic splice-sites (only 19/44 activated cryptic splice-sites lie within 50 nt) and 3/73 exon skipping events. Therefore,**

we adapted SpliceAI to offer a **fuller** prediction of the nature of mis-splicing, by assessing all Δ -scores **generated by inputting pre-mRNA sequence +/-5000 nt of the variant (Figure 4A-C).**

Revision R2. Results section. Lines 150-153

According to these rules, SpliceAI predicts at least one mis-splicing event for all 86 variants and up to 31 predictions for a single variant (Figure 4D, Figure S4). **139/145 mis-splicing events elicited by the 86 variants lie within the maximum 5000 nt SpliceAI window (44/44 cryptics, 68/68 single-exon skipping events, 4/5 double-exon skipping events and 7/12 intron retention events).**

3. Long-read RNA sequencing datasets are available for GTEx samples – **does incorporation of this data into 300K-RNA confer improvements to the detection of unannotated events?**

As mentioned above, GTEx V9 encompasses only 88 specimens with long read RNA sequencing data available, from only a few biospecimen subtypes, currently insufficient for statistical evaluation via a Top-4 method. Further, long read is noisy and subject to its own bioinformatic challenges that we need to rigorously evaluate before promoting its use either to inform alignment strategies for short-read data and the Top-4* method – a substantial study in itself.

4. The authors state in their rebuttal – ‘We confirm the number of mis-splicing events detected in each individual tissue in GTEx is correlated with the number of samples present for that tissue (Figure S4D)’ – **how does this change with sequencing depth? How does this change by individual donor?**

We are unable to answer this question because it relies upon having RNA sequencing data for the same specimens conducted at different read depths using the same library preparation method. Unfortunately, this data is not available. We can informally communicate that we have conducted these analyses for a limited number of in-house specimens (blood, fibroblasts, muscle) that we have RNA-Seq data at different read depths – which *suggests* greater sequencing depth may have greater influence than sequencing breadth to capture the diversity of splice-junctions and reliably rank their prevalence. However, this sequencing was done on different machines with different library preparation methods (some used PolyA enrichment, some ribosomal RNA-depleted RNA) and is not robust enough to defend this question.

We apologise, but we cannot interpret what is being asked in “How does this change by individual donor” that is not answered by the concordance data we provided in Revision 1. The 300K-RNA ranking method is performed by individual donor and acceptor – additional figures provided in R1 show very high concordance in event ranking and sample count for each donor and acceptor across the entire Mendeliome.

5. The authors have kept the usage of PVS1. Given the profile and wide readership of Nature Genetics I **strongly suggest that such recommendations go through the appropriate groups highlighted in response to this issue (SpliceACORD and others) before they are published.** Whilst this is a natural extension of the work presented here it is incredibly important that rule refinement is highly curated before uptaken by the diagnostic medicine community.

We agree that a conservative and consultative approach is essential. However, it is our great preference to maintain the draft guidelines within supplemental Figure S9. We feel its inclusion as Supplemental information, rather than in the manuscript proper, means it is available for anyone who wants it, while not over-promoting it. The first question we receive from members of the Diagnostic Genomic Pathology community when we take them through the 300K-RNA Top-4* ranking for a variant is: “*how do you advise we use this information*”? In full knowledge of the caveats, the predictive insight garnered from 300K-RNA is being accepted and adopted quickly by the SpliceACORD Clinical Genetic community.

We provide Figure S8, that clarifies why a current inability of 300K-RNA to rank IR is of little consequence clinically for application of PVS1:

Lines 278-282.

Though intron retention cannot yet be quantified and ranked by 300K-RNA, Figure S8 shows that intron retention induces a frameshift or encodes a premature termination codon in all three frames for 97% introns in the Mendeliome and is therefore consistent with null outcomes in most instances.

We improve clarification and emphasis that these are *draft guidelines* that will be rigorously evaluated and refined over 2022-2023 by the Australian Consortium for RNA Diagnostics (SpliceACORD). It is likely that this SpliceACORD clinical evaluation will not be published for 2 years. We therefore argue respectfully, that by providing a starting point of draft guidelines, informed by expert users, we are enabling international groups to begin their own independent evaluation from the moment this paper is published. By doing so, in 2-years there may be multiple studies published that refute or refine these guidelines, such that an international consensus *if* and *how* to use SpliceVault to inform variant classification is reached. It is of vital importance to families with rare disorders that we do this as quickly as safely possible – which is best achieved via collective evidence from numerous, independent studies.

Revision R2. Discussion section. Lines 303-313

Informed by the current investigation, Figure S9 details our draft guidelines for possible use of 300K-RNA to assist application of the PVS1 criterion to essential splice-site variants, aligning closely with revised PVS1 guidelines¹⁰. Theorized consideration of intron retention and Top-4* events by genetic pathology workforces provides a pragmatic, evidence-based method to reliably assess variant-associated exon-skipping and/or cryptic splice-site use within a larger distance window of 600 nt. Over 2022-23, the Australasian Consortium for RNA Diagnostics (SpliceACORD)³ will rigorously evaluate the clinical accuracy and usefulness of Figure S9 draft guidelines for cases prospectively recruited into RNA

Diagnostic testing pipelines. Figure S7E indicates these draft guidelines will allow application of PVS1 for ~50% of ES variants (IR and Top-4* ≤ 2 in-frame events).

Decision Letter, second revision:

29th Aug 2022

Dear Professor Cooper,

Your Analysis, "SpliceVault: predicting the precise nature of variant-associated mis-splicing." has now been seen by 1 of the original referees. You will see from their comments below that while they find your work has improved, there are still important points raised. We remain interested in the possibility of publishing your study in Nature Genetics, but would like to consider your response to these concerns in the form of a revised manuscript before we make a final decision on publication.

Reviewer #2 was not able to submit a timely review, and as they had made substantive comments and requests in the last round, I then asked Reviewer #1 to comment on your responses to Reviewer #2. Please note that Reviewer #1 commented on all points raised by both referees, hence the section beginning "Reviewer #2".

Reviewer #1 is now satisfied with the changes regarding their review and has no further comments.

On the other hand, they think that some of your responses to Reviewer #2 are not quite satisfactory. They appreciate the difficulties of a full statistical characterisation of the reads in your analysis, but they point out that certain important statistics (e.g. splice junction supporting information) are vital for interpretation. They also suggest that the long-read sequencing could be analysed in a relatively simple way that would illustrate the value of such data over short reads.

We think both these suggestions are reasonable and we do agree that Point 1 requires clarification, given the importance of read depth for splicing analyses.

To guide the scope of the revisions, the editors discuss the referee reports in detail within the team, including with the chief editor, with a view to identifying key priorities that should be addressed in revision and sometimes overruling referee requests that are deemed beyond the scope of the current study. We hope that you will find the prioritized set of referee points to be useful when revising your study. Please do not hesitate to get in touch if you would like to discuss these issues further.

We therefore invite you to revise your manuscript taking into account all reviewer and editor comments. Please highlight all changes in the manuscript text file. At this stage we will need you to upload a copy of the manuscript in MS Word .docx or similar editable format.

*2) If you have not done so already please begin to revise your manuscript so that it conforms to our Analysis format instructions, available [here](http://www.nature.com/ng/authors/article_types/index.html). Refer also to any guidelines provided in this letter.

[redacted]

We hope to receive your revised manuscript within four to eight weeks. If you cannot send it within this time, please let us know.

Sincerely,

Michael Fletcher, PhD
Senior Editor, Nature Genetics

ORCID: 0000-0003-1589-7087

Reviewers' Comments:

Reviewer #1:

Remarks to the Author:

Reviewer #1:

My concerns have been addressed by the author in this revision.

Reviewer #2:

The author has addressed almost all my concerns, except for points 1, 3 and 4. The concern in point 4 is also related to point 1, regarding the impact of sequencing/read depth on the 300K-RNA Top4.

1,

I agree that neither FARSER nor LeafCutter are appropriate tools for unbiased quantification of mis-splicing events in 300K-RNA. In fact, the author collected splicing junctions from 'recount3', a previous study that claimed to use a so-called Monorail pipeline. The author should mention this clearly in the main text (Line 66-67). In addition, the 'recount3' has already been published in 2021 (Wilks et al. Genome Biology. 2021). However, the author still cited its bioRxiv version (Line 468-469). Even though 'recount3' might use a uniformed pipeline for splicing junctions processing, the author didn't address my concerns at the very beginning, regarding the 'read depth, quality and mapping thresholds'. The quality and mapping thresholds might be challenging for the author to include in the analysis, whereas the impact of sequencing/read depth should be considered. The author might also not be able to access the raw sequencing/read depth of each sample in 300K-RNA, but the total number of splicing junction reads in each sample should be easy to obtain. The author used sample numbers to rank the mis-splicing events, while it's not clear the threshold used by the author (what is the minimal number of reads to be considered as a supported splicing junction? 1, 2, or more?). This threshold might need to be adjusted by sequencing/read depth. Providing there are a total of 10 million and 1K splicing junction reads identified in sample A and sample B, respectively. For the same mis-splicing event, detection of one splicing junction read in sample B might be confident (1/1000), while one splicing junction read in sample A might be just noise (1/1000,000). The author should check the variation of splicing/read depth across different samples in 300K-RNA and take this into account in their analysis.

3,

As the author mentioned that it's challenging for them to incorporate long-read RNA sequencing data in 300K-RNA, it might be possible to investigate it in another direction. i.e. How many and what's the percentage of mis-splicing events in 300K-RNA Top4 are detected in the 88 specimens with long-read RNA sequencing data? I guess there should be at least some of them could be detected and it might become a problem if none of them could be found in the 88 specimens with long read RNA sequencing

data, either the long read sequencing is unable to identify lowly expressed mis-splicing events or it's an outcome only produced by Illumina short-read sequencing.

Author Rebuttal, second revision:

NG-AN59054R3: Author's response to Reviewer and Editorial Comments

Editorial Comments:

1. Reviewer #1 is now satisfied with the changes regarding their review and has no further comments.
2. Some responses to Reviewer #2 are not quite satisfactory.
 - a. Certain important statistics (e.g. splice junction supporting information) are vital for interpretation.
 - b. They also suggest that the long-read sequencing could be analysed in a relatively simple way that would illustrate the value of such data over short reads.

We think both these suggestions are reasonable and we do agree that Point 1 requires clarification, given the importance of read depth for splicing analyses.

Reviewers' Comments:

Reviewer #1:

My concerns have been addressed by the author in this revision.

Reviewer #2:

The author has addressed almost all my concerns, except for points 1, 3 and 4. The **concern in point 4 is also related to point 1, regarding the impact of sequencing/read depth on the 300K-RNA Top4.**

1,

I agree that neither FRASER nor LeafCutter are appropriate tools for unbiased quantification of mis-splicing events in 300K-RNA. In fact, the author collected splicing junctions from 'recount3', a previous study that claimed to use a so-called Monorail pipeline. The author should mention this clearly in the main text (Line 66-67). In addition, the **'recount3' has already been published in 2021** (Wilks et al.

Genome Biology. 2021). However, the author still cited its bioRxiv version (Line 468-469). Even though 'recount3' might use a uniformed pipeline for splicing junctions processing, the author didn't address my concerns at the very beginning, regarding the 'read depth, quality and mapping thresholds'. **The quality and mapping thresholds might be challenging for the author to include in the analysis, whereas the impact of sequencing/read depth should be considered.** The author might also not be able to access the raw sequencing/read depth of each sample in 300K-RNA, but the **total number of splicing junction reads in each sample should be easy to obtain.** The author used sample numbers to rank the mis-splicing events, while **it's not clear the threshold used by the author (what is the minimal number of reads to be considered as a supported splicing junction? 1, 2, or more?).** This **threshold might need to be adjusted by sequencing/read depth.** Providing there are a total of 10 million and 1K splicing junction reads identified in sample A and sample B, respectively. For the same mis-splicing event, detection of one splicing junction read in sample B might be confident (1/1000), while one splicing junction read in sample A might be just noise (1/1000,000). The author should check the variation of splicing/read depth across different samples in 300K-RNA and take this into account in their analysis.

Author's Response:

- 1.1. We have emended the citation to recount3.
- 1.2. Line 116 clarifies that we use a minimum threshold of 1 splice-junction read: *"Mis-splicing events detected at each splice-site in 300K-RNA are ranked by the number of samples in which at least 1 splice-junction read was detected (Figure 2C)."*
- 1.3. We agree, however, with the reviewer(s) and editor that read depth is important and further examine our set of true positives (TP) and false positives (FP).
 - a. In new figures, we show maximum reads for 300K-RNA events in any individual sample (new figure S6B), as well as mean reads for that event across all samples in which it was detected (new Figure S6C), tends to be higher for TP than FP.
 - b. Reviewer 2 suggests a junction detected in a single read in a sample with a lower number of reads overall may be more confident. We take that to mean that a splicing event detected in a higher proportion of transcripts for that gene in that sample is more likely to be a genuine splicing event and not technical noise introduced by alignment and other technical errors. New Figure S6D shows no significant difference between TP and FP for annotated splicing in samples where that 300K-RNA event was seen. If we find the maximum ratio between the unannotated event and annotated splicing (for the proximal splice junction), TP tend to have a higher max ratio than FP (new Figure S6E). However, there is no significant difference between TP and FP when you look at the mean ratio (new Figure S6F), showing that for many samples this doesn't hold true.
 - c. Importantly, **there remains a high amount of overlap between TP and FP in all analyses** - and even in our modest set of TP there are multiple events seen in only a one or few samples and represented by a maximum of 1 read.

- d. Our comprehensive analyses of TP versus FP reinforce the power of our Top-4 ranking method lies in the breadth of 300K-RNA that enables detection of stochastic, rare or ultra-rare events. To highlight how sample breadth is crucial to pick up these rare stochastic events, we provide new Figure 2E-F which shows:
- i. *Line 125: “Figure 2E shows the importance of sequencing breadth in 300K-RNA for detection of all 119 true positive exon-skipping and cryptic splicing events. Taking random subsets within the 335,663 source specimens shows sensitivity only begins to maximise with ~100,000 samples. Deeper scrutiny of the 119 true positive events shows, on average, each event is detected as a single splice-junction read in 78% samples with this event (not shown) – underpinning why all single read events are catalogued in 300K-RNA. Figure 2F reinforces the stochastic nature of these mis-splicing events, showing the Top-1 and Top-2 events around the splice sites affected by our 88 variants typically occur in mutually exclusive specimens – with both events seen, on average, in only 5% of samples where either event was seen.”*
- e. To further address comments relating to read-depth, quality and mapping thresholds, we scrutinized all TP and FP events for our cohort of 88 variants represented by 10 or fewer reads. We used the Broad Institute’s TGG viewer to scrutinize samples with at least one read for this event and recorded the maximum splice-junction overhang. There was no difference between TP and FP (TP mean max overhang 26.7 nt +/- 7.5, n=7; FP mean max overhang 27.0 nt +/- 9.9, n=13). Recount3 doesn’t provide overhang information so we can’t do anything larger scale. We do not wish to present this preliminary overhang analysis in our paper. Though we tried our best to dig into the available data, it is fraught with caveats: there are only 5 tissue types in TGG; only a subset of our TP and FP splice-junctions represented by < 10 samples could be examined; we cannot eyeball the read to check the overhang is not softclipped; rules related to overhang (and soft-clipping) are best implemented at the alignment step rather than post-alignment. We present this information only to show we have done every analysis possible with the available data. We respectfully maintain that the evidence-based Top-4 rules we have set are the best we can do at this time.

In summary, comprehensive analyses of FP vs TP does **not identify an evidence-base** to apply any additional rules that discern artifactual ‘splice junction noise’ from genuine, rare, stochastic splicing events. This remains an ongoing problem for RNA-seq technologies and outside the scope of this discovery and this paper. Further refinement of a Top-4 style ranking methods requires a larger validation cohort of clinical variants across a wider range of introns/genes and more reference RNA-Seq data across a diverse range of specimens. We transparently declare in our discussion ways a Top-4 style ranking method may prospectively be improved upon, though emphasize our current method provides 92% sensitivity with PPV of 29%: a pragmatic starting point and vast improvement over current practice that ‘guesses’ what a variant might do.

3,

As the author mentioned that it's challenging for them to incorporate long-read RNA sequencing data in 300K-RNA, it might be possible to investigate it in another direction. i.e. **How many and what's the percentage of mis-splicing events in 300K-RNA Top4 are detected in the 88 specimens with long-read RNA sequencing data?** I guess there should be at least some of them could be detected and it might become a problem if none of them could be found in the 88 specimens with long read RNA sequencing data, either the long read sequencing is unable to identify lowly expressed mis-splicing events or it's an outcome only produced by Illumina short-read sequencing.

- 3.1. We clarify it is not straightforward to extract splice junction (SJ) data from long-read RNA-Seq data. The only known tool, NanoSplicer (<https://doi.org/10.1093/bioinformatics/btac359>), requires raw Nanopore data (not available for GTEx V9) and is neither optimised for performance (~20 hrs for analysis of 1 specimen) nor is able to conduct a batch analysis. However, to meaningfully redress comments related to long-read RNA-Seq data, we instead reverse-engineered splice-junction information from GTEx v9 long read transcript data generated using FLAIR (PMID 35922509).
- 3.2. New Figure S9 shows splice-junctions concordantly and uniquely identified in long-read versus short-read RNA-Seq data from fibroblasts in an upset plot. We focus on fibroblasts because this is the most abundant specimen in the GTEx V9 dataset (22/88 sequenced specimens) and so we can make a meaningful comparison with short-read RNA-Seq data.
 - a. Long-read RNA-Seq (7M mean read depth, 740 nt average length) for 22 fibroblast specimens identifies **15%** (180K) of the total pool of 1.16M SJ detected in 504 GTEx V8 samples subject to short-read RNA-Seq (n=504 samples, 75 bp paired reads at 80M depth).
 - b. In-house data from 7 fibroblast specimens (150 bp paired reads, 100 – 200M read depth) subject to CHX treatment identifies **36%** (420K/1.16M) of all SJ, substantially more than DMSO treated specimens (340K/1.16M) or in two randomised sets of 7 fibroblast specimens from GTEx v8 (245K/1.16M).
 - c. This preliminary evidence indicates CHX treatment of human cell lines may substantially enhance our detection of unannotated mis-splicing events through inhibition of NMD.

In summary, long-read RNA-Seq data has a related set of challenges as short-read with respect to confidence of a detected splice-junction and whether to apply rules such as: only splice-junctions represented by more than one read, more than one sample, a minimum overhang etc. There is no viable tool able to do a batch extraction of splice junctions from Nanopore sequencing data; splice junctions currently need to be reverse engineered from the transcript information. Therefore, we

maintain there is currently insufficient GTEx V9 data for a SpliceVault long-read database and long-read RNA-Seq data is not yet ready to inform a Top-4 ranking approach.

Specific emendations to R3:

1. Modified Figure 2 with new sub figures E and F
2. Modified Figure S6 with new sub figures S6B-F.
3. New Figure S9.
4. Introduction Line 57: Clarify terminology of mis-splicing:
*“In Dawes et al., 2022¹¹, we analysed 5,145 variants activating cryptic splice-sites and established that 87% of activated cryptic splice-sites are those detected as rare, **unannotated splice junctions** in 40,233 RNA-Seq samples from GTEx12 and Intropolis15 (40K-RNA database¹¹). The key insight that cryptic donors activated by genetic variants are also seen as rare events in population-based RNA-Seq data, led us to explore whether other forms of variant-associated mis-splicing may be predicted by quantifying the relative prevalence of stochastic, natural, unannotated splicing events (**termed hereafter as mis-splicing**).”*
5. Results Lines 124-131: Clarify impact of sample breadth and describe Figure 2E-F:
“Figure 2E shows the importance of sequencing breadth in 300K-RNA for detection of all 119 true positive exon-skipping and cryptic splicing events. Taking random subsets within the 335,663 source specimens shows sensitivity only begins to maximise with ~100,000 samples. Deeper scrutiny of the 119 true positive events shows, on average, each event is detected as a single splice-junction read in 78% samples with this event (not shown) – underpinning why all single read events are catalogued in 300K-RNA. Figure 2F reinforces the stochastic nature of these mis-splicing events, showing the Top-1 and Top-2 events around the splice sites affected by our 88 variants typically occur in mutually exclusive specimens – with both events seen, on average, in only 5% of samples where either event was seen.”
6. Results Line 207: Clarify impact of read-depth and describe modified Figure S6:

“Analysis of features of Top-4 events reveals that relative to false positive, true positives tend to be: identified in more samples (Figure S6A); represented by more unannotated splicing reads (Figure S6B-C); with a higher maximum ratio of the unannotated event relative to read-depth for annotated splicing in any one sample (Figure S6E); relative to false positives. However, there was no significant difference between true and false positives in the mean reads of annotated splicing in samples where events were detected (Figure S6D) or in the mean ratio between unannotated and annotated splicing (Figure S6F); showing these are trends rather than rules. Double exon-skipping is rarely activated by splice-site variants (Figure S6G).”*

7. Discussion Line 320: Elaboration on SJ detection in long-read data and with CHX treatment in new Figure S9

“We suspect the predictive accuracy of our ranking method may be improved further by: 1) higher read depth RNA-Seq data across the breadth of manifesting tissues in rare disorders (e.g. there is no data currently available for cochlear), 2) incorporation of RNA-Seq data from human fetal samples to assess developmental alternative splicing, 3) use of cycloheximide to inhibit nonsense-mediate decay, 4) a better understanding of contexts that influence variant-associated double exon skipping, 5) an ability to rank likelihood of variant-activated intron retention, and 6) improved bioinformatic methods in sequencing read alignment. Long-read RNA may assist evaluation of new methods to improve fidelity of short-read alignment and reduce splice-junction artifacts in 300K-RNA. Figure S9 shows that long-read sequencing (7M mean read depth, 740 nt average length) of 22 fibroblast specimens available in GTEx V9 identifies only 15% (180K) of the 1.16M total splice-junctions detected in 504 GTEx fibroblast specimens (75 bp paired reads, 80M read depth). For comparison, in-house RNA-Seq data from 7 fibroblast specimens (150 bp paired reads, 100 – 200M read depth) subject to CHX treatment identifies 36% (420K/1.16M), substantially more than DMSO treated specimens (340K/1.16M) or in two randomised sets of 7 fibroblast specimens from GTEx v8 (245K/1.16M). This preliminary evidence indicates CHX treatment of human cell lines may substantially enhance our detection of unannotated mis-splicing events through inhibition of NMD.”

8. Methods Line 372: *The R package Snapcount²⁷ was used to retrieve information on individual samples for Figures 2E-F and S6B-F.*

Decision Letter, third revision:

18th Oct 2022

Dear Dr. Cooper,

Thank you for submitting your revised manuscript "SpliceVault: predicting the precise nature of variant-associated mis-splicing." (NG-AN59054R3). It has now been seen by the original Reviewer #1 - we decided that given Reviewer #2 did not submit a review in the previous round and your revision is in response to Reviewer #1's comments on Reviewer #2's last review, that it would be best to send it back just to this referee rather than both - and their comments are below. The reviewers find that the paper has improved in revision, and therefore we'll be happy in principle to publish it in Nature Genetics, pending minor revisions to satisfy the referees' final requests and to comply with our editorial and formatting guidelines.

Sincerely,

Michael Fletcher, PhD
Senior Editor, Nature Genetics

ORCID: 0000-0003-1589-7087

Reviewer #1 (Remarks to the Author):

I'm satisfied with the revision. One minor point is that the label of the x-axis in Fig S9B is not appropriately aligned.

Author Rebuttal, third revision:

Reviewers' Comments:

Reviewer #1:

Remarks to the Author:

I'm satisfied with the revision. One minor point is that the label of the x-axis in Fig S9B is not appropriately aligned.

We've amended the alignment

Final Decision Letter:

15th Dec 2022

Dear Dr. Cooper,

I am delighted to say that your manuscript "SpliceVault predicts the precise nature of variant-associated mis-splicing." has been accepted for publication in an upcoming issue of Nature Genetics.

Your paper will be published online after we receive your corrections and will appear in print in the next available issue. You can find out your date of online publication by contacting the Nature Press Office (press@nature.com) after sending your e-proof corrections. Now is the time to inform your Public Relations or Press Office about your paper, as they might be interested in promoting its publication. This will allow them time to prepare an accurate and satisfactory press release. Include your manuscript tracking number (NG-AN59054R4) and the name of the journal, which they will need when they contact our Press Office.

Acceptance is conditional on the data in the manuscript not being published elsewhere, or announced in the print or electronic media, until the embargo/publication date. These restrictions are not intended to deter you from presenting your data at academic meetings and conferences, but any

enquiries from the media about papers not yet scheduled for publication should be referred to us.

Please note that *Nature Genetics* is a Transformative Journal (TJ). Authors may publish their research with us through the traditional subscription access route or make their paper immediately open access through payment of an article-processing charge (APC). Authors will not be required to make a final decision about access to their article until it has been accepted. [Find out more about Transformative Journals](https://www.springernature.com/gp/open-research/transformative-journals)

Authors may need to take specific actions to achieve [compliance with funder and institutional open access mandates](https://www.springernature.com/gp/open-research/funding/policy-compliance-faqs). If your research is supported by a funder that requires immediate open access (e.g. according to [Plan S principles](https://www.springernature.com/gp/open-research/plan-s-compliance)) then you should select the gold OA route, and we will direct you to the compliant route where possible. For authors selecting the subscription publication route, the journal's standard licensing terms will need to be accepted, including [self-archiving-and-license-to-publish](https://www.nature.com/nature-portfolio/editorial-policies/self-archiving-and-license-to-publish). Those licensing terms will supersede any other terms that the author or any third party may assert apply to any version of the manuscript.

Please note that Nature Portfolio offers an immediate open access option only for papers that were first submitted after 1 January, 2021.

If you have not already done so, we invite you to upload the step-by-step protocols used in this

manuscript to the Protocols Exchange, part of our on-line web resource, natureprotocols.com. If you complete the upload by the time you receive your manuscript proofs, we can insert links in your article that lead directly to the protocol details. Your protocol will be made freely available upon publication of your paper. By participating in natureprotocols.com, you are enabling researchers to more readily reproduce or adapt the methodology you use. Natureprotocols.com is fully searchable, providing your protocols and paper with increased utility and visibility. Please submit your protocol to <https://protocolexchange.researchsquare.com/>. After entering your nature.com username and password you will need to enter your manuscript number (NG-AN59054R4). Further information can be found at <https://www.nature.com/nature-portfolio/editorial-policies/reporting-standards#protocols>

Sincerely,

Michael Fletcher, PhD
Senior Editor, Nature Genetics

ORCID: 0000-0003-1589-7087

Click here if you would like to recommend Nature Genetics to your librarian
<http://www.nature.com/subscriptions/recommend.html#forms>